# CANONICAL LATENT REPRESENTATIONS IN CONDITIONAL DIFFUSION MODELS

## ABSTRACT

Conditional diffusion models (CDMs) have shown impressive performance across a range of generative tasks. Their ability to model the full data distribution has opened new avenues for analysis-by-synthesis in downstream discriminative learning. However, this same modeling capacity causes CDMs to entangle the class-defining features with irrelevant context, posing challenges to extracting robust and interpretable representations. To this end, we introduce *Canonical Latent Representation Identifier* (CLARID), a training-free procedure to identify ***Cano**nical Latent **Rep**resentations* (CanoReps), latent codes whose internal CDM features preserve essential categorical information while discarding non-discriminative signals. When decoded, CanoReps produce representative samples for each class, offering an interpretable and compact summary of the core class semantics with minimal irrelevant details. Exploiting CanoReps, we develop a novel diffusion-based feature distillation paradigm, ***CaDistill***. While the student has full access to the training set, the CDM as teacher transfers core class knowledge only via CanoReps, which amounts to merely 10% of the training data in size. After training, the student achieves strong adversarial robustness and generalization ability, focusing more on the class signals instead of spurious background cues. Our findings suggest that CDMs can serve not just as image generators but also as compact, interpretable teachers that can drive robust representation learning.

## 1 INTRODUCTION

Diffusion models (DMs) excel at generative modeling of images (Dhariwal & Nichol, 2021; Ho et al., 2020; Rombach et al., 2022; Song et al., 2021a;b). When conditioned on class labels (Bao et al., 2023; Ho & Salimans; Peebles & Xie, 2023) or text prompts (Podell et al., 2024; Rombach et al., 2022), conditional diffusion models (CDMs) faithfully generate samples with desired characteristics of the condition. This generative capability has sparked a wave of analysis-by-synthesis approaches (Baranchuk et al., 2022; Chen et al., 2024b; Li et al., 2023b; Meng et al., 2024; Mukhopadhyay et al., 2024; Xiang et al., 2023; Yang & Wang, 2023; Zhang et al., 2024; Zhao et al., 2023b), where DMs are used to probe or improve downstream discriminative tasks. However, a key challenge remains: since DMs model the full data distribution, they often encode redundant or irrelevant information, which can obscure the discriminative signal. For example, in the *Tench* row of Figure 1, modifying the CDM latent code of the training sample changes the angler in the background while the fish remains almost unchanged, showing that the model encodes background cues that correlate with the class but not semantically essential to it. This entanglement between class semantics and extraneous factors limits interpretability and hinders the effective use of CDMs in representation learning. This motivates our central question:

*How can we identify the underlying core categorical semantics in a conditional diffusion model?*

We answer this by introducing **C**anonical **LA**tent **R**epresentation **ID**entifier (**CLARID**), a method for identifying the latent codes in CDMs that capture essential categorical information and filter out irrelevant details. We call the resulting latent codes **Cano**nical Latent **Rep**resentations (CanoReps). We begin with the assumption that the essential semantics of a class typically lie on a low-dimensional manifold embedded in the high-dimensional data space (Collins et al., 2018; Wang et al.; Zhang et al., 2019). We postulate that such semantic manifolds also exist within the latent space of diffusion models. Our key insight is that altering the latent code along the tangent directions of

the manifold, referred to as extraneous directions, modifies visual appearance without affecting class identity. We find that projecting out the extraneous directions in the latent space of CDMs effectively eliminates class-irrelevant factors such as background clutter or co-occurring objects from other categories. When projected back to the data space, CanoReps produce representative samples of each category, namely *Canonical Samples*, providing an intuitive and interpretable summary of the essential categorical semantics. Additionally, the internal CDM features of CanoReps, *i.e. Canonical Features*, contain mostly the core class information. We first validate our method in a toy hierarchical generative model, where CLARID recovers a low-dimensional class manifold while standard classifier-free guidance (CFG) produces dispersed, high-likelihood samples. Scaling up, we apply CLARID to ImageNet-pretrained CDMs and develop strategies for finding the projection time step and the number of extraneous directions. Our method also generalizes to text-conditioned DMs and is compatible with different diffusion samplers.

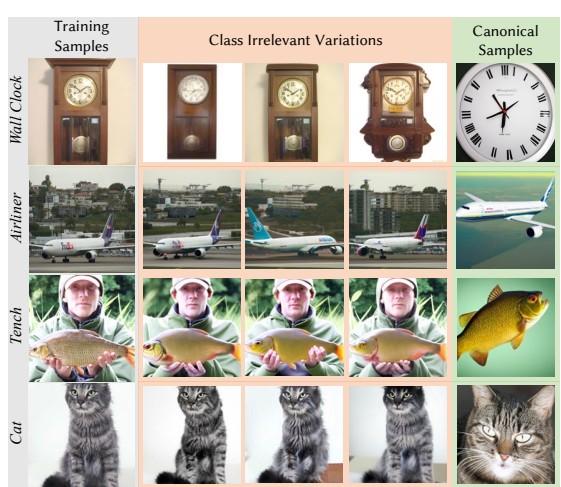

Building on the discovery of CanoRep, we propose a novel feature distillation paradigm, *CaDistill*. This method leverages the interpretable nature of CanoReps, which encapsulates the core semantics of each class, to supervise a student network. *CaDistill* aligns the student network's representations on both Canonical Samples and original training samples with Canonical Features using a novel feature distillation loss, which helps the student network encode the core class information. The student network's representations of the original training samples are also forced to be close to those of the Canonical Samples, treating them as anchors in the student's representation space. The student learns on the full training set, while the teacher CDM transfers the essential class knowledge by only exploiting CanoReps, which amounts to merely 10% of the training data in size. In contrast, existing state-of-the-art methods require transferring the teacher CDM's knowledge using the entire training set. *CaDistill* improves the student's adversarial robustness as well as the generalization capability on CIFAR10 (Krizhevsky et al., 2009) and ImageNet (Deng et al., 2009). Our contributions are as follows:

Figure 1: Conditional diffusion models (CDMs) encode both core class features and irrelevant context in a CDM. **Left**: training samples. **Middle**: samples obtained by modifying CDM latent codes that preserve the class label but alter class-irrelevant parts. **Right**: Canonical Samples produced by CLARID, which retain the class-defining content while removing extraneous context. CLARID benefits the extraction of robust and interpretable representations from the CDMs.

- We introduce *Canonical Latent Representation* (CanoRep) in CDMs—latent codes whose internal CDM features, *i.e. Canonical Features*, encapsulate core categorical semantics with minimal irrelevant signals. When decoded to the data space, these latent codes produce *Canonical Samples*, which serve as compact and interpretable prototypes for each class.

- To extract *CanoReps*, we propose **C**anonical **LA**tent **R**epresentation **ID**entifier (CLARID), a method that identifies these latent codes by projecting out non-discriminative directions in the CDM's latent space. The optimal configurations of CLARID are selected through a systematic analysis of the CDM's features.

- Leveraging the CanoReps, we develop a novel diffusion-based feature distillation paradigm, *CaDistill*. While the student is being trained on the full training set, the CDM as the teacher transfers essential class knowledge only via exploiting CanoReps, which amounts to merely 10 % of the data. The resulting student network achieves strong adversarial robustness and generalization performance, focusing more on the core class information.

## 2 RELATED WORKS

### 2.1 INTERPRETABILITY IN DIFFUSION MODELS

Recent research in diffusion models (DMs) reveals the interpretable semantic information in them. We categorize these efforts into two main groups. The first line of this work focuses on semantic editing by manipulating the reverse diffusion trajectory to produce semantically meaningful changes in generated images (Chen et al., 2024a; Haas et al., 2024; Kwon et al., 2023; Park et al., 2023a). Kwon et al. (2023) uncover a semantic space inside a pre-trained DM, termed $h$-space, where particular vector directions yield high-quality image editing results. Park et al. (2023a) analyze the latent input space, namely x-space, of DMs from a Riemannian geometry perspective. They define the pullback metric on x-space from the $h$-space Euclidean metric, obtaining certain vector directions in x-space that can yield semantic editing results. Chen et al. (2024a) provides more theoretical insights into this framework and extends it to local editing scenarios.

Another line of work leverages the attention mechanism (Vaswani et al., 2017) in DMs to interpret the conditional information (Chefer et al., 2023; Hertz et al.; Kim et al., 2023; Liu et al., 2024; Xu et al., 2023; Zhao et al., 2023a). Hertz et al. propose that the cross-attention map between the text prompts and image tokens in text-conditioned DMs encodes rich spatial cues. Building on this insight, subsequent studies analyze these attention maps to improve the precision and controllability of semantic image editing (Chefer et al., 2023; Kim et al., 2023; Liu et al., 2024). Other efforts investigate this property in visual recognition tasks such as semantic segmentation (Xu et al., 2023; Zhao et al., 2023a), using the attention maps to interpret the model's spatial reasoning. All existing methods either focus on image editing or are tailored to a specific DM architecture. We take the first step to uncover the underlying core class semantics in DMs without any supervision. Our method is compatible with different DM architectures and samplers.

### 2.2 DMS AS TEACHERS IN ANALYSIS-BY-SYNTHESIS

**DM-based feature distillation**. Recent works show that the intermediate features in DMs contain rich discriminative information (Baranchuk et al., 2022; Chen et al., 2024b; Li et al., 2023b; Meng et al., 2024; Mukhopadhyay et al., 2024; Xiang et al., 2023; Yang & Wang, 2023; Zhang et al., 2024; Zhao et al., 2023b). Here, we focus on the utilization of DMs as teachers in feature distillation frameworks. Li et al. (2023b) proposes a framework in which the intermediate features of the student network are aligned to those of a generative teacher, improving the student's performance on dense prediction tasks. Yang & Wang (2023) use reinforcement learning to select a proper diffusion timestep for distillation, enhancing the student's performance in image classification, semantic segmentation, and landmark detection benchmarks.

**DMs for data generation and augmentation**. DMs faithfully model the full training data distribution, which allows the generation of new training samples or augmentation of the existing ones (Azizi et al.; Bansal & Grover; Fu et al., 2024; Gowal et al., 2021; He et al., 2023; Islam et al., 2024; Sarıyıldız et al., 2023; Sehwag et al., 2022; Shama et al., 2024; Trabucco et al., 2024; Wang et al., 2023). Gowal et al. (2021) and Sehwag et al. (2022) improve the robustness of adversarially-trained classifiers by using diffusion-generated data. Bansal & Grover demonstrate that supplementing training data with diffusion-generated images leads to consistent gains on out-of-distribution test sets. Diffusion-generated data is also effective in data-scarce settings, *i.e.*, zero-shot and few-shot learning (Fu et al., 2024; He et al., 2023; Trabucco et al., 2024). Regarding data augmentation, Islam et al. (2024) propose blending images while preserving their labels using pre-trained text-to-image DMs (Rombach et al., 2022). Shama et al. (2024) utilize denoised samples for augmenting the training data, improving the generalization of downstream classifiers.

Despite the progress, current methods use raw diffusion features and outputs, which contain class-irrelevant information. Such irrelevant signals prevent the student from efficiently and accurately learning the class semantics, leading to vulnerable models. On the contrary, our method transfers the core class semantics using CanoReps to the student, enhancing its adversarial robustness and generalization capability. Notably, CanoReps amounts to only 10% of the original data in size.

## 3 METHOD

An overview of Canonical Latent Representation Identifier (CLARID) is shown in Figure 2. We describe the procedure of finding the CanoRep of a given sample in Section 3.2. We then provide the intuition of the effect of CLARID in a proof-of-concept experiment in Section 3.3.

Figure 2: Overview of Canonical Latent Representation Identifier (CLARID). Starting from a training sample $x_0$, we invert it using a CDM until $t = t_e$. We compute the extraneous directions using the method described in Section 3.2, and remove them from the inverted sample $x_{te}$ to obtain a CanoRep $\tilde{x}_{te}$. We then generate the Canonical Sample $\tilde{x}_0$ and extract the Canonical Feature at timestep $t = t_r$. The CDM receives the ground truth condition of $x_0$ in the whole process, here the *Tench* class.

## 3.1 PRELIMINARIES

Diffusion models (DMs) generate images by learning to reverse a fixed forward diffusion process (Ho et al., 2020). Let $x_0$ be a training sample and $t \in \{1, ..., T\}$ denote the diffusion time steps. A forward kernel $q(x_t \mid x_{t-1})$ progressively corrupts $x_0$ into a noisy sample $x_t$. A neural network $f_{\theta,t}(x_t)$, parameterized by $\theta$, is trained to approximate the reverse process $p_\theta(x_{t-1} \mid x_t)$. More details are in Appendix B. Certain parameterizations of the diffusion process (Karras et al., 2022; 2024b; Song et al., 2021a) allow for partial or full inversion of a given input sample, producing a noisy sample $x_t$ that preserves the semantic information of $x_0$ (Zhou et al., 2024a). Hereafter, we denote the inversion process of a sample by $F_{inv}$ and the corresponding decoding process as $F_{dec}$.

## 3.2 THE PROCEDURE OF CANONICAL LATENT REPRESENTATION IDENTIFIER (CLARID)

Given a sample $x_0$ as the seed, which belongs to class condition $c$, we first invert it to timestep $t_e$ via $F_{inv}$. Denote $x_{te}$ as the latent code of $x_0$ at $t_e$. In this latent space, we identify a set of extraneous directions that preserve class identity yet induce large changes. Concretely, let $f_\theta(\cdot)$ be the CDM's feature extractor at layer $l$ and timestep $t_e$. A first-order Taylor expansion around $x_{te}$ gives:

$$f_{\theta,te}^l(x_{te} + v) \approx f_{\theta,te}^l(x_{te}) + \nabla f_{\theta,te}^l(x_{te}) \cdot v = f_{\theta,te}^l(x_{te}) + J_{\theta,te}^l(x_{te}) \cdot v. \tag{1}$$

$J_{\theta,te}^l(x_{te})$ denotes the Jabobian of $f_{\theta,te}^l(x_{te})$. Hereafter, we drop $\theta$ and $l$ to avoid clutter. A vector $v$ that can cause large changes in the output carries some semantic information (Park et al., 2023a; Song et al., 2023), but such information is **not necessarily class-relevant**, as shown in Figure 1. The change caused by vector $v$ is defined as the L2-norm of the Jacobian vector product: $||J_{te}(x_{te}) \cdot v||_2$. Accordingly, the directions that lead to large changes in the output are the right singular vectors of $J_{te}$. When $f$ receives conditional information $c$, modifying the latent code along those directions tend to **preserve the class identity** while primarily altering the appearance of the decoded sample. By construction, these directions capture high-variance aspects of the image that the CDM can vary freely with the class condition unchanged. We show examples in Appendix D. We therefore remove the components of $x_{te}$ along those extraneous directions to obtain CanoRep $\tilde{x}_{te}$, by projecting $x_{te}$ onto the subspace which is orthogonal to these directions, as described in Eq.2.

$$\tilde{x}_{te} := x_{te, \perp V_k} = \left(\mathbf{I} - \mathbf{V}_k \mathbf{V}_k^\mathsf{T}\right) x_{te}. \tag{2}$$

Here, $\mathbf{V} = [v_1, v_2, ...]$ is the right singular vector matrix of $J_{te}$, and $k$ denotes the number of top singular vectors to be removed. We then apply $F_{dec}$ to $\tilde{x}_{te}$ to obtain one Canonical Sample $\tilde{x}_0$ for the input condition $c$. Note that the CDM is conditioned on $c$ in the whole procedure. Additional discussion on the choice of the layer index $l$ for computing $J_{te}$ is provided in Appendix E.

### 3.2.1 FINDING APPROPRIATE $t_e$ AND $k$

Choosing an appropriate $t_e$ is critical for CLARID to be effective. If $t_e$ is too small, *e.g.*, $t_e \approx \frac{T}{100}$, the synthesized sample barely changes and remains close to the original input. On the other hand, when $t_e$ is too large, the conditional signal becomes ineffective and fails to guide the generation (Kynkäänniemi et al., 2024; Zhang et al., 2022). To find the largest time step where the conditional signal is still able

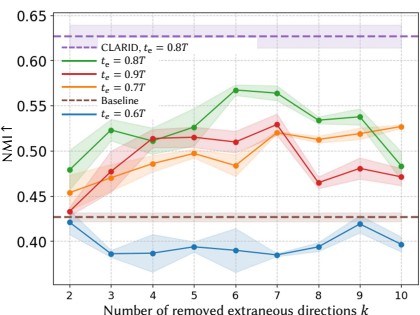

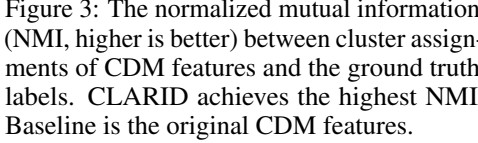

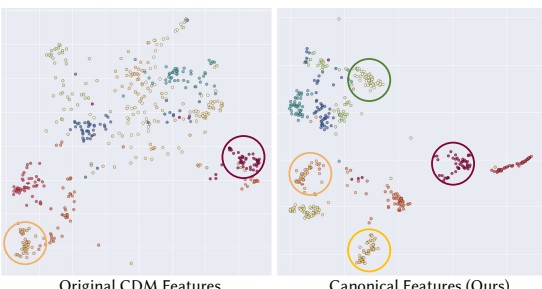

Figure 3: The normalized mutual information (NMI, higher is better) between cluster assignments of CDM features and the ground truth labels. CLARID achieves the highest NMI. Baseline is the original CDM features.

Figure 4: A 2D UMAP (McInnes et al., 2018) projection of the CDM feature space, showing clusters for ten classes. Colors indicate classes. CLARID yields more compact feature clusters than the original samples (green, yellow) and preserve the existing ones (red, orange).

to steer the CDM's output toward the desired class, we perform a two-stage sampling process starting from $p(\boldsymbol{x}_T) \sim \mathcal{N}(\boldsymbol{0}, \boldsymbol{I})$:

1. **Unconditional stage** ($T \leq t < t_e$): Forward the model without class conditioning, *i.e.*, $\boldsymbol{c} = \varnothing$.

2. **Conditional stage** ($t_e \leq t \leq 0$): The class condition is introduced and retained down to $t = 0$.

We generate $m$ samples for each condition in a given dataset, and measure the accuracy of pre-trained classifiers on the generated samples. We find the saturation point of the accuracy curve as the appropriate $t_e$. We show the accuracy curve and more details for an ImageNet $256 \times 256$-pretrained DiT (Peebles & Xie, 2023) and a Stable Diffusion model (Rombach et al., 2022) in Appendix H.1.

We fix the inversion time step $t_e$ across all samples, but the number of directions to discard should be adapted to each sample. To decide $k$, we examine how strongly the top-$k$ extraneous directions alter the sample's visual appearance, quantified by the explained variance ratio (EVR) $S_k = \sum_{i=1}^{k} \sigma_i^2 / \sum_{j=1}^{n} \sigma_j^2$, where $\sigma_i$ is the $i$th singular value of $\boldsymbol{J}_{te}$ and $n$ is a hyperparameter. Intuitively, the extraneous direction that leads to larger variations carries less core class semantics. We compute a sequence of the EVR $S_1, S_2, ..., S_n$ and set the elbow point of the sequence to be the optimal $k$ for a given sample. Compared to a fixed $k$, the adaptive choice will find the point where the effect of the extraneous directions diminishes. Visual examples illustrating the necessity of adapting $k$ for each sample, along with details of deciding $n$ and the calculation of the elbow point, are provided in Appendix H.2. It is worth noting that all hyperparameters in CLARID are **model-level** instead of sample-level. The hyperparameter selection takes less than 5h for DiT and 10h for SD.

**Empirical validation**. To demonstrate the effectiveness of our strategies on finding $t_e$ and $k$, we quantify the feature quality of a CDM and show that our strategies achieve the best one. Intuitively, by retaining the essential class features with minimal class-irrelevant information, the CDM features associated with the CanoReps (Canonical Features) should be more easily separable in the feature space. To show this, we first perform K-means clustering on the CDM features of 1000 samples from 20 different ImageNet classes, as well as their corresponding Canonical Features. We then quantify the feature quality using normalized mutual information (NMI) between the cluster assignments and the ground truth class labels. This method is training-free, hence efficient. We perform the analysis on an ImageNet $256 \times 256$-pretrained DiT-XL model (Peebles & Xie, 2023). The result is shown in Figure 3: each curve corresponds to a fixed inversion time step $t_e \in \{0.6T, 0.7T, 0.8T, 0.9T\}$, and the x-axis varies the number $k$ of removed extraneous directions; the "Baseline" curve shows the NMI of the original CDM features without removing any direction. The figure also plots error bars, which denote the standard deviation over three independent runs, each using a different set of 20 ImageNet classes. Removing different numbers of extraneous directions, *i.e.,* varying $k$, yields different NMI scores, whereas CLARID adaptively chooses $k$ for each sample and achieves the highest NMI. Qualitatively, CLARID produces feature clusters that are more compact than those formed by the original samples while preserving the existing clusters, as shown in Figure 4. The results of Stable Diffusion model in Appendix H.2.1 show a similar trend as DiT, which further validates our conclusion. We refer the readers to Appendix G and H for the details of the ImageNet20 dataset and experiment setup.

### 3.3 A PROOF-OF-CONCEPT EXPERIMENT

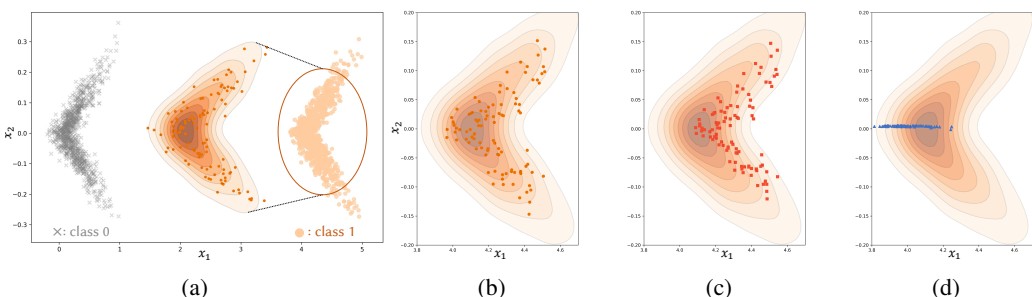

(a)  (b)  (c)  (d)

Figure 5: A toy example of CLARID. **(a)**: The samples of class 0 and class 1; **(b,c,d)**: The generated class-1 samples of **(b):** Plain $F_{dec}$, **(c):** CFG, and **(d):** CLARID, respectively, after applying $F_{inv}$ to the samples. CLARID produces Canonical Samples that lie on a 1D manifold inside class 1, offering an intuitive visual summary of the core class semantics.

We demonstrate the effect of CLARID with a simple yet illustrative example. Figure 5a shows samples generated from our toy generative process and the corresponding density map. The process first generates class-specific samples on a segment $L = \{(x_1, 0)|4y - 0.1 \leq x_1 \leq 4y + 0.1\}$, where $y \in \{0, 1\}$ denotes the class labels. It then adds class-independent noise to the points to generate the observed data. The samples from class 0 are included solely to introduce an inter-class contrast. We train a small class-conditional diffusion model on this data. After training, we perform CLARID on randomly selected samples from class 1. As a baseline, we take the latent codes obtained with $F_{inv}$ and perform classifier-free guidance (CFG), steering the generation process toward regions with higher class-1 likelihood. The results are shown in Figure 5. Notably, CLARID pushes most samples to a 1D manifold inside class 1, whereas CFG mainly steers the samples away from class 0. This low-dimensional manifold described by Canonical Samples can be regarded as a summarization of class 1 information in this case. The underlying structure revealed by Canonical Samples corresponds to one of the true generative processes for the data, which is the one used here. We refer readers to Section K in the Appendix for details of the toy experiments. Reliably recovering the exact generative model is intractable due to the identifiability issue (Locatello et al., 2019). The solution generally requires extra inductive bias in modeling data distribution, which we leave for future work.

### 3.4 QUALITATIVE RESULTS

Scaling up, we demonstrate the qualitative effect of CLARID for two CDMs, a class-conditioned DiT (Peebles & Xie, 2023) trained on the ImageNet $256 \times 256$ dataset and a Stable Diffusion 2.1 model (Rombach et al., 2022) trained on a subset of LAION-5B (Schuhmann et al., 2022). We use DDIM (Song et al., 2021a) as the sampler and use 100 diffusion steps for both inversion and decoding. Visual results are shown in Figure 6 (more in Appendix Q). We adopt classifier-free guidance (CFG) after removing extraneous directions in CLARID to ensure the data fidelity, and compare the results against pure CFG. The CanoReps visualized as Canonical Samples show that our method preserves the core class information in the original images, while CFG focuses on increasing the class-conditional likelihood. Occasionally, CLARID can select suboptimal $t_e$ and $k$, leading to artefacts in the generated images. The failure cases are discussed in Appendix H.3.

### 3.5 ON THE GENERALIZATION OF CLARID

While we focus on class-conditional DMs to develop the CLARID framework, it also extends naturally to text-conditioned models *e.g.* Stable Diffusion (Rombach et al., 2022), as shown in Figure 6 and Appendix F. Text prompts span a far richer semantic space than one-hot class labels, giving finer control over where the CanoReps lie. We show examples of the effect using different prompts on the same input in Appendix F.1. Understanding how this semantic structure shapes the located CanoReps is an interesting direction to explore. CLARID is also compatible with different DM samplers, as shown in Appendix F.2.

## 4 THE APPLICATION OF CANOREPS

As demonstrated in Section 3, CanoReps correspond to the essential class information learned by the CDMs. Building on this insight, we design a feature distillation framework for CanoReps, termed

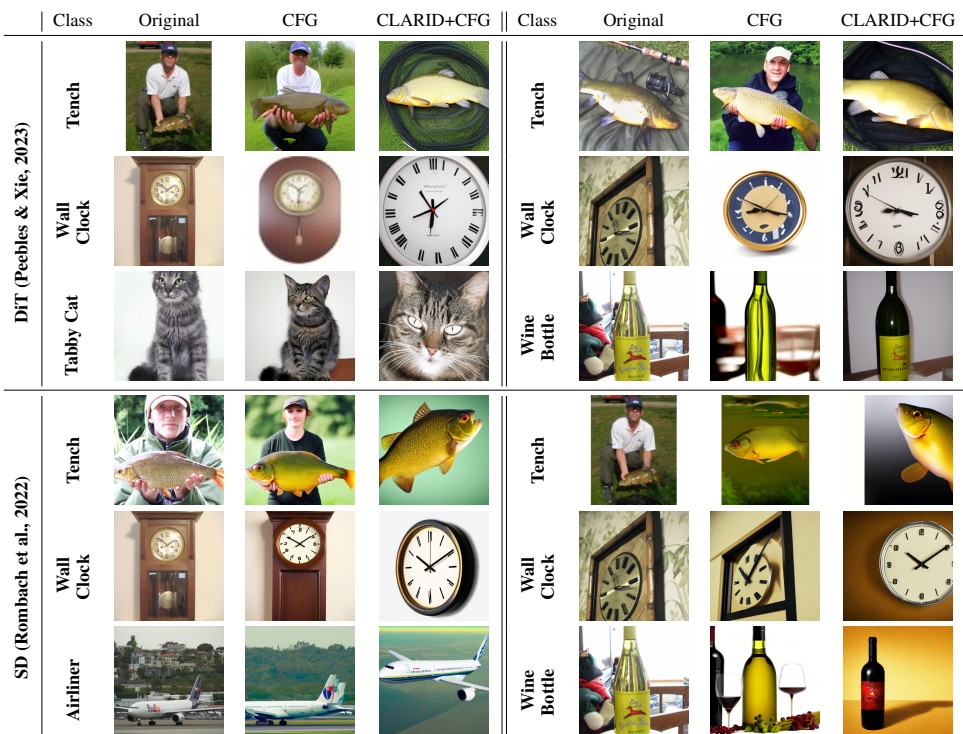

Figure 6: Comparison of classifier-free guidance (CFG) and CLARID on an ImageNet $256 \times 256$ DiT (Peebles & Xie, 2023) and Stable Diffusion 2.1 (SD) (Rombach et al., 2022). We use the following prompt template for SD: *a photo of <Class>* (Li et al., 2023a). CLARID focuses on identifying CanoReps to preserve the core class information, yielding Canonical Samples that provide an interpretable summary of the essential class semantics, whereas CFG aims at finding high-likelihood images. Two Canonical Samples from the *Tench* and *Wall Clock* class are presented to show that CanoReps do not collapse to a single constant vector. All "Original" images are taken from ImageNet. More visual results are in Appendix Q.

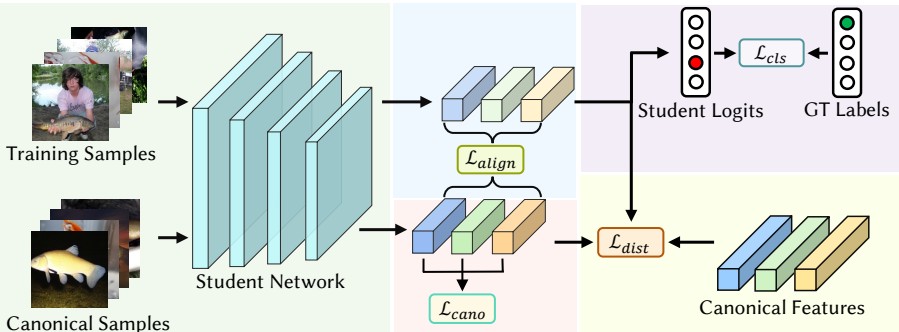

Figure 7: Overview of *CaDistill*. We align the student's features of training images to those of Canonical Samples using $\mathcal{L}_{align}$. The student is trained to discriminate between Canonical Samples in different classes by optimizing $\mathcal{L}_{cano}$. The CDM, as a teacher, provides Canonical Features for feature distillation with $\mathcal{L}_{dist}$. Finally, the student is supervised on the ground-truth labels via $\mathcal{L}_{cls}$.

**CaDistill**, as illustrated in Figure 7. Given a training batch $B = \{(\boldsymbol{x}_1, \boldsymbol{c}_1), (\boldsymbol{x}_2, \boldsymbol{c}_2), ..., (\boldsymbol{x}_b, \boldsymbol{c}_b)\}$ where $\boldsymbol{x}_i$ is an image and $\boldsymbol{c}_i$ is the class label, we first **randomly** select a CanoRep $\tilde{\boldsymbol{x}}_i$ corresponding to the category of each sample. The random sampling relaxes the constraint of the size equivalence between the training dataset and the set of CanoReps. Let $P_i = \{j \in \{1, 2, ..., b\} \mid \boldsymbol{c}_j = \boldsymbol{c}_i\}$ be the set of indices of samples in the batch with class label $\boldsymbol{c}_i$. We then compute student features $\boldsymbol{z}_i = g_\phi(\boldsymbol{x}_i) \in \mathbb{R}^d$, $\tilde{\boldsymbol{z}}_i = g_\phi(\tilde{\boldsymbol{x}}_i) \in \mathbb{R}^d$, where $g$ denotes the student network and $\phi$ is the set of its training parameters. The first component in **CaDistill**, $\mathcal{L}_{align}$, is designed to pull each training image feature towards all same-class Canonical Samples, and push away the ones from other classes. It is given in Eq. 3.

$$\mathcal{L}_{align} = -\frac{1}{b} \sum_{i=1}^{b} \frac{1}{|P_i|} \sum_{j \in P_i} \log \frac{\exp(\boldsymbol{z}_i \cdot \tilde{\boldsymbol{z}}_j / \tau)}{\sum_{k=1}^{B} \exp(\boldsymbol{z}_i \cdot \tilde{\boldsymbol{z}}_k / \tau)} . \tag{3}$$

$\tau$ is a temperature hyperparameter. Second, we encourage all Canonical Samples of the same class to cluster in the student's feature space, and separate those of different classes. It is achieved by minimizing $\mathcal{L}_{cano}$ in Eq. 4. When a Canonical Sample has no same-class positives, we simply optimize the denominator. See Appendix M.7.5 for more details.

$$\mathcal{L}_{cano} = -\frac{1}{b} \sum_{i=1}^{b} \frac{1}{|P_i| - 1} \sum_{j \in P_i, j \neq i} \log \frac{\exp(\tilde{\boldsymbol{z}}_i \cdot \tilde{\boldsymbol{z}}_j / \tau)}{\sum_{k \neq i} \exp(\tilde{\boldsymbol{z}}_i \cdot \tilde{\boldsymbol{z}}_k / \tau)} . \tag{4}$$

We perform an ablation study on the design of $\mathcal{L}_{cano}$ by replacing it with a classification loss on the Canonical Samples in Appendix M.7.5, demonstrating the advantage of the current $\mathcal{L}_{cano}$. For feature distillation, we transfer the structures of Canonical Features to the student via minimizing $\mathcal{L}_{dist}$. Specifically, we use the Centered Kernel Alignment (CKA) (Kornblith et al., 2019) metric to align the student's representations of both the training images and the Canonical Samples with the Canonical Features extracted from the CDM. Maximizing CKA, *i.e.* minimizing $\mathcal{L}_{dist}$, aligns the linear subspace spanned by the student's feature vectors with that of the teacher, effectively distilling the teacher's class-discriminative structure into the student. Denote the student feature matrices of training images and Canonical Samples as $\boldsymbol{Z} \in \mathbb{R}^{b \times d}$ and $\tilde{\boldsymbol{Z}} \in \mathbb{R}^{b \times d}$, respectively, and the Canonical Feature matrix as $\mathcal{A} \in \mathbb{R}^{b \times d'}$. The Canonical Features are extracted from a frozen CDM for all $\tilde{x}_i$ in a training batch.

$$\mathcal{L}_{dist} = \lambda_{cka} \log(1 - \mathrm{CKA}(\boldsymbol{Z}, \mathcal{A})) + (1 - \lambda_{cka}) \log(1 - \mathrm{CKA}(\tilde{\boldsymbol{Z}}, \mathcal{A})). \tag{5}$$

We find that using CKA for structural alignment outperforms existing diffusion-based feature distillation methods (Li et al., 2023b; Park et al., 2019; Romero et al., 2015; Yang & Wang, 2023) and refer the readers to Appendix M.1 for more details. Finally, the student is trained with the standard cross-entropy loss $\mathcal{L}_{cls}$ on ground-truth labels. The final loss function for **CaDistill** is given in Eq. 6.

$$\mathcal{L}_{CaDistill} = \mathcal{L}_{cls} + \lambda_{cs}(\lambda_{cf}\mathcal{L}_{align} + (1 - \lambda_{cf})\mathcal{L}_{cano}) + \lambda_{dist}\mathcal{L}_{dist} \tag{6}$$

## 4.1 CANOREPS IN PRACTICE

We conduct experiments of **CaDistill** on CIFAR10 (Krizhevsky et al., 2009) using a pre-trained UNet-based CDM (Ho et al., 2020; Xiang et al., 2023) and on ImageNet using the ImageNet $256 \times 256$-trained Diffusion Transformer (DiT) (Peebles & Xie, 2023). We choose two different DM architectures to demonstrate the applicability of our method. The CDM on CIFAR10 does not have an unconditional branch. The DMs are trained on the same dataset as the student models, hence no extra data is considered (Shama et al., 2024). We follow this principle to choose the baselines, comparing our method with the SOTA diffusion-based feature distillation (Yang & Wang, 2023) and data augmentation (Shama et al., 2024) methods. In addition, we design two important baselines. The first one, DMDistill, is to distill the raw space structure in CDMs using $\mathcal{L}_{dist}$ on all training samples, which represents the current mainstream idea of using DMs as teachers in feature distillation. It outperforms existing diffusion-based feature distillation losses (Li et al., 2023b; Park et al., 2019; Romero et al., 2015; Yang & Wang, 2023) (Appendix M.1). Second, for the CFGDistill baseline, we train the student model using the same framework of **CaDistill**, except that the images and features are obtained using CFG. For the CDM on CIFAR10, we sample new images for CFGDistill as this CDM lacks the unconditional branch and cannot perform CFG. We fix $t_r$, *i.e.* the feature extraction time step, for all methods that do not adaptively change it (Yang & Wang, 2023). For the student network, we use ResNet18 (He et al., 2016) on CIFAR10 and ResNet152 (He et al., 2016) on ImageNet, two well-established convolutional neural network baselines (Yang & Wang, 2023). ResNet50 results on

Table 1: Quantitative comparisons of *CaDistill* and baselines on CIFAR-10 (Krizhevsky et al., 2009) (ResNet-18) and ImageNet (Deng et al., 2009) (ResNet-152). $\mathcal{D}$: Dataset. Adversarial robustness benchmarks: PGD (Madry et al., 2018), CW (Carlini & Wagner, 2017), APGD-DLR / APGD-CE (Croce & Hein, 2020); Evaluations of generalization ability : Corruption (CIFAR10-C and ImageNet-C) (Hendrycks & Dietterich, 2018), ImageNet-A (Djolonga et al., 2021), ImageNet-ReaL (Beyer et al., 2020), CIFAR10.1 (Recht et al., 2018), CIFAR10.2 (Lu et al., 2020). $\text{Data}_{\text{DM}}$ is the portion of data for which the DM acts as teacher. Higher is better. Values lower than the vanilla model are in red. $\dagger$: the model relies on unconditional DMs, or the training cannot be performed, see Appendix M.6.

| $\mathcal{D}$ | Model | $\text{Data}_{\text{DM}}$ | Clean | PGD | CW | APGD-DLR | APGD-CE | Corruption | C10.1/IM-A | C10.2/IM-ReaL |
|---|---|---|---|---|---|---|---|---|---|---|
| CIFAR10 | Vanilla | / | 92.4 | 33.4 | 20.9 | 34.2 | 32.0 | 76.1 | 82.3 | 78.0 |
| | SupCon (2020) | / | 92.7 | 29.1 | 16.8 | 34.8 | 29.9 | 76.9 | 81.8 | 78.3 |
| | RepFusion$^{\dagger}$ (2023) | 100% | 92.7 | 30.3 | 17.2 | 32.1 | 29.2 | 75.3 | 83.4 | 78.5 |
| | DMDistill | 100% | 92.9 | 41.3 | 32.8 | 38.7 | 36.0 | 76.7 | 83.2 | 79.1 |
| | CFGDistill | 10% | 92.9 | 43.7 | 37.3 | 40.9 | 39.0 | 76.6 | 83.8 | 79.5 |
| | *CaDistill* | 10% | **93.1** | **47.9** | **43.1** | **44.1** | **43.3** | **77.7** | **84.5** | **79.7** |
| ImageNet | Vanilla | / | 79.3 | 22.0 | 20.7 | 24.7 | 23.3 | 54.2 | 14.1 | 85.5 |
| | DiffAug (2024) | 100% | **79.6** | 24.5 | 21.7 | 26.3 | 25.7 | **55.5** | 13.0 | 85.6 |
| | DMDistill | 100% | 79.4 | 23.8 | 23.1 | 25.7 | 24.9 | 51.8 | 12.5 | 85.6 |
| | CFGDistill | 10% | 79.1 | 27.6 | 30.8 | 30.2 | 28.2 | 54.1 | 13.7 | 85.6 |
| | *CaDistill* | 10% | 79.5 | **29.7** | **32.2** | **32.5** | **29.6** | 55.1 | **14.9** | **85.8** |

ImageNet are in Appendix M, showing a similar trend as ResNet152. We evaluate the adversarial robustness as well as the generalization ability, including in-distribution and out-of-distribution, of the student. We refer readers to Appendix L for more training and evaluation procedure details. For all *CaDistill* experiments, CLARID is used as a single, offline preprocessing step. This avoids any online computational burden, making it efficient for scaling up. More details of the computational costs and scalability of CLARID are in Appendix I.

The results are shown in Table 1. *CaDistill* consistently improves the adversarial robustness and generalization ability of the student, while the main effect of SOTA methods is improving the clean accuracy or a single aspect of the robustness. A more detailed discussion of the results, including the significance of such a consistent improvement and the importance of both our Canonical Samples and loss functions, is in Appendix N. Moreover, we consider an interesting and challenging baseline that is capable of extracting class semantics, *i.e.,* using the class token in an ImageNet-pretrained ViT (Touvron et al., 2021; 2022) as the teacher. We show the results in Appendix M.4. *CaDistill* is effective when the student has transformer-based vision backbones, as shown in Appendix M.8.

The student, trained with *CaDistill*, focuses more on the core class signal. We demonstrate this by a test on the Backgrounds Challenge (Xiao et al.), where the background of an image that is irrelevant to the class identity is either removed or shuffled. The results are given in Table 2. *CaDistill* improves the student performance on BG-Rand and Only-FG while maintaining the accuracy on the Original and BG-Same splits, indicating a mitigation in the model's dependence on the spurious background cues for classification. A more detailed discussion of the results is in Appendix M.5.

Table 2: Results on the Backgrounds Challenge (Xiao et al.). Higher is better. See Appendix M.5 for details.

| Model | Original | BG-Same | BG-Rand | Only-FG |
|---|---|---|---|---|
| Vanilla | 96.9 | 91.3 | 85.6 | 89.6 |
| DiffAug (2024) | **97.0** | 90.5 | 85.2 | 89.2 |
| DMDistill | **97.0** | 90.0 | 84.6 | 88.1 |
| CFGDistill | 97.0 | **91.5** | 85.4 | 89.4 |
| *CaDistill* | 97.0 | **91.5** | **86.5** | **90.4** |

**Ablation studies.** Appendix M.7 include the ablation analysis on *CaDistill*, including:

- **Number of CanoReps**. We demonstrate that 10% of CanoReps is sufficient for achieving competitive performance, implying the low-dimensionality property of the class manifolds in CDMs.

- **Necessity of $\mathcal{L}_{align}$, $\mathcal{L}_{cano}$, $\mathcal{L}_{dist}$, and their balance.** We empirically conclude the effects of all loss functions and choose the optimal weighting schemes, including $\lambda_{cs}, \lambda_{cf}, \lambda_{dist}, \lambda_{cka}$.

- **CFG magnitude.** We perform CFG after obtaining CanoReps, and show a proper choice of its magnitude. Importantly, a higher CFG magnitude does not contribute to better performance,

indicating that ***CaDistill*** is not merely providing a converging prior on the student features. We also provide a discussion on this topic in Appendix G.3.

## 5 DISCUSSION AND LIMITATION

**Discussion.** Our work offers a fresh perspective on representation learning: rather than designing objectives to enforce invariance or discriminative properties, we *subtract* non-discriminative components from DM's rich feature space to reveal CanoReps that encode core class semantics. This complements the current representation learning research. We further identify several properties, including its relationship to generative similarity (Marjieh et al., 2024), and potential applications of CLARID in Appendix O.

**Limitation.** CLARID has certain limitations. It can occasionally select a suboptimal projection time step, $t_e$, or the total number of extraneous directions considered, $n$, as discussed in Section 3.4 and Appendix Section H. We discuss potential solutions to those issues in Appendix H.3. Calculating the singular vectors of the Jacobian of CDMs is computationally intensive. Whether the application of CanoReps can be effective on larger-scale problems, *e.g.,* ImageNet22K, remains a question.

## 6 CONCLUSION

We introduce **C**anonical **LA**tent **R**epresentation **ID**entifier (CLARID), a principled method to uncover the core categorical information encoded in pre-trained conditional diffusion models (CDMs). By removing extraneous directions from a sample's latent code, CLARID produces *CanoRep*, whose internal feature—Canonical Feature—distills the class-defining semantics of each category. Decoding CanoReps yields Canonical Samples that offer an interpretable and compact summary of the class. Quantitatively, Canonical Features form more compact and easily separable clusters in CDM feature space than the original inputs. Building on CanoReps, we have proposed ***CaDistill***, a diffusion-based feature distillation framework. The teacher CDM transfers core class semantics to the student only via CanoReps, which is equivalent to merely 10% of the original data, while the student is being trained on the full training set. The student achieves strong adversarial robustness and generalization, focusing more on the true class signal instead of spurious background cues than the original model. Together, CLARID and ***CaDistill*** demonstrate that CDMs can be transformed from black-box generators into compact, interpretable teachers for robust representation learning.

## 7 REPRODUCIBILITY STATEMENT

We disclose all the method and experiment details regarding CLARID in Section 3.2, Appendix E, G, H. For giving an intuitive understanding of CLARID, we provide a 2D toy model whose details are in Appendix K. The application of CLARID is **CaDistill**, for which we provide detailed training and evaluation settings in Appendix L.

## 8 ETHICS STATEMENT

The image generators used in our experiments may sometimes contain wrong information. That said, we are not focusing on generating high-fidelity images or improving the synthesizing quality.

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

APPENDIX: TABLE OF CONTENTS

## A  AN OVERVIEW OF THE PAPER

We outline the paper in Figure 8.

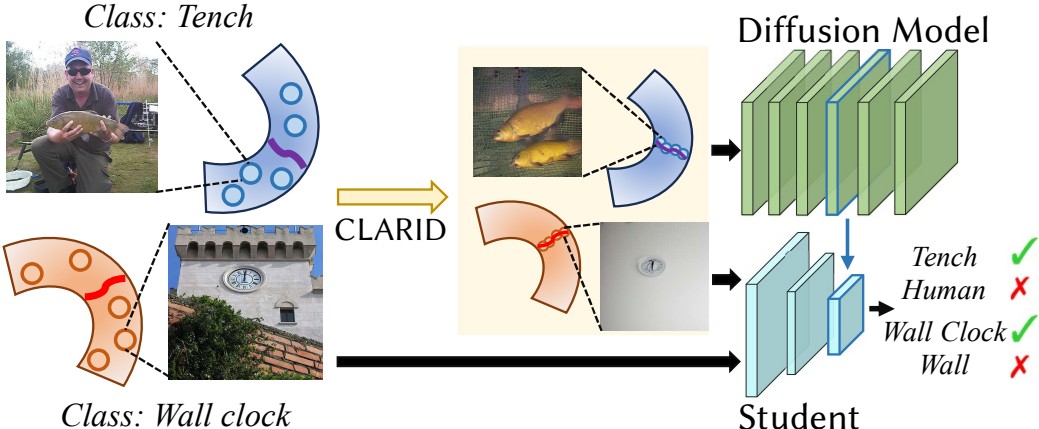

Figure 8: An overview of our paper. We identify the *CanoReps* via **C**anonical **LA**tent **R**epresentation **ID**entifier (CLARID) inside conditional diffusion models (CDMs) as a compact family of latent codes that contain the core class information with minimal class-irrelevant signals. These prototypes lie on low-dimensional manifolds ( red $\sim$ and purple $\sim$ tildes) within the latent space of CDMs. Leveraging the CanoReps, we design a diffusion-based feature distillation paradigm, improving the adversarial robustness and generalization of downstream models in image classification.

## B  DETAILED PRELIMINARIES

### B.1  DENOISING DIFFUSION PROBABILISTIC MODEL (DDPM)

DDPM (Ho et al., 2020) models the generation process as an inversion of a fixed forward Gaussian diffusion $q(\boldsymbol{x}_{1:T} \mid \boldsymbol{x}_0) := \prod_{t=1}^{T} q(\boldsymbol{x}_t \mid \boldsymbol{x}_{t-1})$. The forward kernel $q(\boldsymbol{x}_t \mid \boldsymbol{x}_{t-1})$ is described in Eq. 7.

$$q(\boldsymbol{x}_t \mid \boldsymbol{x}_{t-1}) = \mathcal{N}\big(\boldsymbol{x}_t;\ \sqrt{1-\beta_t}\,\boldsymbol{x}_{t-1},\ \beta_t\boldsymbol{I}\big) =:= \mathcal{N}\Big(\sqrt{\tfrac{\alpha_t}{\alpha_{t-1}}}\,\boldsymbol{x}_{t-1},\ (1-\tfrac{\alpha_t}{\alpha_{t-1}})\boldsymbol{I}\Big), \quad (7)$$

where $\{\beta_t\}_{t=1}^{T}$ is the variance schedule, $\alpha_t = \prod_{k=1}^{t}(1-\beta_k)$. The inversion process is defined as $p_\theta(\boldsymbol{x}_{0:T}) := p(\boldsymbol{x}_T) \prod_{t=1}^{T} p_\theta(\boldsymbol{x}_{t-1} \mid \boldsymbol{x}_t)$, where $\boldsymbol{x}_T \sim \mathcal{N}(\mathbf{0}, \boldsymbol{I})$. $\theta$ denotes the parameter set of a trainable noise predictor $\boldsymbol{f}_\theta$. A single reverse step is formalized in Eq. 8.

$$\boldsymbol{x}_{t-1} = \frac{1}{\sqrt{1-\beta_t}}\Big(\boldsymbol{x}_t - \frac{\beta_t}{\sqrt{\alpha_t}}\,\boldsymbol{f}_{\theta,t}(\boldsymbol{x}_t)\Big) + \sqrt{\beta_t}\,\boldsymbol{\epsilon}_t, \quad \boldsymbol{\epsilon}_t \sim \mathcal{N}(\mathbf{0}, \boldsymbol{I}). \quad (8)$$

$\boldsymbol{f}_{\theta,t}$ means that the noise predictor receives $t$ as a conditional input.

### B.2  DENOISING DIFFUSION IMPLICIT MODELS (DDIM)

DDIM (Song et al., 2021a) proposes a non-Markovian forward diffusion process, implying the parametrization described in Eq. 9.

$$q_\xi(\boldsymbol{x}_{t-1} \mid \boldsymbol{x}_t, \boldsymbol{x}_0) = \mathcal{N}\Big(\sqrt{\alpha_{t-1}}\,\boldsymbol{x}_0 + \sqrt{1-\alpha_{t-1}-\xi_t^2}\,\frac{\boldsymbol{x}_t - \sqrt{\alpha_t}\,\boldsymbol{x}_0}{\sqrt{1-\alpha_t}},\ \xi_t^2\,\boldsymbol{I}\Big), \quad (9)$$

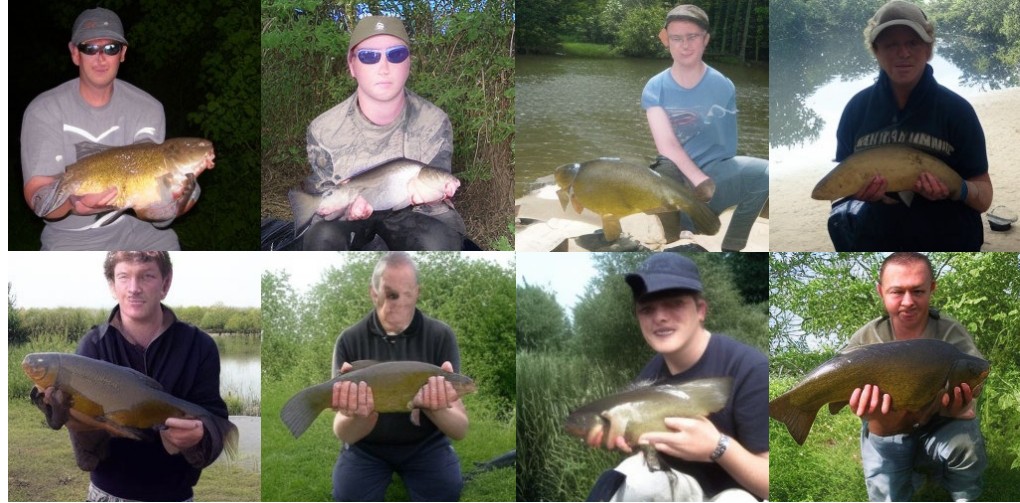

Figure 9: Random samples generated by DiT $256 \times 256$ (Peebles & Xie, 2023) when conditioned on the *Tench* class (0th class in ImageNet).

where $\xi_t = \eta \sqrt{\frac{1-\alpha_{t-1}}{1-\alpha_t}} \sqrt{1 - \frac{\alpha_t}{\alpha_{t-1}}}$. The reverse step is described in Eq. 10.

$$\boldsymbol{x}_{t-1} = \left( \frac{\boldsymbol{x}_t - \sqrt{1-\alpha_t}\, \boldsymbol{f}_\theta(\boldsymbol{x}_t)}{\sqrt{\alpha_t}} \right) + \sqrt{1 - \alpha_{t-1} - \xi_t^2}\, \boldsymbol{f}_\theta(\boldsymbol{x}_t) + \xi_t\, \boldsymbol{\epsilon}_t. \quad (10)$$

DDIM and other parametrizations of the diffusion process (Karras et al., 2022; 2024b) can perform inversion on the input sample, retaining certain semantic information of it at $t = T$ (Zhou et al., 2024a).

## C  RANDOM SAMPLES FROM *Tench* CLASS GENERATED BY DiT

We sample random images using DiT-XL $256 \times 256$ (Peebles & Xie, 2023) from the *Tench* class in ImageNet. We use a classifier-free guidance scale of 1.5, which is the one used in the original paper that achieves the best generation quality. The results are shown in Figure 9. Most images that we observe contain an angler.

## D  MOTIVATION AND THEORETICAL EXPLANATION OF CLARID

**Motivation.** We show that the extraneous directions (Section 3.2) carry semantics that are not related to the class identity in Figure 10. We adopt the method proposed by Park et al. (2023a). The editing focuses mostly on the background and preserves the class identity, as long as the movement does not orthogonalize the latent code and the editing direction. This phenomenon motivates our experiments.

**Theoretical explanation.** We provide an explanation of CLARID from the perspective of the change of mutual information. We denote $X$ as the image, $Y$ as the corresponding label, and $Z$ as the CDM feature at time step $t_e$. Because the CDM is trained to model the full $p(X|Y)$, its internal feature $Z$ will contain a part of the information that is predictive of $Y$, and the remaining variability in $X$ that is largely independent of $Y$, such as background, co-occurring objects, and style. We have the following decomposition of the mutual information $I(Z; X, Y)$:

$$I(Z; X, Y) = I(Z; X) + I(Z; Y|X) = I(Z; Y) + I(Z; X|Y)$$

We can get $I(Z; X) = I(Z; Y) + I(Z; X|Y) - I(Z; Y|X)$. $I(Z; Y)$ measures how much of the latent is aligned with class semantics. $I(Z; X|Y)$ represents how much extra information $Z$ contains about the specific $X$ given $Y$. $I(Z; Y|X)$ is zero, assuming that the label is correct and unique (and hence $I(Z; Y|X) \leq H(Y|X) = 0$) in the dataset. CLARID projects the full $Z$ onto the orthogonal complement of the top Jacobian singular directions, as introduced in Section 3.2. By

| Original | $E_1$ | $E_2$ | $E_3$ |
|---|---|---|---|

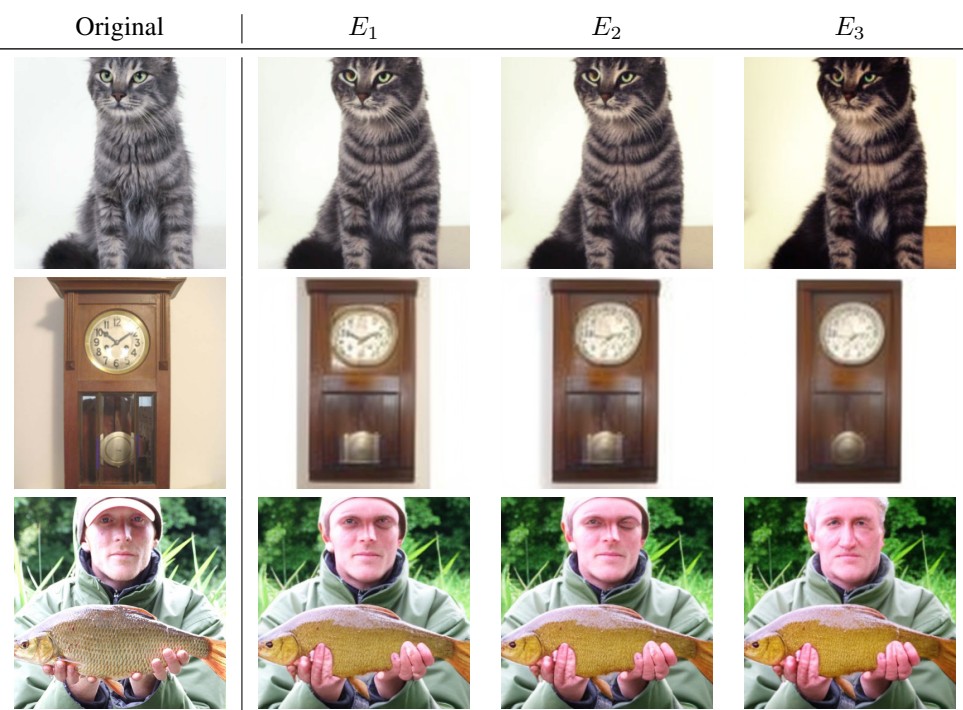

Figure 10: Moving along extraneous directions (Section 3.2) will alter the appearance of the image while preserving the class identity. $E_i$ represents the editing strength.

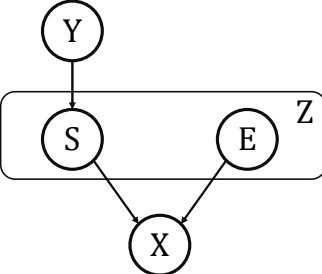

Figure 11: A toy generative model. $X$ is the generated data. $Y$ denotes the label of $X$. The latent $Z$ is partitioned into two parts $S$ and $E$, where $S$ depends on $Y$ while $E$ is independent of the label. Given $Y$, the generative model will have to encode all the variance of $X$ into $E$ to successfully model the full $p(X|Y)$. CLARID identifies those high-variance components given $Y$ and remove them from $Z$ to preserve core class information $S$, creating an information bottleneck without retraining the CDM.

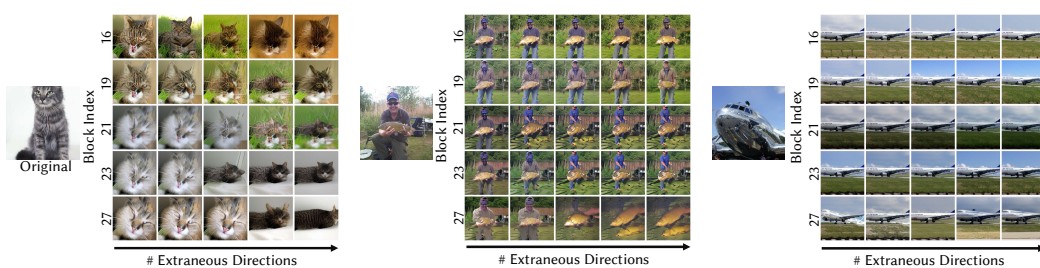

Figure 12: Canonical Samples when computing extraneous directions at different layers in a DiT (Peebles & Xie, 2023), with $t_e = 0.8T$. Note that we do not use CFG after the extraneous directions projection, to present a more straightforward comparison between different layers. We choose $l = 27$, *i.e.* the last layer, to ensure an adequate change in the input.

construction and by our editing experiments in Figure 10, moving along those directions changes the image appearance while keeping the class identity unchanged, so they are a proxy for directions that contribute significantly to $I(Z; Y)$ but not to $I(Z; X|Y)$. Denote the new latent obtained by CLARID as $\tilde{Z}$. CLARID reduces the total information that $Z$ contains about $X$, as it removes certain things from $Z$, hence $I(\tilde{Z}; X) < I(Z; X)$. Meanwhile, it aims to keep the class-relevant term $I(\tilde{Z}; Y) \approx I(Z; Y)$ as unchanged as possible. From this perspective, CLARID implicitly creates an internal information bottleneck inside a pretrained CDM. Therefore, the fraction of latent information that is label-relevant is increased by CLARID. As shown in Figure 3 and 30 in our submitted version, after removing the extraneous directions, the normalized mutual information between the feature cluster assignments and the ground truth labels increases, indicating that the remaining latent is more predictive of the semantic label $Y$.

We develop a toy generative model as shown in Figure 11. Consider that the full distribution $p(X|Y)$ can be decomposed into $p(Y)p(Z|Y)p(X|Z)$. The latent $Z$ can be further decomposed as $Z = S + E$, where $S$ depends on the label $Y$ and $E$ is independent of it. $S$ is known when $Y$ is given. Therefore, to model the full $p(X|Y)$, the generative model must encode all variance of $X$ in $E$. CLARID identifies those high-variance components given $Y$ and removes them from $Z$ to preserve core class information $S$. After removing $E$, $I(Z; X)$ decreases to $I(S; X)$ while $I(Z; Y)$ does not change and equals $I(S; Y)$, hence the information bottleneck effect.

This perspective is related to the information-theoretic goals of $\beta$-TCVAE (Chen et al., 2018) and DBAE (Kim et al.), but differs in where and how the information split happens. In $\beta$-TCVAE, the focus is on the total correlation (TC) term in the ELBO decomposition. Increasing the weight on TC explicitly penalizes entangled latent dimensions while preserving information that is predictive of the ground truth factors. Conceptually, the extraneous directions that CLARID identifies play a similar role. They correspond to high-variance directions that lead to many visually different generations within the same class, and thus tend to contribute more to $I(Z; X|Y)$ than to $I(Z; Y)$. In this sense, the effect of CLARID (from our information bottleneck explanation) parallels that of the TC penalty. Crucially, however, CLARID implements this bottleneck in a pretrained CDM, whereas $\beta$-TCVAE and DBAE achieve the information constraints by modifying the training objective, adding new modules, and retraining the entire model.

# E    THE LAYER INDEX FOR JACOBIAN CALCULATION

In Section 3.2, we treat the CDM as a feature extractor for calculating the Jacobian. We visualize the effect of selecting different $l$, *i.e.* the layer index for computing the Jacobian, in Figure 12 with $t_e$ fixed to $0.8T$, and Figure 13 with $k$ chosen by the method described in Section 3.2.1. While in some cases different layers can yield similar effects, only the top one, *i.e.* $l = 27$, can ensure an adequate change in the output image, or the background can remain unchanged in certain cases.

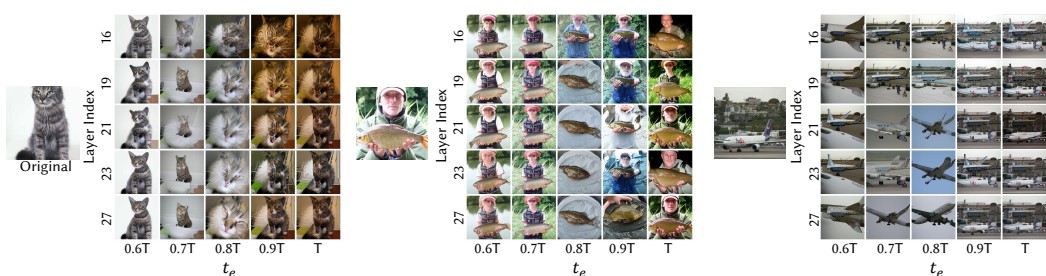

Figure 13: Canonical Samples when computing extraneous directions at different layers in a DiT (Peebles & Xie, 2023). For each image, we choose the number of extraneous directions to be removed automatically according to the method in Section 3.2.1. Note that we do not use CFG after the extraneous directions projection, to present a more straightforward comparison between different layers. We choose $l = 27$, *i.e.* the last layer, to ensure an adequate change in the input.

For Stable Diffusion (Rombach et al., 2022), we follow the practice in Park et al. (2023a) to select the layer index. Concretely, we extract the features after the middle block to ensure that the extraneous directions are semantically meaningful (Jeong et al., 2024; Kwon et al., 2023). We adopt the same strategy for all UNet-based CDMs, including the one used in our CIFAR10 experiments in Section 4.1 and the CDM in the EDM framework (Karras et al., 2022; 2024b).

# F THE GENERALIZATION OF CLARID

## F.1 FINE-GRAINED CONTROL OF CANOREPS WITH TEXT CONDITIONING

CLARID naturally extends to text-conditioning CDMs. Text-conditioning offers a more flexible control over where CanoReps lie than one-hot label conditioning. We show visual results in Figure 14 and 15. The used CDM, a Stable Diffusion 2.1 (Rombach et al., 2022), successfully adapts to different text prompts on the same image, which is in line with previous findings (Park et al., 2023a). In Figure 15, the CDM finds a CanoRep that does not exist in the real world, but all the components in it are real, *e.g.* the water and the airplane. The results demonstrate that it is possible to perform a more fine-grained control over where CanoReps lie. We believe investigating the relationship between the CanoReps and the text conditioning on the same image can reveal the image understanding capability of CDMs, which we leave as a promising future direction.

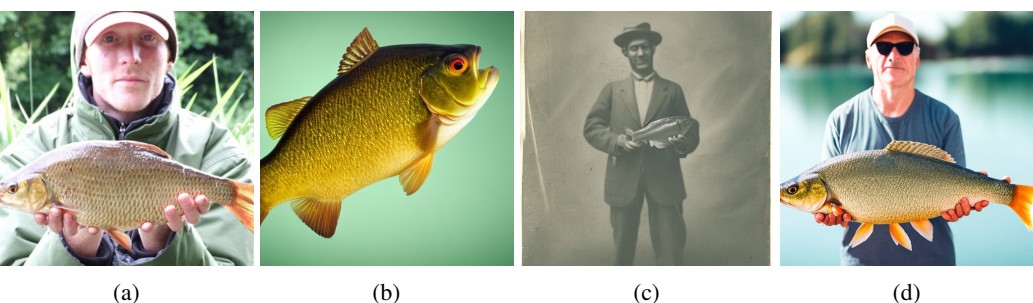

(a)          (b)          (c)          (d)

Figure 14: (a): The original image from the class *Tench*. (b): The Canonical Sample obtained with prompt: *a photo of tench*. (c): The Canonical Sample obtained with prompt: *a photo of a man holding a fish*. (d): An image generated with CFG using the same prompt as in (c), using the same starting noise as in (b) and (c).

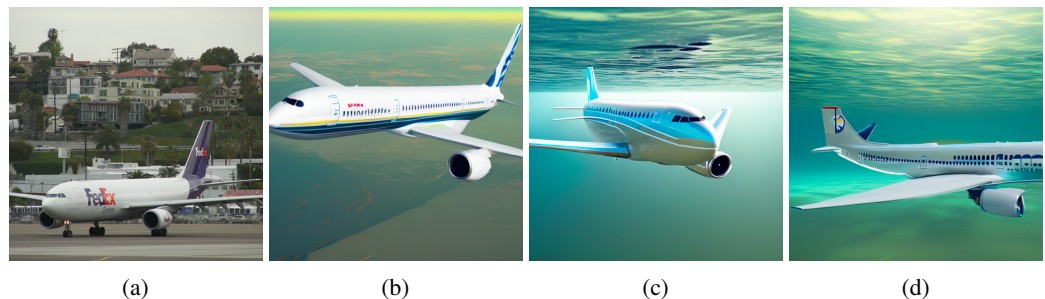

|  | (a) | (b) | (c) | (d) |

Figure 15: (a): The original image from the class *Airliner*. (b): The Canonical Sample obtained with prompt: *a photo of airliner*. (c): The Canonical Sample obtained with prompt: *an airplane flying under water*. (d): An image generated with CFG using the same prompt as in (c), using the same starting noise as in (b) and (c).

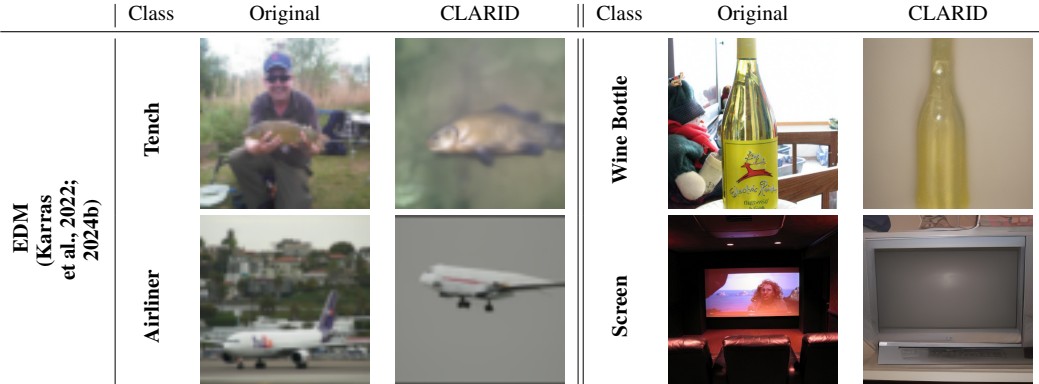

Figure 16: Preliminary results on Canonical Samples generated by the EDM framework (Karras et al., 2022; 2024b). The left column is an EDM trained on ImageNet $64 \times 64$, the right one is on ImageNet $512 \times 512$. We implement the inversion sampler of EDM. It indicates that the main idea behind CLARID is also effective when facing inputs with different resolutions. Note that we do not use CFG or Autoguidance (Karras et al., 2024a) to improve the visual quality, to provide a more straightforward insight into the effectiveness of CLARID on EDM.

## F.2 CLARID IS COMPATIBLE WITH THE EDM SAMPLER AND UViT ARCHITECTURE

The main idea behind CLARID is to identify the latent vectors that carry non-discriminative information and render the latent code of a $F_{inv}$-inverted sample orthogonal to them. Such a formulation does not require any knowledge about the sampler or the architecture of the model, as long as the DM has enough capability to model the conditional data distribution. Here, we demonstrate preliminary results on the generalization of CLARID. Specifically, we test the main idea behind CLARID on the EDM sampler (Karras et al., 2022; 2024b) with a UNet-based CDM, and a UViT (Bao et al., 2023) model using the same DDIM sampler as in the main paper. All CDMs are trained on ImageNet. We implement the inversion sampler of EDM. Here, we aim at showing the effectiveness of the identification of extraneous directions but not on the feature quality. Hence, we do not perform the same analysis as in Section 3.2.1. We present some visual results to demonstrate the intuitive summary of the categorical semantics, as shown in Figure 16 and 17. We believe investigating the relationship between the performance of the DM in generative tasks and it as teacher in *CaDistill* is promising, as previous works have found (Wang et al., 2023; Xiang et al., 2023), and leave it as a future work.

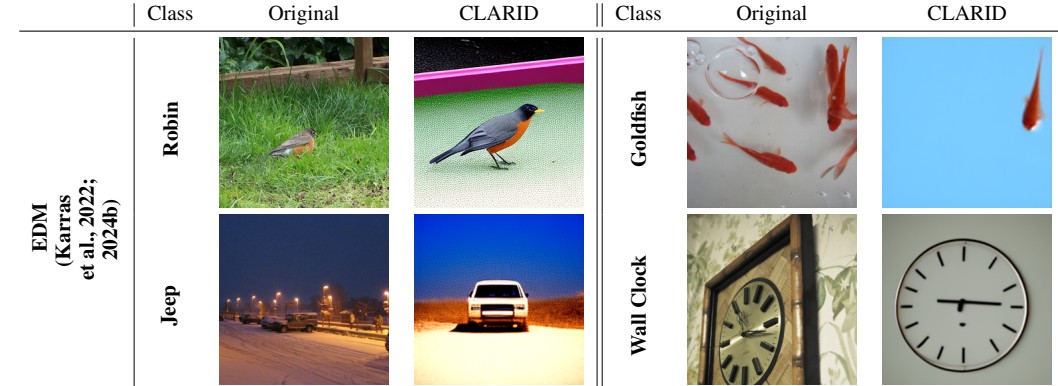

Figure 17: Preliminary results on Canonical Samples generated by a ImageNet-256 UViT (Bao et al., 2023) using DDIM (Song et al., 2021a) sampler. It indicates that the main idea behind CLARID is effective on different architectures of CDMs. Note that we do not use CFG to improve the visual quality, to provide a more straightforward insight into the effectiveness of CLARID on UViT.

## G  DETAILS OF THE IMAGENET20 EXPERIMENT

To develop the CLARID framework, we perform quantitative experiments on a 20-class subset of ImageNet (Deng et al., 2009). This choice balances between computational costs and statistical significance. Our goal is to construct a subset with a clear structure while keeping the class-separation task neither too easy nor too hard. On the one hand, the classes should be dissimilar enough so that separation is meaningful; on the other hand, if they are too dissimilar, separation becomes trivial, and it is hard to draw reliable conclusions. To balance this, we start from the widely used 16-way ImageNet split (Geirhos et al., 2018b;a; Subramanian et al., 2024; Gavrikov & Keuper, 2024), and randomly select one class from each of the 16 superclasses. We then increase the difficulty by randomly selecting 4 additional classes from the bird and dog superclasses, which contain the most subclasses. The resulting subset has a similar number of classes from the two main partitions of ImageNet, animals and artifacts, ensuring a well spread over ImageNet1K.

To demonstrate the robustness and generalization of our method, we conduct three runs by changing the selection of the 20 classes and report standard deviation in our main paper Figure 3. To ensure reproducibility, we list all the classes in the three runs:

- Run1. Indices of the 20 classes: [15, 95, 146, 151, 211, 242, 281, 294, 385, 404, 407, 409, 440, 444, 499, 544, 579, 717, 765, 814]. The corresponding class names: ['robin', 'jacamar', 'albatross', 'Chihuahua', 'vizsla', 'boxer', 'tabby', 'brown bear', 'Indian elephant', 'airliner', 'ambulance', 'analog clock', 'beer bottle', 'bicycle-built-for-two', 'cleaver', 'Dutch oven', 'grand piano', 'pickup', 'rocking chair', 'speedboat'].

- Run2. [7, 94, 97, 143, 152, 266, 281, 294, 385, 405, 409, 436, 440, 499, 554, 555, 559, 671, 687, 766]. The corresponding class names: ['cock', 'fireboat', 'drake', 'oystercatcher', 'toy poodle', 'Irish setter', 'folding chair', 'brown bear', 'Indian elephant', 'airship', 'Japanese spaniel', 'tabby', 'beach wagon', 'cleaver', 'beer bottle', 'fire engine', 'analog clock', 'mountain bike', 'organ', 'rotisserie'].

- Run3. [15, 96, 146, 152, 212, 268, 282, 295, 386, 405, 436, 530, 623, 671, 687, 717, 720, 765, 766, 814]. The corresponding class names: ['robin', 'toucan', 'albatross', 'Japanese spaniel', 'English setter', 'Mexican hairless', 'tiger cat', 'American black bear', 'African elephant', 'airship', 'beach wagon', 'digital clock', 'letter opener', 'mountain bike', 'organ', 'pickup', 'pill bottle', 'rocking chair', 'rotisserie', 'speedboat'].

We plot the pair-wise Wu-Palmer (WUP) distances (Wu & Palmer, 1994) between the selected classes in Figure 18. The selected classes can be similar or dissimilar to each other, demonstrating certain structures. This is appropriate for analyzing the class separability in our case. We choose 50 images from the ImageNet20, building a 1000-sample dataset for the following analysis.

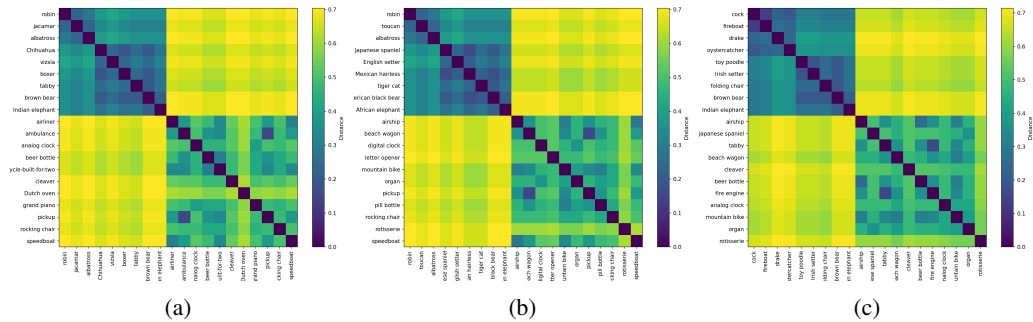

(a)                                   (b)                                   (c)

Figure 18: Wu-Palmer (WUP) distances (1.0-WUP similarity (Wu & Palmer, 1994)) between the classes in ImageNet20 across three runs. (a): Run1. (b): Run2. (c): Run3. The class relationships are structured, hence appropriate for analyzing the class separability in our case. Note that we select the classes from the 16-way ImageNet (Geirhos et al., 2018b;a; Subramanian et al., 2024; Gavrikov & Keuper, 2024) to ensure a fair task difficulty, which can lead to a few class overlaps between runs.

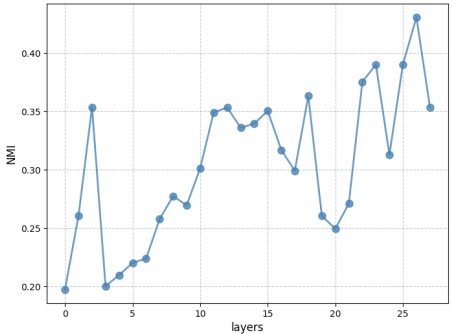
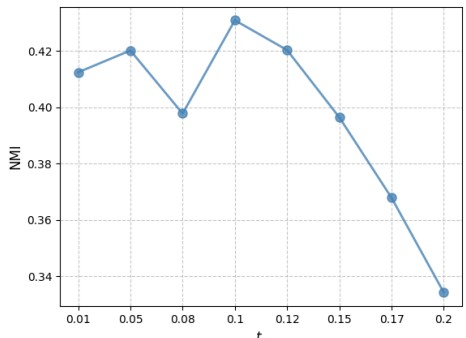

Figure 19: NMI v.s. layers on ImageNet20 in a DiT (Peebles & Xie, 2023), fixing $t_r = 0.1T$. We choose the penultimate layer, *i.e.* the 27th layer (the figure has a 0-start index), in all our experiments.

Figure 20: NMI v.s. feature extraction time step ($t_r$) on ImageNet20 using a DiT (Peebles & Xie, 2023), fixing the layer index to be 27. We choose $t_r = 0.1T$ in all our experiments.

### G.1 SELECTING THE OPTIMAL LAYER AND TIME STEP FOR FEATURE EXTRACTION

We choose an ImageNet $256 \times 256$-pretrained DiT-XL (Peebles & Xie, 2023) model for the quantitative analysis. We consider the outputs of all 28 ViT blocks in it. For the time step, we select $t_r = \{0.01, 0.05, 0.08, 0.1, 0.12, 0.15, 0.17, 0.2\}$. We perform K-means clustering on the feature map after average pooling, and compute the normalized mutual information between the cluster assignments and the ground truth class labels. The average pooling reduces noise in the feature map, making the cluster more accurate and compact. The results for different layers and time steps are shown in Figure 19, 20. Our conclusion on the time step for feature extraction is consistent with previous studies (Mukhopadhyay et al., 2023; Yang & Wang, 2023) that adopt linear probing on the features for quantifying the feature quality, albeit with different DM architectures. Such a result provides **evidence on the validity of our metric, *i.e.* normalized mutual information.**

### G.2 COMPARISON BETWEEN CLARID COMBINED WITH CFG AND PURE CFG

In Section 3.4 in the main paper, we show that Canonical Samples, obtained via CLARID combined with CFG, contain fewer class-irrelevant components than samples obtained via CFG. We now show that Canonical Samples are more separable in the CDM feature space than using pure CFG to quantitatively demonstrate that they encode different information. We compare the cluster quality between the features corresponding to the two kinds of samples, namely Canonical Features and

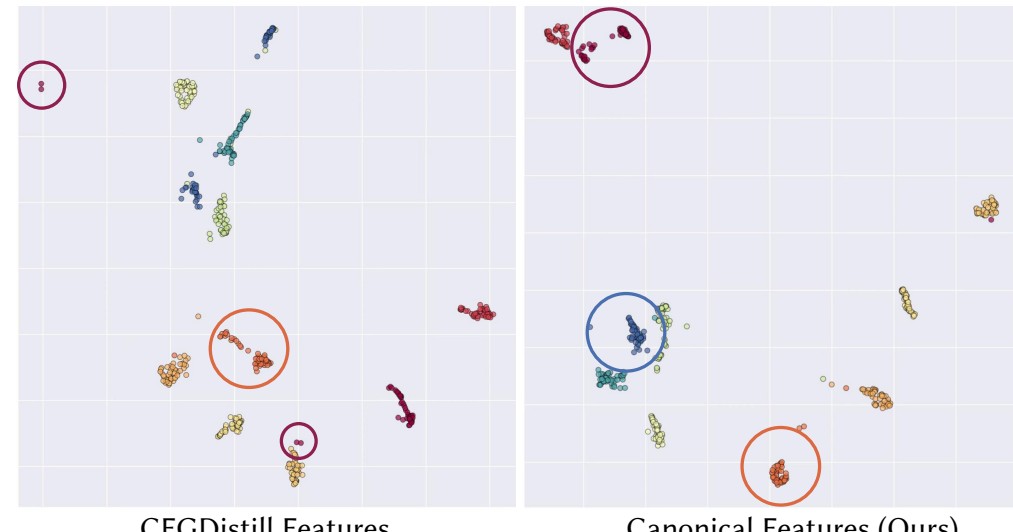

CFGDistill Features      Canonical Features (Ours)

Figure 21: A 2D UMAP (McInnes et al., 2018) projection of the CDM feature space, showing clusters for ten classes. In this case, Canonical Features are produced by CLARID combined with CFG. Colors indicate classes. Canonical Features form tighter, more uniformly shaped clusters, whereas CFGDistill features often result in irregular cluster shapes (orange), more outliers (red), and disconnected regions even for samples sharing the same class label (blue).

CFGDistill features, using the pipeline in Section 3.2.1. Canonical Features achieve an NMI score of 0.7762 while CFGDistill features achieve 0.7308. The UMAP projections of the two types of features are shown in Figure 21. Canonical Features form better separated clusters, implying that they capture different information from using pure CFG.

### G.3 ON USING NMI FOR MEASURING FEATURE QUALITY AND THE EFFECTIVENESS OF *CaDistill*

While NMI is valid for self-evaluation of the feature quality within the CLARID framework, it is not valid for a direct comparison between the quality of features obtained via different methods, such as CFG. For example, when using CFG on the same 1000 samples after $F_{inv}$ and extracting features at the same $t_r$ as in CLARID, it can yield a higher NMI (0.6108 in CLARID v.s. 0.7808 with CFG magnitude being 4.0). Adopting CFG after performing extraneous direction projection can also improve the metric (0.7762 in CLARID with CFG being 3.0). However, NMI only examines the compactness and separability of each cluster, ignoring the low-dimensional manifold structure of the data. For example, a line-shaped manifold and a circle-shaped manifold can yield the same NMI, while they capture different characteristics of the data. In an extreme case where the features are constant for all samples belonging to the same class, the NMI will be 1.0, while the features are not meaningful in this case. **In other words, we do not want the student to just learn a "converging" prior on the features belonging to the same classes, but to learn actual class semantics**. This claim is validated in our extended ablation studies in Section M.7, where the student does not perform well when the CFG magnitude is high. These results undermine the notion that the student only mimics the CDM's label-conditioning embeddings: those embeddings behave as a trivial collapsing prior rather than conveying the encoded semantic structure. A promising future direction is to develop new feature quality metrics or adapt existing ones into CLARID that consider the structure of the data. This metric is also invalid on datasets with simple data structures, such as CIFAR10 (Krizhevsky et al., 2009). The CDM features of the original samples are already perfectly separable and achieve an NMI of 1.0. However, as shown in Table 1 in the main paper, CLARID can still find more semantically meaningful samples than the original ones, and convey the knowledge to the student via *CaDistill*. While the simple structure of the data prevents the utilization of NMI, it is significantly easier to process than real image datasets. Either calculating the extraneous directions or

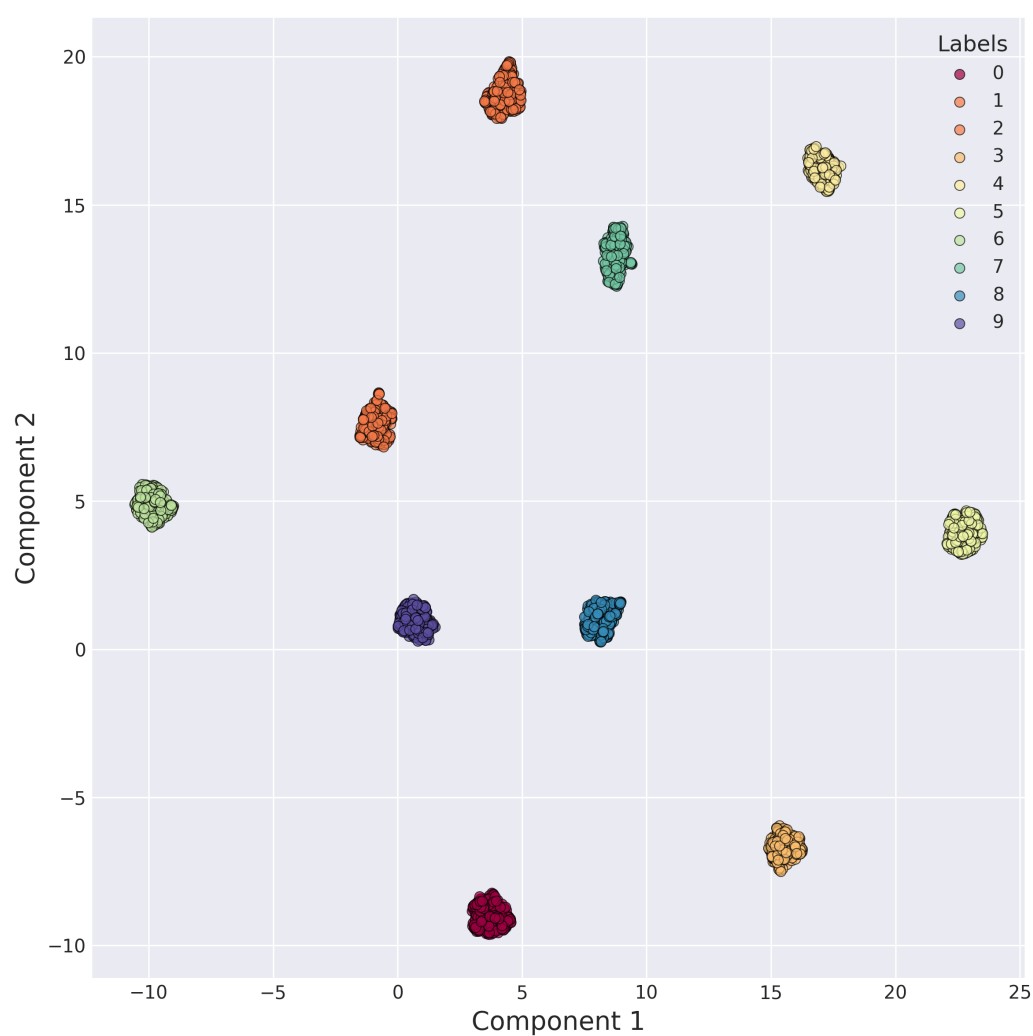

Figure 22: The CDM features of the original samples in CIFAR10 (Krizhevsky et al., 2009) are already separable and achieve an NMI of 1.0. However, CLARID is still effective in this case, as shown in Table 1.

training with **CaDistill** requires far less computational resources than large, real-image datasets such as ImageNet (Deng et al., 2009). Therefore, we simply brute-force search the best hyperparameters in CLARID on CIFAR10. The hyperparameters on CIFAR10 are: $t_e = 0.8, n = 10, t_r = 0.13T$, layer $= up.0$. We also only use a 10% subset for obtaining CanoReps, as shown in Table 1.

# H    CHOOSING HYPERPARAMETERS $t_e$ AND $n$ FOR CLARID

## H.1    FINDING THE OPTIMAL $t_e$

In Section 3.2, we decide $t_e$ by finding the saturation point of classification accuracy on samples generated by our two-stage strategy. We use 3 classifiers and compute the average accuracy of them:

1. A ViT-Large pre-trained on ImageNet12K (Deng et al., 2009). The input size is 224.

2. An ImageNet22k-pre-trained Swin V2 (Liu et al., 2022). The input size is 256.

3. An ImageNet22k-pre-trained ConvNeXt V2 (Woo et al., 2023). The input size is 384.

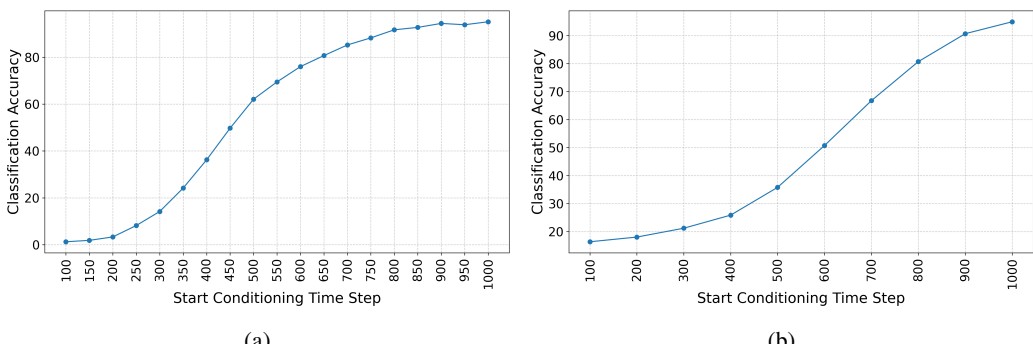

(a)                                                                (b)

Figure 23: The classification accuracy on samples generated by the two-stage strategy in Section 3.2, using **(a)** DiT (Peebles & Xie, 2023) and **(b)** Stable Diffusion (Rombach et al., 2022). The accuracy is averaged over 3 classifiers.

All model weights are downloaded from the PyTorch Image Model library (timm) (Wightman, 2019). The accuracy curve for DiT is shown in Figure 23. The maximum time step $T$ is 1000 for both DiT (Peebles & Xie, 2023) and Stable Diffusion (Rombach et al., 2022). We use DDIM as the sampler and set the total diffusion time step to 50. For each class, we generate 5 images. Hence $m = 5$ in Section 3.2. The prompt template for Stable Diffusion is: *a photo of <class name>*, where the class name is the WordNet name of each ImageNet class (Deng et al., 2009). We identify $t_e = 800$ $(0.8T)$ for DiT and $t_e = 1000$ $(T)$ for Stable Diffusion. We provide visual comparisons between the images generated by selecting different $t_e$ in Figure 24. Qualitatively, a small $t_e$ might lead to insufficient changes in the input image, while a large $t_e$ in DiT can fail to make the model aware of the class conditioning, resulting in less meaningful extraneous directions.

### H.2 Choosing the total number of extraneous directions $n$ for adaptively selecting $k$

Our key observation on selecting $k$, *i.e.* the number of extraneous directions to be projected, is that the effect caused by projecting extraneous directions is not smooth with varying $k$. That is, projecting one more extraneous directions can cause significant changes in the input. It is because extraneous directions are orthogonal to each other by design, hence there is no guarantee that their semantics are correlated. Importantly, projecting more extraneous directions does not necessarily mean a more separable set of Canonical Features. It can lead to a loss of the class-defining cues, as shown in the fish image in Figure 25b. We show the visual effects of selecting different $k$ in Figure 24, 25. Our method CLARID selects $k$ by adaptively choosing the elbow point on the explained variance ratio (EVR) sequence with total number of extraneous directions being $n$. The algorithm for finding the elbow point is presented in Algorithm 1.

---

**Algorithm 1** Find Elbow via Knee Method(Sequence $S[0 \ldots n-1]$)

---

1: $n \leftarrow |S|$
2: $P_{\text{start}} \leftarrow (0, S[0])$
3: $P_{\text{end}} \leftarrow (n-1, S[n-1])$
4: $v \leftarrow P_{\text{end}} - P_{\text{start}}$
5: $u \leftarrow v/\|v\|$
6: **for** $i = 0$ **to** $n-1$ **do**
7:    $w \leftarrow (i, S[i]) - P_{\text{start}}$
8:    $\text{projLen} \leftarrow w \cdot u$
9:    $\text{projVec} \leftarrow \text{projLen} \times u$
10:   $\text{perpVec} \leftarrow w - \text{projVec}$
11:   $d[i] \leftarrow \|\text{perpVec}\|$
12: **end for**
13: $k \leftarrow \arg\max_{0 \le i < n} d[i]$
14: **return** $k$

---

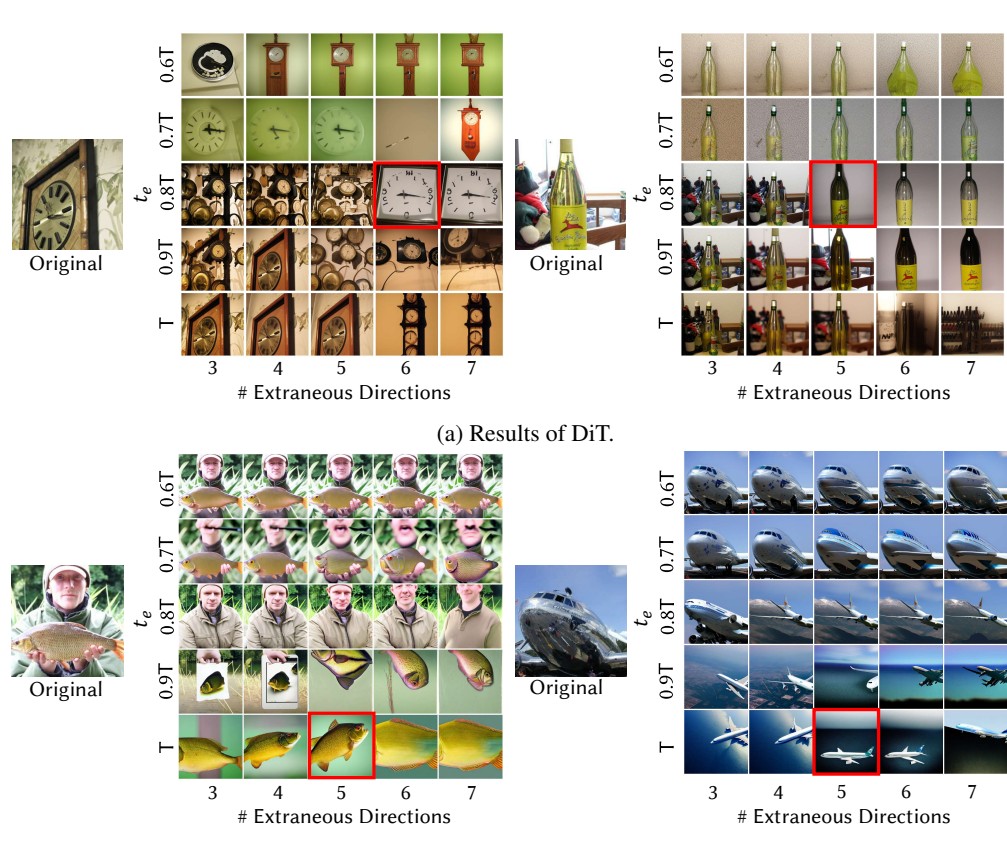

(a) Results of DiT.

(b) Results of Stable Diffusion.

Figure 24: Visual comparisons between the images generated by selecting different $t_e$ and $k$ in CLARID. Red boxes indicate the one automatically selected by our method.

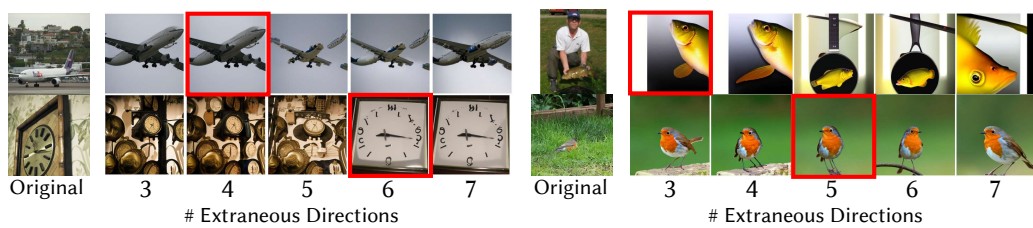

(a) Results of DiT without CFG for a more straightforward comparison.

(b) Results of Stable Diffusion with CFG magnitude being 7.5.

Figure 25: Visual comparisons between the images generated by selecting different $k$ in CLARID. Red boxes indicate the one automatically selected by our method.

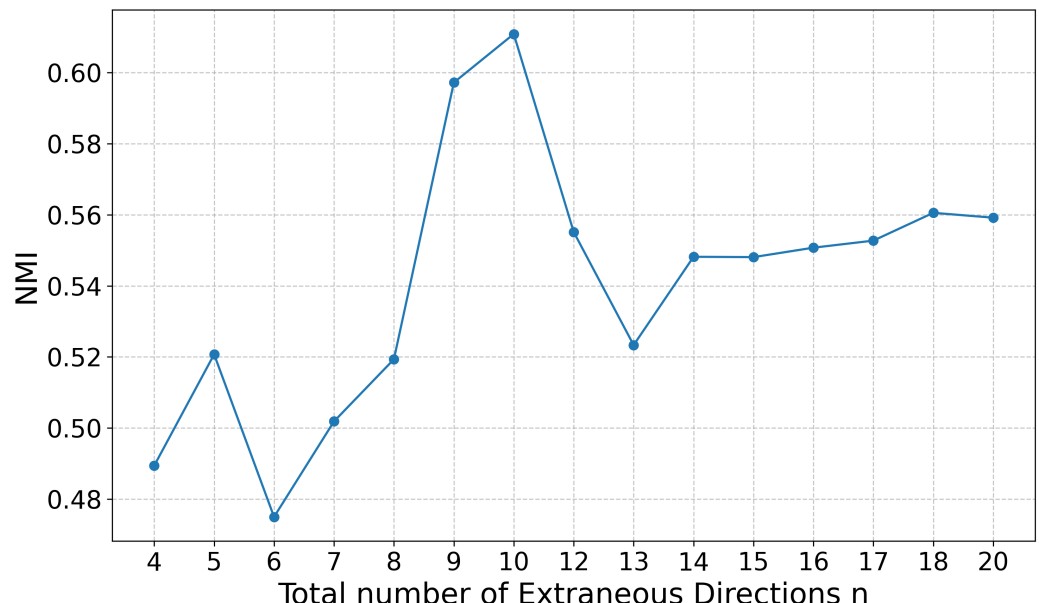

Figure 26: NMI on ImageNet20 v.s. total number of extraneous directions $n$ on DiT. A large $n$ can diminish the discriminative power of the input images, whereas a small one cannot change the inputs too much. Neither case is desired. Hence, we choose $n = 10$ for DiT in our experiment. Note that this is a self-evaluation within the CLARID framework, hence NMI is valid in this case.

We show a quantitative comparison between different $n$ in Figure 26 on ImageNet20. Too large $n$ tends to select a larger $k$, leading to eliminating too many components in the input image and diminishing discriminative power. A small $n$ will lead to a small $k$ and cannot change the inputs too much. We select $n = 10$ for DiT, and $n = 10$ for SD in our experiment. Note that this $n$ is tailored to specific CDMs, and the same $n$ for DiT and SD is merely a coincidence. We plot the histogram of $k$ when fixing $n = 10$ on 78000 images from ImageNet100, as we used in our experiments in Section M.7, in Figure 27. The selection process does not converge to a single $k$, supporting the effectiveness of our method. CLARID has certain fault tolerance capacity, *i.e.* slightly changing the number of projected extraneous directions can still result in desired images. For example, in Figure 25, selecting $k = 3$ or $k = 4$ for the airplane image on DiT, or selecting $k = 3 \sim 7$ for the bird image on Stable Diffusion, can all result in desired outputs.

### H.2.1 STABLE DIFFUSION 2.1

We provide additional results on Stable Diffusion 2.1 model (Rombach et al., 2022). We first choose $t_r = 0.13T$ and the layer to be up_blocks.1, according to the results in Figure 28, 29. We then perform the same analysis as done in Section 3.2.1 in the main paper and Section H. Figure 30 shows the results. Observe that $t_e = T$, which is $t_e = 0.999$ in the figure, yields the best results when performing fixed-$k$ projection. This validates our saturation-point-based $t_e$ selection, as described in Section 3.2.1 in the main paper. Again, our CLARID produces the highest NMI among all other methods. In Figure 31, we show the different NMI results achieved by selecting different $n$ for deciding $k$ adaptively, and choose $n = 10$.

### H.3 FAILURE CASES AND DISCUSSION

We identify some promising future directions and discuss the failure cases. First, we fix $t_e$ for all samples, which can be suboptimal on certain images. We only perform CLARID in a single time step instead of selecting multiple $t_e$. A series of projecting away extraneous directions has the potential of discarding more class-irrelevant information. Regarding the number of extraneous directions, we only perform experiments on cumulative projection, *i.e.* projecting away the top-$k$ extraneous

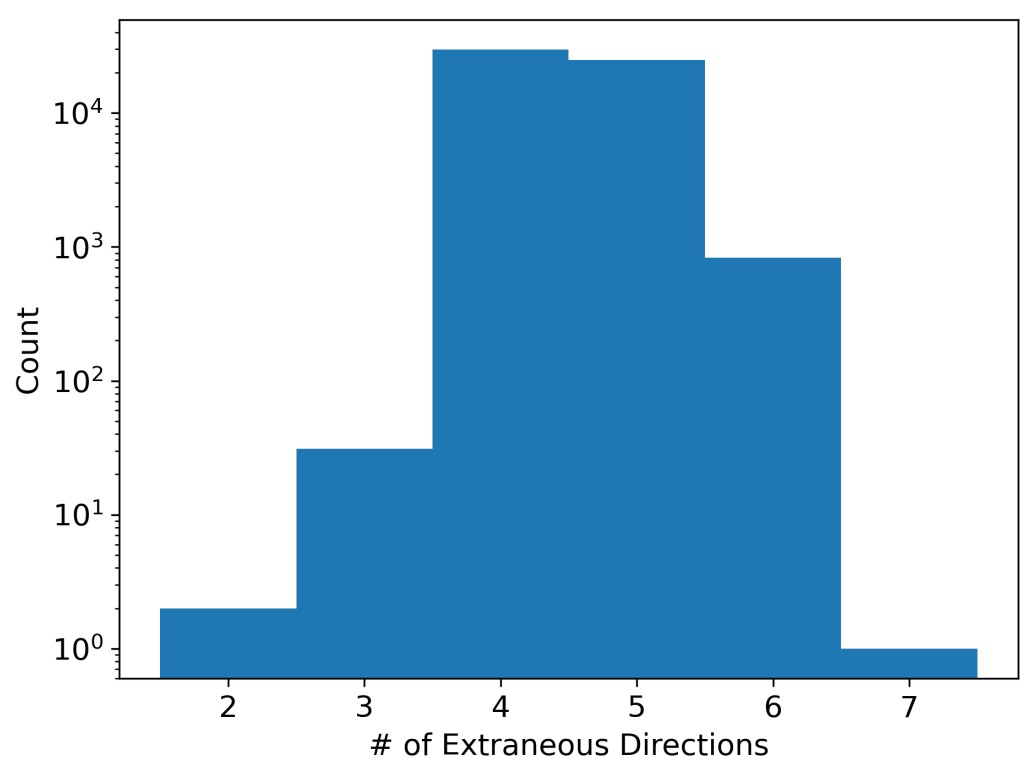

Figure 27: Histogram of $k$ when fixing $n = 10$ on 78000 images from ImageNet100. We use this dataset in our experiments in Section M.7.

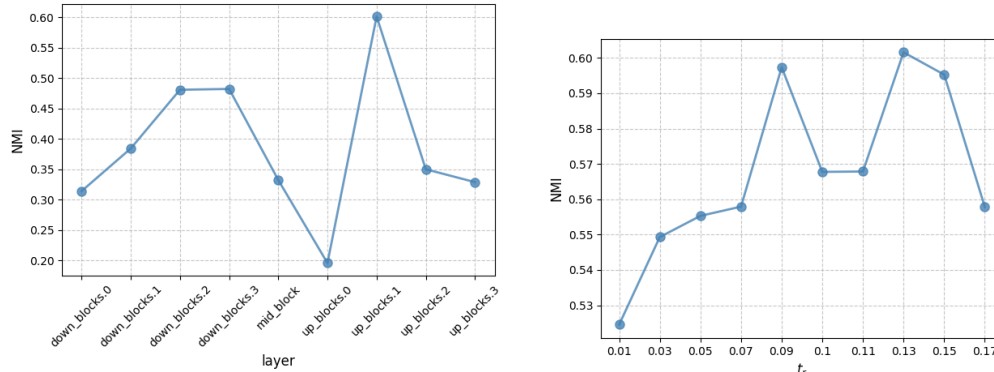

Figure 28: NMI v.s. layers on ImageNet20 in a Stable Diffusion 2.1 (Rombach et al., 2022), fixing $t_r = 0.13T$. We choose the up_blocks.1 layer in all our experiments.

Figure 29: NMI v.s. feature extraction time step ($t_r$) on ImageNet20 using a Stable Diffusion 2.1 (Rombach et al., 2022), fixing the layer index to be up_blocks.1.

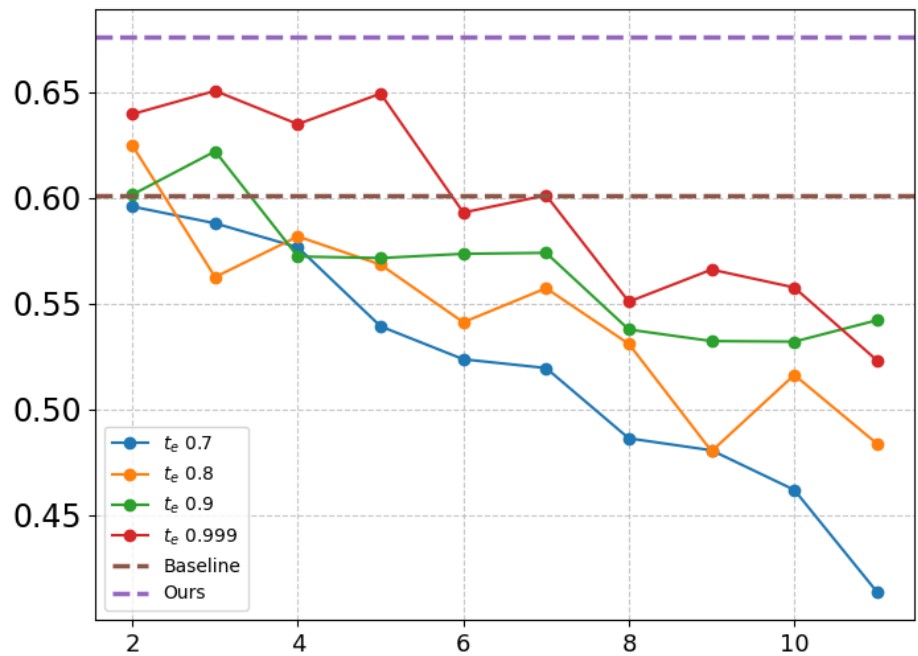

Figure 30: The normalized mutual information (NMI, higher is better) between cluster assignments of Stable Diffusion (SD) features using a Stable Diffusion 2.1 (Rombach et al., 2022) and the ground truth labels. CLARID achieves the highest NMI. Baseline is the original SD features.

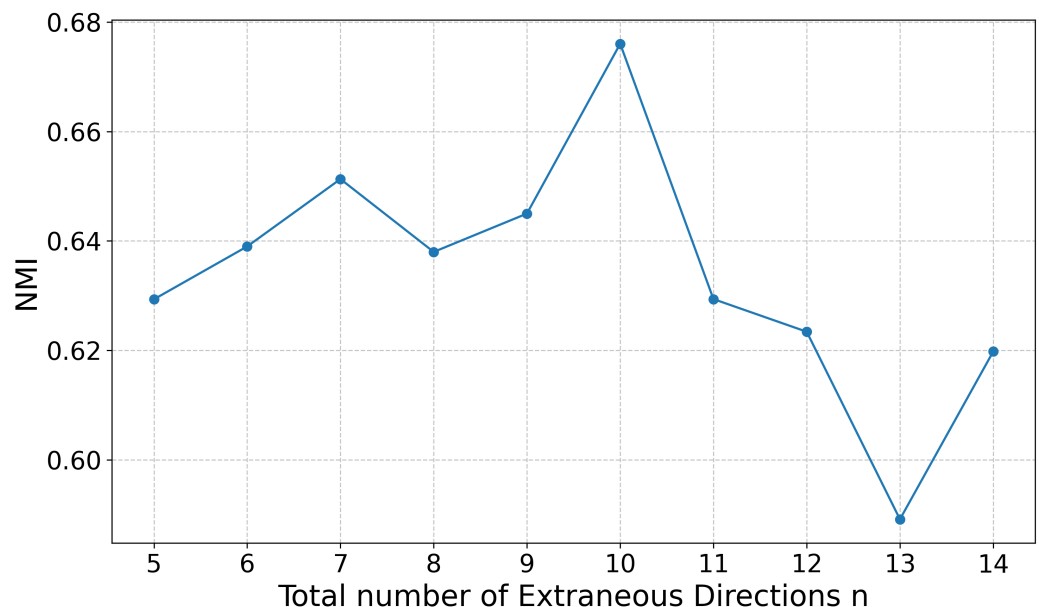

Figure 31: NMI on ImageNet20 v.s. total number of extraneous directions $n$ on a Stable Diffusion 2.1 (Rombach et al., 2022). A large $n$ can diminish the discriminative power of the input images, whereas a small one cannot change the inputs too much. Neither case is desired. Hence, we choose $n = 10$ for Stable Diffusion in our experiment. Note that this is a self-evaluation within the CLARID framework, hence NMI is valid in this case.

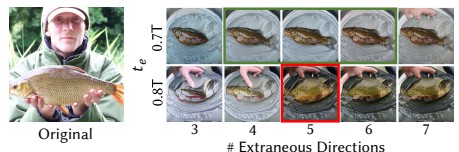

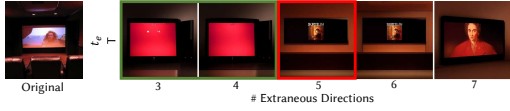

Figure 32: A Failure case of CLARID on selecting $t_e$. Green boxes are the optimal choice, qualitatively. Red boxes are the ones CLARID selects.

Figure 33: A Failure case of CLARID on selecting $k$ when $n = 10$. Green boxes are the optimal choice, qualitatively. Red boxes are the ones CLARID selects.

directions. A careful selection of the extraneous directions can contribute to better results. Due to the limits in computational resources, we leave them as future work. Occasionally, CLARID can select suboptimal $k$, leading to artefacts in the generated images. We show failure cases for $t_e$ and $k$ in Figure 32, 33. The aforementioned future directions can potentially serve as solutions to those failure modes.

# I    COMPUTATIONAL COSTS AND SCALABILITY OF CLARID

We approximate only the top singular vectors of the Jacobian via the established Jacobian subspace iteration method (Haas et al., 2024; Park et al., 2023a), avoiding a full SVD. Complexity scales quadratically in the number of vectors. We use $n = 10$ per image. Measured on a single Nvidia A100, the runtime is 9.5s for DiT and 22s for Stable Diffusion. A simple speed-up is early truncation: reducing max iters from 100 to 20 and relaxing the threshold from 1e-4 to 1e-3 yields 1.5 s (DiT) and 4.4 s (SD) with visually similar outputs. This is valid because we actually use only the top singular vectors, while the remaining ones serve merely to locate the elbow. Developing faster Jacobian singular-vector estimators is a promising future work. On scalability, CLARID is an offline preprocessing step. A single trial suffices for a dataset. In classification (Section M.7), processing only 10% of data attains competitive downstream performance. CLARID works across different

Table 3: Quantitative comparison between Training samples, the samples used in CFGDistill (CFGDistill samples), and Canonical Samples. All metrics are computed following the pipeline in (Dhariwal & Nichol, 2021).

|  | FID | sFID | Inception Score | Precision | Recall | Classification Accuracy |
|---|---|---|---|---|---|---|
| Training samples | 0.8 | 3.4 | 67.7 | 0.76 | 0.71 | 85.1±0.2 |
| CFGDistill samples | 14.2 | 17.6 | 74.1 | 0.91 | 0.28 | 96.3±0.3 |
| Canonical Samples | 12.5 | 11.7 | 74.1 | 0.93 | 0.30 | 96.3±0.2 |

diffusion models, architectures, and samplers (Section F.2). Advances in either further improve scalability. Our goal is to demonstrate that core class semantics can be extracted from conditional diffusion models. We view improving efficiency as an important future work.

## J   QUANTITATIVE ANALYSIS OF CANONICAL SAMPLES

We provide quantitative metrics for a better understanding of the visual quality and representativeness of Canonical Samples generated by DiT (Peebles & Xie, 2023). We select 50000 images from the training set (Training samples), the samples used in CFGDistill (CFGDistill samples), and Canonical Samples, respectively. We compute the FID, sFID, Inception Score, Precision, and Recall, following the pipeline in (Dhariwal & Nichol, 2021). We report an additional metric called Classification Accuracy to show the discriminativeness of the samples. We use a pre-trained ResNet50, and randomly pick 10000 images from all sets of samples (Training, CFG, Canonical). We average the results over 3 seeds and report the mean and standard deviation as the error bars. The results are given in Table 3.

The reported values are different from random generative sampling in (Peebles & Xie, 2023), because we use DDIM inversion and decoding with a small number of time steps, instead of sampling from noise using a large number of function evaluations (NFEs). Note that our work does not focus on the visual quality of Canonical Samples, but instead on their semantics. Yet we observe that Canonical Samples have lower FID and sFID, indicating a better visual quality. The high precision and low recall suggest that samples that lie inside the data distribution while forming tight clusters, which is desired for class prototypes. We observe that the classifier gives higher classification accuracy on Canonical Samples, which means that they are more discriminative than the original samples.

We note that the pure CFG can also give similar results, particularly in Inception Score and Classification Accuracy. However, we emphasize that Canonical Samples and CFG yield different kinds of representative samples. Canonical Samples retain minimal class-irrelevant information, while CFG produces both class-irrelevant and class-related results. The visual results in Section Q support this claim. For example, in the "Tusker" class (Figure 44), pure CFG emphasizes "elephant" while Canonical Samples preserves both elephant and tusks, the joint defining attributes. The quantitative evidence is provided by our *CaDistill* experiments, in which CFGDistill yields models that suffer more from spurious features (Table 2) and generalize worse than the models produced by *CaDistill*.

## K   DETAILS OF THE PROOF-OF-CONCEPT EXPERIMENT

### K.1   DATA GENERATION PROCESS

We adopt a hierarchical generative process described in Eq. 11 to generate 2D data points from two classes.

$$p(\boldsymbol{x}) = p(y)\, p(\boldsymbol{x}_{\text{core}} \mid y)\, p(\boldsymbol{x}_{\text{var}} \mid \boldsymbol{x}_{\text{core}}),$$

$$Y \sim \text{Bernoulli}\left(\tfrac{1}{2}\right), \quad U \sim \mathcal{U}(-0.1, 0.1), \quad \boldsymbol{\varepsilon} = (\varepsilon_x, \varepsilon_y)^{\mathsf{T}} \sim \mathcal{N}\big(\boldsymbol{0}, 0.01\boldsymbol{I}_2\big),$$

$$\boldsymbol{s}(Y) = \begin{cases} (0,0)^{\mathsf{T}}, & Y = 0, \\ (4,0)^{\mathsf{T}}, & Y = 1, \end{cases} \qquad \boldsymbol{x}_{\text{core}} = (U,0)^{\mathsf{T}} + \boldsymbol{s}(Y), \qquad \boldsymbol{x}_{\text{var}} = \boldsymbol{\varepsilon}, \tag{11}$$

$$\boldsymbol{x} = \boldsymbol{x}_{\text{core}} + \boldsymbol{x}_{\text{var}} + \big(3|\varepsilon_y|, 0\big)^{\mathsf{T}}.$$

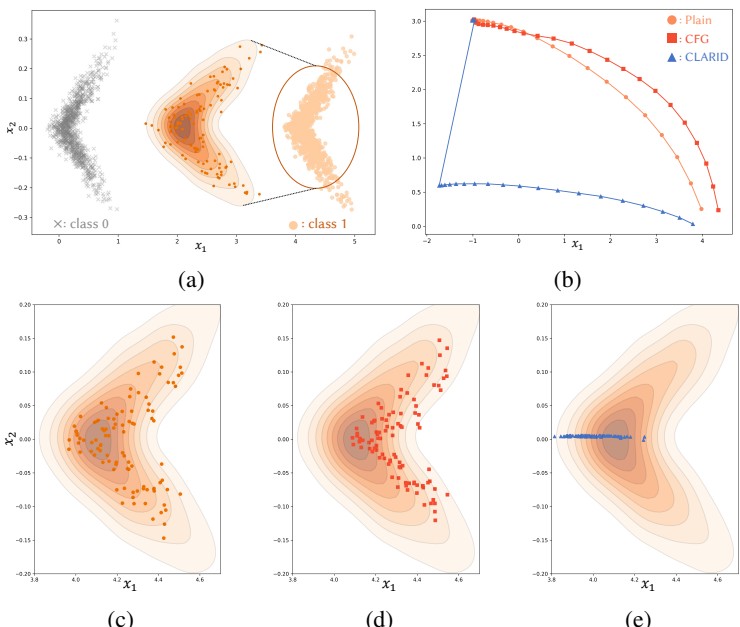

Figure 34: A toy example of CLARID. **(a)**: The samples of class 0 and class 1; **(b)**: Sampling trajectory of plain conditioning (Plain), classifier-free guidance (CFG), and CanoRep (CLARID) starting from $F_{inv}([4.0, 0.2])$. CanoRep (CLARID) orthogonalize the data latent code and class-irrelevant components encoded in the latent code in the CDM, yielding Canonical Samples; **(c,d,e)**: The generated samples of Plain, CFG, and CLARID, respectively. CLARID produces Canonical Samples that lie on a 1D manifold inside class 1, offering an intuitive visual summary of the core class semantics.

Table 4: The hyperparameters used in our toy experiment in Section 3.3.

| # data points | Epoch | Optimizer | Batch size | Learning rate | Weight decay | Label drop rate |
|---|---|---|---|---|---|---|
| 1000 | 1000 | Adam (Kingma & Ba, 2015) | 128 | 1e-3 | 0 | 0.1 |

In our toy model, we have two classes. The process first generates class-specific samples on a segment $L = \{(x_1, 0) | 4y - 0.1 \leq x_1 \leq 4y + 0.1\}$, where $y \in \{0, 1\}$ denotes the class labels. It then adds class-independent noise to the points to generate the observed data. The samples from class 0 are included solely to introduce an inter-class contrast. We shift the $x$-axis value according to the $y$-axis variation to increase the complexity of the distribution.

## K.2 ARCHITECTURE AND TRAINING DETAILS

We design a simple 3-layer multilayer-perceptron-based (MLP-based) diffusion model to model this 2D distribution. We set the hidden dimensionality to 80, and the dimensionality for both label and time step is 16. We use sinusoidal embeddings for uniquely encoding the 1000 time steps (Dosovitskiy et al., 2020). We train the diffusion model using the standard DDPM loss (Ho et al., 2020). The hyperparameters of training are given in Table 4.

## K.3 CLARID AND BASELINE CONFIGURATION

When performing CLARID, we analytically compute the Jacobian of the network and calculate the corresponding singular vectors. We remove the first right singular vector at time step $t = 0.99T$. As a baseline, we take the latent codes obtained with $F_{inv}$ and perform classifier-free guidance (CFG), steering the generation process toward regions with higher class-1 likelihood.

Table 5: The hyperparameters used in our experiments. We train a ResNet18 (He et al., 2016) on CIFAR10 and a ResNet50 on ImageNet. SGDM is SGD with momentum=0.9. *CaDistill* is effective with different training settings, as shown in Section M.2 and M.8.

| $\mathcal{D}$ | Epoch | Optimizer | Batch size | Learning rate | Weight decay | LR scheduler | LR decay rate |
|---|---|---|---|---|---|---|---|
| CIFAR10 | 200 | SGDM | 128 | 0.1 | 5e-4 | Step 100,150 | 0.1 |
| ImageNet100 | 100 | SGDM | 256 | 0.1 | 1e-4 | Cosine | / |
| ImageNet | 100 | SGDM | 512 | 0.1 | 1e-4 | Cosine | / |

### K.4 RESULTS AND ANALYSIS

The results are shown in Figure 34. Notably, CLARID pushes most samples to a 1D manifold inside class 1, whereas CFG mainly steers the samples away from class 0. This low-dimensional manifold described by Canonical Samples can be regarded as a summarization of class 1 information in this case. The underlying structure revealed by Canonical Samples corresponds to one of the true generative processes for the observed data, which is the one used in our toy model. Reliably recovering the exact generative model is intractable due to the identifiability issue (Locatello et al., 2019). The solution generally requires extra inductive bias in modeling data distribution, which we leave for future work.

## L   TRAINING AND EVALUATION DETAILS OF *CaDistill*

It is often computationally demanding to compute the analytical Jacobian in Eq. 1. Because we only need the singular vectors, we adopt a well-established method, as in (Park et al., 2023a), to calculate only the top singular vectors of the Jacobian.

Regarding CaDistill, we perform all the experiments using the PyTorch platform. We provide the training hyperparameters in Table 5. The temperature parameter $\tau$ in *CaDistill* is fixed to 0.1 in all cases, as in (Khosla et al., 2020). On CIFAR10, we use random crop (`torchvision.transforms.RandomCrop(32,padding=4)`) and random horizontal flip (`torchvision.transforms.RandomHorizontalFlip`) for data augmentation. Note that the SupCon (Khosla et al., 2020) baseline uses a different data augmentation strategy and is trained for much longer epochs (1000). We follow the official code implementation (link) to reproduce this baseline on CIFAR10. On ImageNet and ImageNet100, we adopt the data augmentation used in Khosla et al. (2020), to improve the generalization performance of the trained model so that the baselines have meaningful results on the used generalization benchmarks. For example, the ResNet50 trained with random resized crop and random horizontal flip, as in He et al. (2016), will have 0% accuracy on ImageNet-A (Djolonga et al., 2021), while the baseline ResNet50 (Vanilla) in our experiments achieves 6.3%. The difference in data augmentation results in the performance difference on the ImageNet validation set between our baseline model and the one trained in He et al. (2016). However, we are not focusing on the clean performance in our settings. We use a single Nvidia A100 GPU for the CIFAR10 experiments, and four A100 GPUs for the ImageNet and ImageNet100 experiments. Training one model on CIFAR10 takes around $0.5 \sim 1$ hour. Training on ImageNet takes around 30 hours for ResNet50, and 60 hours for ResNet152.

Our CaDistill involves the usage of CDM features, *i.e.*, Canonical Features. Instead of forwarding the CDM during training, which will lead to a large computational cost, we pre-compute the Canonical Features and load them during training, contributing to an efficient training pipeline (also see Section I). All Canonical Features are 1D vectors, which are the results of performing average pooling on the original feature map, as done in a previous work (Yang & Wang, 2023).

### L.1   ADVERSARIAL ROBUSTNESS BENCHMARKING

We examine the adversarial robustness of different student models using four adversarial attacks, PGD (Madry et al., 2018), CW (Carlini & Wagner, 2017), APGD-DLR (Croce & Hein, 2020), and APGD-CE (Croce & Hein, 2020). The detailed settings are given in Table 6, 7,and 8 for CIFAR10, ImageNet, and ImageNet100 (AutoAttack), respectively. We choose PGD (Madry et al., 2018) since it is the most popular method for examining adversarial robustness (Xu et al., 2024). Moreover, we want to examine whether the drawback of cross-entropy loss can lead to false robustness (Croce &

Table 6: Hyperparameters in different adversarial attacks on CIFAR10.

|  | PGD | CW | APGD-DLR | APGD-CE |
|---|---|---|---|---|
| Max magnitude | 2.0/255 | / | 2.0/255 | 2.0/255 |
| Steps | 5 | 5 | 5 | 5 |
| Step size | 0.5 | / | / | / |
| $\kappa$ | / | 0.0 | / | / |
| $c$ | / | 0.2 | / | / |
| $\rho$ | / | / | 0.75 | 0.75 |
| EOT | / | / | 1 | 1 |

Table 7: Hyperparameters in different adversarial attacks on ImageNet.

|  | PGD | CW | APGD-DLR | APGD-CE |
|---|---|---|---|---|
| Max magnitude | 0.33/255 | / | 0.33/255 | 0.33/255 |
| Steps | 5 | 5 | 5 | 5 |
| Step size | 0.5 | / | / | / |
| $\kappa$ | / | 0.0 | / | / |
| $c$ | / | 0.1 | / | / |
| $\rho$ | / | / | 0.75 | 0.75 |
| EOT | / | / | 1 | 1 |

Hein, 2020). The step size in PGD can also largely affect the result. Hence, we choose the Auto-PGD family (Croce & Hein, 2020) to automatically decide the step size and incorporate the new Difference of Logits Ratio (DLR) loss, resulting in APGD-CE and APGD-DLR, respectively. We also want to include an optimization-based adversarial attack and thus select the CW (Carlini & Wagner, 2017) attack. The hyperparameters of different attacks are chosen to ensure a meaningful comparison between different models, avoiding the case in which all models have 0% accuracy after the attack. Our choices ensure a thorough test of the adversarial robustness in a white-box setting, revealing the multifacetedness of the adversarial robustness. Information about the generalization benchmarks is given in Section S.1.

### L.2 GENERALIZATION ABILITY EVALUATION

On CIFAR10, we report the Top-1 accuracy on the CIFAR10-C (Hendrycks & Dietterich, 2018), CI-FAR10.1 (Recht et al., 2018), CIFAR10.2 (Lu et al., 2020), as the metric for evaluating generalization. CIFAR10-C tests the out-of-distribution (OOD) generalization and CIFAR10.1, CIFAR10.2 evaluate the in-distribution (ID) one. On ImageNet, we show the Top-1 accuracy on ImageNet-C (IM-C) (Hendrycks & Dietterich, 2018), ImageNet-A (IM-A) (Djolonga et al., 2021), and ImageNet-ReaL (IM-ReaL) (Beyer et al., 2020). IM-C and IM-A are designed for benchmarking OOD generalization. IM-ReaL tests ID generalization.

Table 8: Hyperparameters of all attacks used in AutoAttack (Croce & Hein, 2020) in our ablation studies on ImageNet100.

|  | PGD | CW | APGD-DLR | APGD-CE |
|---|---|---|---|---|
| Max magnitude | 0.5/255 | / | 0.5/255 | 0.5/255 |
| Steps | 5 | 10 | 5 | 5 |
| Step size | 0.5 | / | / | / |
| $\kappa$ | / | 0.0 | / | / |
| $c$ | / | 0.1 | / | / |
| $\rho$ | / | / | 0.75 | 0.75 |
| EOT | / | / | 1 | 1 |

Table 9: Quantitative comparisons of *CaDistill* ImageNet (Deng et al., 2009) (ResNet-50). Adversarial robustness benchmarks: PGD (Madry et al., 2018), CW (Carlini & Wagner, 2017), APGD-DLR / APGD-CE (Croce & Hein, 2020); Evaluations of generalization ability: ImageNet-C (Hendrycks & Dietterich, 2018), ImageNet-A (Djolonga et al., 2021), ImageNet-ReaL (Beyer et al., 2020). Data$_{DM}$ is the portion of data for which the DM acts as teacher. Higher is better. Values lower than the vanilla model are in red. See Section M for an analysis of the comparison between SimCLR and CLIP.

| Model | Data$_{DM}$ | Clean | PGD | CW | APGD-DLR | APGD-CE | IM-C | IM-A | IM-ReaL |
|-------|-------------|-------|-----|-----|----------|---------|------|------|---------|
| Vanilla | / | 75.9 | 15.6 | 13.7 | 17.2 | 16.7 | 45.9 | 6.3 | 82.8 |
| SimCLR (2020b) | / | 74.9 | 9.0 | 7.0 | 10.8 | 10.4 | 42.8 | 4.8 | 81.2 |
| CLIP (2021) | / | 68.2 | 7.0 | 8.8 | 0.2 | 1.1 | 21.2 | 11.4 | 74.4 |
| DiffAug (2024) | 100% | **76.0** | 15.9 | 13.1 | 17.2 | 17.0 | **47.2** | 4.8 | 83.1 |
| DMDistill | 100% | 75.7 | 15.7 | 14.1 | 17.0 | 16.7 | 43.6 | 5.0 | 82.8 |
| CFGDistill | 10% | 75.7 | 20.8 | 20.3 | 20.8 | 21.4 | 45.6 | 6.0 | 82.7 |
| *CaDistill* | 10% | 75.9 | **21.9** | **21.7** | **22.5** | **22.3** | 46.1 | **6.7** | **83.1** |

Due to the limits in computational resources, we do not report error bars in our experiments. During testing, we find that different runs of adversarial attacks result in similar performance. We test each adversarial attack 3 times with different seeds and find that the resulting performance has standard deviations all smaller than 0.05.

## M  MORE RESULTS ON IMAGENET

We provide additional results of *CaDistill* using ResNet50 on ImageNet in Table 9, 10. ResNet50 shows a similar trend as ResNet152 in the main paper. Hence, we perform extensive ablation studies and comparisons on ResNet50 in this section for efficiency.

**Comparison to a vision-language model and a self-supervised learning method.** In Table 9, 10, we compare models trained with *CaDistill* against two well-established baselines:

1. CLIP (Radford et al., 2021). It is a vision-language model that learns visual representations from image-text contrastive supervision. We use the ResNet50 vision backbone from the CLIP family. CLIP is pretrained on web-scale data rather than ImageNet. This difference in data distribution helps explain its strong performance on ImageNet-A, in which the images fall outside the ImageNet distribution but can be closer to CLIP's pretraining distribution. For ImageNet1K probing, we use a nonlinear head (Linear-GELU-Linear) trained for 90 epochs with batch size 256, learning rate $1e-3$, AdamW, and no weight decay. The original paper reports 73.3% Top-1 using a single-batch L-BFGS linear probe, which is impractical in our setting due to memory requirements. Our linear head achieves 65.6% Top-1. Hence, we use the nonlinear head to improve the performance.

2. SimCLR (Chen et al., 2020a;b). It is a self-supervised method that learns invariances by contrasting two strongly augmented views of the same image. The augmentation is designed to be semantic-preserving. We use the official ResNet50 ($1\times$, no selective kernels) model (Chen et al., 2020b) trained on ImageNet1K and the official 100% finetuned checkpoint, which includes an ImageNet1K classification head.

Note that the data augmentation approaches and training epochs are not controlled in the comparison, because CLIP needs text as a source of supervision and SimCLR requires long training to be effective. Only the model architecture and size are controlled. Note that the CLIP ResNet50 has a slightly different architecture from the one used in our experiments and SimCLR. We emphasize that our approach complements mainstream supervised and self-supervised representation learning (Section 5 and O). These comparisons are provided for context rather than as a replacement claim: our goal is to offer a fresh perspective on learning representations, not to supplant existing paradigms.

### M.1  THE PERFORMANCE OF $\mathcal{L}_{dist}$ ALONE IN FEATURE DISTILLATION

In Section 4.1, we design a baseline experiment that distills the structure of the raw diffusion features into the representation space of the student network, which is based on $\mathcal{L}_{dist}$. Our design of $\mathcal{L}_{dist}$

Table 10: Results on the Backgrounds Challenge (Xiao et al.) using ResNet50. Higher is better. See Section M.5 for details.

| Model | Original | BG-Same | BG-Rand | Only-FG |
|---|---|---|---|---|
| Vanilla | 96.0 | 88.0 | 81.1 | 87.6 |
| SimCLR (2020b) | 94.9 | 86.2 | 79.7 | 84.9 |
| CLIP (2021) | 87.7 | 67.9 | 56.2 | 71.7 |
| DiffAug (2024) | 96.1 | 87.5 | 80.3 | 87.4 |
| DMDistill | **96.3** | 88.0 | 80.6 | 84.5 |
| CFGDistill | **96.3** | **89.0** | 82.6 | 87.8 |
| *CaDistill* | 96.3 | 89.0 | **83.6** | **88.5** |

Table 11: Comparison between our CKA-based (Kornblith et al., 2019) $\mathcal{L}_{dist}$ and other loss functions in diffusion-based feature distillation.

| $\mathcal{L}_{cano}$ | Clean | AutoAttack (2020) |
|---|---|---|
| Vanilla (2016) | 86.5 | 15.9 |
| FitNet (2023b; 2015; 2023) | 86.7 | 16.4 |
| AT (2023b; 2023; 2017) | 86.6 | 16.3 |
| RKD (2019; 2023) | 86.2 | 16.4 |
| Ours | **87.3** | **18.8** |

differs from all previous works on diffusion-based feature distillation. We use a CKA (Kornblith et al., 2019) metric for measuring the linear subspace alignment between the feature vectors of the student and the teacher. CKA is invariant to isotropic scaling as well as orthonormal transformations. Such an invariance lets us transfer the class-discriminative structure encoded in Canonical Features without over-constraining the student's own feature basis. Previous works focus on using three classical feature distillation losses: (1) FitNet (Li et al., 2023b; Romero et al., 2015; Yang & Wang, 2023), which is the L2 distance between student features and the teacher's; (2) Attention transfer (AT) (Li et al., 2023b; Yang & Wang, 2023; Zagoruyko & Komodakis, 2017), which distills the saliency structure of the activation map to the student; (3) Relational knowledge distillation (RKD) (Park et al., 2019; Yang & Wang, 2023), which aligns the relational representations of the samples between the teacher and the student. However, in our case, these loss functions do not significantly contribute to the student's performance, as shown in Table 11. Note that in these experiments, the student features and the teacher ones are one-to-one matching to mimic the typical feature distillation framework, instead of the random strategy as we designed in *CaDistill*.

Our CKA-based (Kornblith et al., 2019) feature distillation loss outperforms all previous designs without introducing additional parameters during training. This is a novel loss function used in a diffusion-based feature distillation framework, inspired by previous works (Dapello et al., 2023; Saha et al., 2022; Zhou et al., 2024b). Performing knowledge transfer with the teacher and/or the student being a ViT is still an open question (Yang et al., 2024; Yao et al., 2022), and can lead to a performance drop in the student. Our $\mathcal{L}_{dist}$, however, achieves good performance in the diffusion-based settings.

## M.2 *CaDistill* IS EFFECTIVE WITH DIFFERENT DATA AUGMENTATION STRATEGIES

We show that *CaDistill* is effective when the data augmentation strategy is different, demonstrating the generalization of the paradigm. Specifically, the training lasts 120 epochs, and the data augmentations are:

- `torchvision.transforms.RandomResizedCrop(224)`,

- `torchvision.transforms.RandomHorizontalFlip()`,

- `torchvision.transforms.ColorJitter(0.3, 0.3, 0.3)`,

Table 12: Quantitative comparisons between *CaDistill* and baselines on ImageNet (Deng et al., 2009) with a ResNet50 (He et al., 2016), **using a different training setting (Section M.2)** from the one in Section L. Higher is better.

| Model | Data$_{DM}$ | Clean | PGD | CW | APGD-DLR | APGD-CE | IM-C | IM-A | IM-ReaL |
|---|---|---|---|---|---|---|---|---|---|
| Vanilla | / | 76.6 | 17.3 | 13.5 | 18.8 | 17.7 | 40.6 | 3.4 | 83.3 |
| *CaDistill* | 10% | 76.7 | **21.3** | **21.3** | **22.6** | **21.3** | **41.2** | **4.2** | **83.3** |

Table 13: Quantitative comparisons between *CaDistill*, and baselines on ImageNet (Deng et al., 2009) with a ResNet50 (He et al., 2016), on black-box adversarial robustness. Higher is better. Red is lower than the vanilla model. Data$_{DM}$: the portion of the subset on which the diffusion model serves as the teacher. DMDistill: Feature distillation by $\mathcal{L}_{dist}$ on the whole dataset using a DiT model; CFGDistill: Using the framework of *CaDistill*, but replace CanoReps by samples with CFG from the CDM.

| Model | Data$_{DM}$ | Clean | Square (Andriushchenko et al., 2020) |
|---|---|---|---|
| Vanilla | / | 75.9 | 23.5 |
| DiffAug (Shama et al., 2024) | 100% | 76.0 | 20.7 |
| DMDistill | 100% | 75.7 | 22.9 |
| CFGDistill | 10% | 75.7 | 23.3 |
| *CaDistill* | 10% | 75.9 | **25.6** |

The learning rate is 0.2, and the decay happens every 30 epochs with a decay rate of 0.1. The results are in Table 12. *CaDistill* yields a student that outperforms the vanilla model on all benchmarks, proving the effectiveness of our method in this case and implying its generalization capability.

## M.3   *CaDistill* IMPROVES THE STUDENT'S BLACK-BOX ADVERSARIAL ROBUSTNESS

In Table 1, we demonstrate that *CaDistill* improves the student's white-box robustness. Here, we show that *CaDistill* improves the student performance when facing black-box adversarial attacks. Specifically, we test all models on ImageNet using the Square attack (Andriushchenko et al., 2020), which is a black-box adversarial attack algorithm. We use $L_{inf}$ metric with attacking budget 4.0/255.0, and the query number is 1000. To reduce computational costs, we randomly select 2000 samples from ImageNet to perform the evaluation. The result is given in Table 13.

## M.4   EXTRACTING CLASS SEMANTICS USING CLASS TOKENS IN VISION TRANSFORMER

Our main claim on the application of CanoReps is that they distill the core class semantics of each category. Here, we investigate another mainstream approach to distill such information, which is the class token in vision transformer (ViT) (Dosovitskiy et al., 2020; Touvron et al., 2021; 2022). The class token performs the attention operation (Vaswani et al., 2017) to all spatial tokens, collecting the discriminative signals inside the feature map for classification. We adopt a challenging baseline network, DeiT-III-Huge (Touvron et al., 2022), which is a ViT model solely trained on ImageNet with the number of parameters (DeiT: 632.1M; DiT: 675M) and FLOPS (DeiT: 167.4G; DiT: 118.6G) matching the DiT (Peebles & Xie, 2023) used in our experiments. It is challenging because the DeiT-III-Huge model is trained with advanced data augmentation techniques and performs well on ImageNet classification tasks (Touvron et al., 2022), whereas DiT is trained with a plain horizontal flip augmentation and is not good at classification (Li et al., 2023a) (85.2 v.s. 77.5 Top-1 accuracy). We use our CKA-based $\mathcal{L}_{dist}$ to align the representations of the student network to the class token in DeiT-III-Huge, termed DeiT$_{dist}$. All the settings are the same as used in Section 4.1 and L. The results are given in Table 14. Notably, the student trained with *CaDistill* outperforms the one trained with DeiT$_{dist}$ in terms of clean accuracy and generalization. Achieving good performance in feature distillation between ViT and CNNs is still an open problem in the field (Yao et al., 2022). Despite this, our experiments control the architecture (both teachers are ViTs), the number of parameters, and the FLOPS. Moreover, the DeiT$_{dist}$ can yield a student that outperforms DMDistill in terms of all adversarial attack benchmarks, which demonstrates the effectiveness of the method and validity of our

Table 14: Quantitative comparisons between *CaDistill*, and baselines on ImageNet (Deng et al., 2009) with a ResNet50 (He et al., 2016). Higher is better. Red is lower than the vanilla model. Data$_{DM}$: the portion of the subset on which the diffusion model serves as the teacher. DeiT$_{dist}$: Feature distillation by $\mathcal{L}_{dist}$ on the whole dataset using the class token in an ImageNet-pretrained DeiT-III-Huge model (Touvron et al., 2022); DMDistill: Feature distillation by $\mathcal{L}_{dist}$ on the whole dataset using a DiT model; CFGDistill: Using the framework of *CaDistill*, but replace CanoReps by samples with CFG from the CDM.

| Model | Data$_{DM}$ | Clean | PGD | CW | APGD-DLR | APGD-CE | IM-C | IM-A | IM-ReaL |
|---|---|---|---|---|---|---|---|---|---|
| Vanilla | / | 75.9 | 15.6 | 13.7 | 17.2 | 16.7 | 45.9 | 6.3 | 82.8 |
| DeiT$_{dist}$ (Touvron et al., 2022) | 100% | 75.1 | 19.7 | 18.4 | 20.8 | 20.3 | 45.2 | 5.4 | 82.5 |
| DMDistill | 100% | 75.7 | 15.7 | 14.1 | 17.0 | 16.7 | 43.6 | 5.0 | 82.8 |
| CFGDistill | 10% | 75.7 | 20.8 | 20.3 | 20.8 | 21.4 | 45.6 | 6.0 | 82.7 |
| *CaDistill* | 10% | 75.9 | **21.9** | **21.7** | **22.5** | **22.3** | **46.1** | **6.7** | **83.1** |

experiments. We believe investigating the difference between the mechanisms of how discriminative models and generative ones encode class information is an interesting future direction.

## M.5 DETAILS OF THE BACKGROUND CHALLENGE

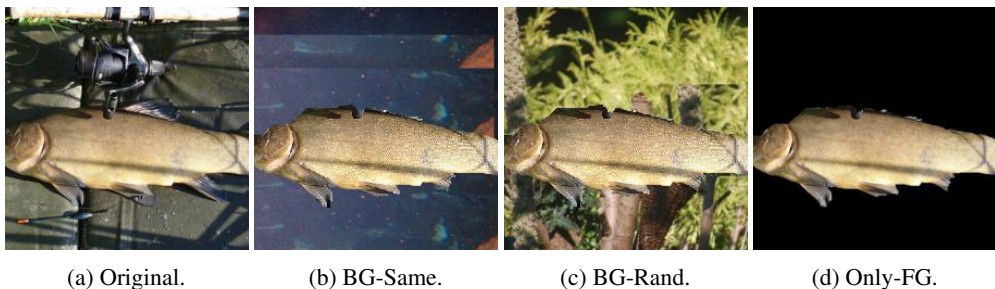

(a) Original.    (b) BG-Same.    (c) BG-Rand.    (d) Only-FG.

Figure 35: Samples from the Backgrounds Challenge (Xiao et al.). (a) Original: The original image. (b) BG-Same: Put a random background from the same class onto the image. (c) BG-Rand: Put a random background from a different class onto the image. (d) Only-FG: Discard the background and make it black.

In Section 4.1, we test the student model on the Backgrounds Challenge (Xiao et al.). Figure 35 illustrates its three variants: BG-Same re-uses a single background per class; BG-Rand pairs each foreground with randomly chosen backgrounds from other classes; Only-FG removes the background entirely. These operations preserve the foreground object while removing background cues. Hence, the performance in this test quantifies a model's ability to rely on true class signals rather than spurious background correlations.

As shown in Table 2, *CaDistill* achieves the highest accuracy across all splits. CFGDistill matches *CaDistill* on the Original and BG-Same sets, but *CaDistill* outperforms it on BG-Rand and Only-FG. This gap indicates that the student trained with CFGDistill still uses background information shared within each class; when those backgrounds are shuffled or removed, its accuracy declines. The evidence suggests that CFG introduces label-correlated yet non-essential background signals into the training data, whereas *CaDistill* suppresses those signals and encourages the student to focus on the foreground object.

## M.6 ON THE REPRODUCTION OF REPFUSION

RepFusion (Yang & Wang, 2023) proposes a novel framework for diffusion-based feature distillation. It uses a neural network for adaptively selecting the time step of feature extraction from the teacher DM. This neural network is trained using the REINFORCE (Williams, 1992) algorithm, using the task performance as the reward. The task performance is the classification accuracy. In this case, the neural network is non-linear and can directly decode the label conditioning in the CDM to maximize

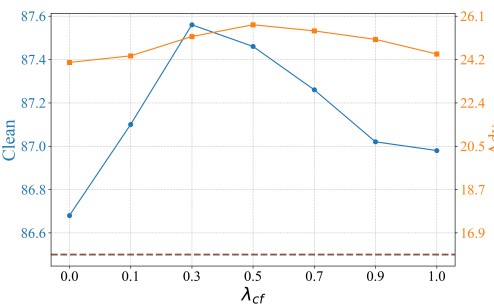 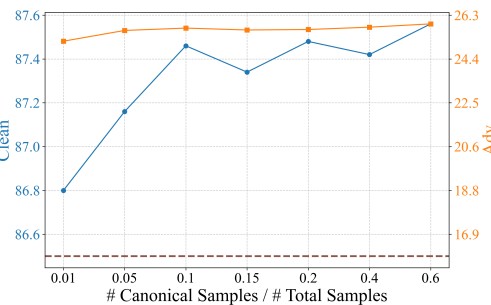

Figure 36: Ablation on $\lambda_{cf}$. An effective training requires a trade-off between $\mathcal{L}_{align}$ and $\mathcal{L}_{cano}$, and necessitates both of them. We choose $\lambda_{cf} = 0.5$ to balance between the Clean accuracy and robustness. Brown is the baseline of both Clean and Adv.

Figure 37: Ablation on the number of CanoReps. A small amount of CanoReps, *e.g.* 10%, is sufficient for achieving competitive performance. It implies the low-dimensionality property of the class manifolds inside CDMs, which is in line with previous findings (Wang et al.).

the reward, leading to a failed training. Hence, we reproduce the method on an unconditional DM. Despite this, we provide a strong baseline using CDM, DMDistill, which uses a feature distillation loss that can outperform all the loss functions used in RepFusion, as shown in Section M.1.

## M.7 ABLATION STUDIES

We conduct ablation studies on ImageNet100, a 100-class subset of ImageNet. Previous studies (Cheng et al., 2023; Douillard et al., 2022; Xu et al., 2024; Yan et al., 2021; Yu et al., 2022) have shown that ImageNet100 serves as a representative subset of ImageNet1K. Hence, we can obtain representative results for the self-evaluation of the model while efficiently using our computational resources. All the ablations are based on a ResNet50 (He et al., 2016) model. Note that we still use an ImageNet-pretrained DiT (Peebles & Xie, 2023) as the teacher. We report the test accuracy on the ImageNet100 validation set as the clean accuracy (Clean), and the adversarial accuracy (Adv) under AutoAttack (Croce & Hein, 2020) that consists of PGD (Madry et al., 2018), CW (Carlini & Wagner, 2017), APGD-DLR (Croce & Hein, 2020), and APGD-CE (Croce & Hein, 2020).

### M.7.1 NUMBER OF CANOREPS

Figure 37 shows the student performance when trained with different numbers of CanoReps. Remarkably, training with as little as 10% of the available CanoReps already yields near-optimal performance. The result suggests that a small data subset is enough to capture the core class semantics in CDMs, because those semantics lie on a low-dimensional manifold, which is consistent with earlier findings (Wang et al.).

### M.7.2 THE NECESSITY OF $\mathcal{L}_{align}$ AND $\mathcal{L}_{cano}$ AND THEIR BALANCE

Our design includes on two complementary objectives: the alignment loss, $\mathcal{L}_{align}$, which pulls each sample towards CanoReps from its class, and the CanoRep separation loss, $\mathcal{L}_{cano}$, which drives the CanoReps of different classes apart. Figure 36 demonstrates the trade-off between the two. If $\mathcal{L}_{align}$ is omitted ($\lambda_{cf} = 0$), samples remain distant from their canonical counterparts, preventing the student from learning the core semantics of each class. Conversely, dropping $\mathcal{L}_{cano}$ ($\lambda_{cf} = 1$) can lead to CanoReps that collapse together, leaving the student unable to discriminate between categories. Therefore, the optimal choice requires a balance between them.

### M.7.3 THE NECESSITY OF $\mathcal{L}_{dist}$

The CDM transfers the core class features using Canonical Features via $\mathcal{L}_{dist}$. Without this loss, the student can fail to learn the encoded features of the Canonical Samples, which can negatively affect the student's clean accuracy and adversarial robustness, as shown in Table 15.

Table 15: The ablation study of the effects of $\mathcal{L}_{dist}$ on ImageNet (Deng et al., 2009) with a ResNet50 (He et al., 2016). Vanilla: The original student network. Data$_{DM}$: the portion of the subset on which the diffusion model serves as the teacher. DMDistill: Feature distillation by $\mathcal{L}_{dist}$ on the whole dataset; CFGDistill: Using the framework of **CaDistill**, but replace Canonical Samples by samples generated with CFG after $F_{inv}$, and use their corresponding features in the CDM. Higher is better. Green is lower than the vanilla model. Without $\mathcal{L}_{dist}$, the student cannot learn the teacher's encoding of CanoReps, limiting the student's adversarial robustness.

| Model | Data$_{DM}$ | Clean | PGD | CW | APGD-DLR | APGD-CE |
|---|---|---|---|---|---|---|
| Vanilla | / | 75.9 | 15.6 | 13.7 | 17.2 | 16.7 |
| DiffAug (Shama et al., 2024) | 100% | 76.0 | 15.9 | 13.1 | 17.2 | 17.0 |
| DMDistill | 100% | 75.7 | 15.7 | 14.1 | 17.0 | 16.7 |
| CFGDistill | 10% | 75.7 | 20.8 | 20.3 | 20.8 | 21.4 |
| **CaDistill** | 10% | 75.9 | **21.9** | **21.7** | **22.5** | **22.3** |
| No $\mathcal{L}_{dist}$ | 10% | 75.6 | 20.3 | 19.3 | 20.5 | 21.9 |

### M.7.4 THE WEIGHTS OF LOSSES, $\lambda_{cs}, \lambda_{dist}, \lambda_{cka}$

**CaDistill** involves 3 losses, $\lambda_{cs}, \lambda_{dist}, \lambda_{cka}$, each having its own weights. Here, we perform thorough ablation studies on $\lambda_{cs}, \lambda_{dist}, \lambda_{cka}$ on ImageNet100. The results are given in Figure 38. We perform the ablation study on one loss function by fixing the other weights to their own optimal values. We empirically conclude this setting: $\lambda_{cs} = 0.4, \lambda_{dist} = 1.0, \lambda_{cka} = 0.5$, for all experiments on ImageNet. For CIFAR10, we perform grid search over several parameter combinations and fix $\lambda_{cs} = 0.2, \lambda_{dist} = 0.25, \lambda_{cka} = 0.5$.

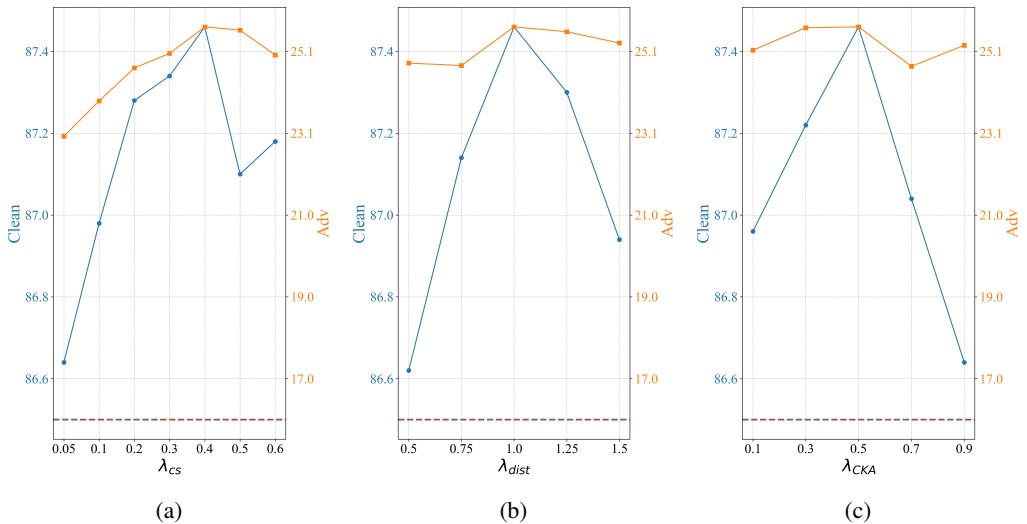

(a)  (b)  (c)

Figure 38: Ablation studies on $\lambda_{cs}, \lambda_{dist}, \lambda_{cka}$, introduced in Section 4. We select $\lambda_{cs} = 0.4, \lambda_{dist} = 1.0, \lambda_{cka} = 0.5$ in all of our experiments on ImageNet.

### M.7.5 THE DESIGN OF $\mathcal{L}_{cano}$

We design $\mathcal{L}_{align}$ and $\mathcal{L}_{cano}$ to both have push-together and pull-away effects, inspired by Khosla et al. (2020). In Eq. equation 4, each Canonical Sample treats other Canonical Samples of the same class (excluding itself) as positive examples. If a class happens to contribute only a single Canonical Sample, no such positives exist. In that case, we optimize only the "pull-away" term—the denominator that separates the anchor from negatives in other classes, so the loss remains well-defined

Table 16: The ablation study on using cross-entropy as $\mathcal{L}_{cano}$. Our design in Eq. 4 achieves better results.

| $\mathcal{L}_{cano}$ | Clean | AutoAttack (Croce & Hein, 2020) |
|---|---|---|
| Vanilla (He et al., 2016) | 86.5 | 15.9 |
| Cross-entropy | 86.8 | 25.3 |
| Ours | 87.5 | 25.7 |

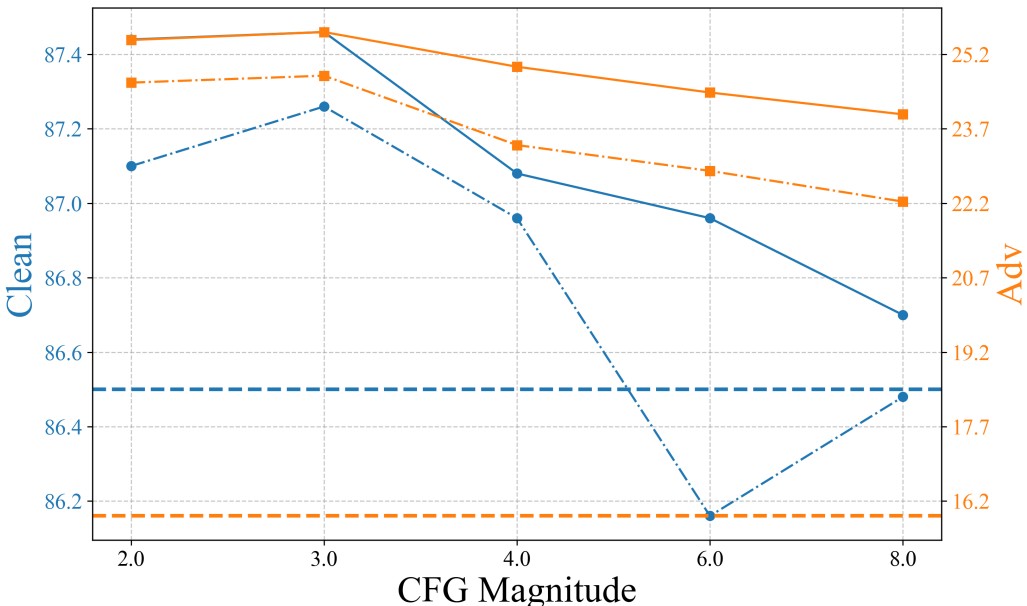

Figure 39: The ablation study on the magnitude of CFG used in our experiments on ImageNet. The student is trained with *CaDistill* (solid lines) and CFGDistill (dash-dot lines). The dashed lines are the baselines. Larger CFG magnitudes do not necessarily contribute to a better performance, indicating that our design in *CaDistill* is not simply a converging prior on the features (Section G.3).

and informative. In this case, $\mathcal{L}_{cano}$ becomes:

$$\mathcal{L}_{cano} = \frac{1}{b} \sum_{i=1}^{b} \log \sum_{k \neq i} \exp\left(\tilde{z}_i \cdot \tilde{z}_k / \tau\right). \tag{12}$$

We perform a simple ablation study on using cross-entropy for discriminating between CanoReps from different classes, using ImageNet100. The results are given in Table 16. Our design achieves better results in both clean accuracy and adversarial robustness, which is in line with the previous claim (Khosla et al., 2020). In our case, CanoReps are far less than the original images and are easier to classify, which can cause overfitting issues when using cross-entropy and lead to suboptimal results.

### M.7.6 THE MAGNITUDE OF CLASSIFIER-FREE GUIDANCE

On ImageNet, we use CFG after projecting away the extraneous directions. We perform an ablation study on the CFG magnitude. The results are shown in Figure 39. Notably, larger CFG magnitudes do not correspond to better performance. An overly large CFG scale can even worsen student performance. This is because our *CaDistill* are not simply providing a converging prior over the student features, as discussed in Section G.3. We choose the magnitude to be 3 for both *CaDistill* and CFGDistill.

Table 17: Comparison between a vanilla Swin-Tiny (Liu et al., 2021) model, a FAN-Small (Zhou et al., 2022) model, and the ones trained with **CaDistill**. Our method is effective with transformer students. It also proves that **CaDistill** is effective with modern data augmentation techniques such as Mixup (Zhang et al., 2018) and CutMix (Yun et al., 2019).

| Model | Clean | AutoAttack (Croce & Hein, 2020) |
|---|---|---|
| Swin-Tiny (Liu et al., 2021) | 81.8 | 9.3 |
| **CaDistill** | **84.4** | **13.8** |
| FAN-Tiny (Zhou et al., 2022) | 84.3 | 25.8 |
| **CaDistill** | **84.4** | **29.1** |

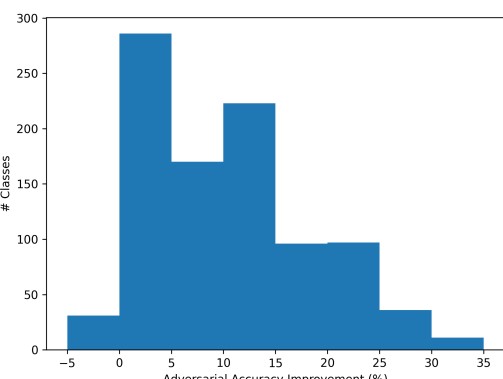

Figure 40: Class-wise adversarial robustness improvement of ResNet50 on ImageNet100 using AutoAttack (Croce & Hein, 2020). **CaDistill** brings performance gain in almost all classes.

## M.8 GENERALIZATION OF *CaDistill* TO DIFFERENT STUDENT ARCHITECTURES

We demonstrate that **CaDistill** is effective when the student is a transformer architecture. Specifically, we train a Swin-Tiny (Liu et al., 2021) model and a FAN-Tiny (Zhou et al., 2022) model on ImageNet100. The result is given in Table 17. We train the networks using the same setting as described in Section L, except that we follow the timm data augmentation with Mixup (Zhang et al., 2018) and CutMix (Yun et al., 2019), and we have a 5-epoch learning rate warm-up.

## M.9 CLASS-WISE ROBUSTNESS IMPROVEMENT

We investigate whether the improvement is limited to a small set of classes or is spread across the entire dataset. In Figure 40, we plot a histogram of the per-class percentage improvement in adversarial accuracy under AutoAttack (Croce & Hein, 2020), using the same setting in Section M.7. The histogram shows that almost all classes benefit from our method, indicating that the gains are broadly distributed rather than concentrated in only a few categories.

We further check whether the most-improved or least-improved classes align with specific super-classes, such as "animals" or "artifacts". We list the 20 most improved classes:

['hen-of-the-woods', 'racket', 'hip', 'go-kart', 'hyena', 'jacamar', 'Afghan hound', 'orangutan', 'three-toed sloth', "potter's wheel", 'lion', 'proboscis monkey', 'ostrich', 'steam locomotive', 'cannon', 'Komodo dragon', 'black and gold garden spider', 'partridge', 'gong', 'yurt'];

And 20 least improved classes:

['window screen', 'horned viper', 'beach wagon', 'rugby ball', 'hoopskirt', 'bib', 'Doberman', 'kuvasz', 'Loafer', 'dial telephone', 'daisy', 'revolver', 'thunder snake', 'Kerry blue terrier', 'cocktail shaker', 'electric guitar', 'miniature pinscher', 'convertible', 'patio', 'Ibizan hound'].

The results show that these classes are not clustered within only a small number of superclasses, suggesting that the robustness improvement is not biased toward any particular semantic group.

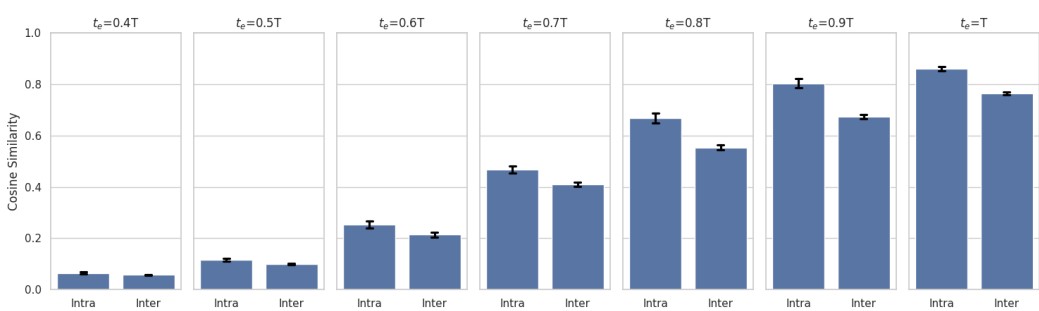

Figure 41: The cosine similarity between extraneous directions in both intra and inter class cases, with different $t_e$ values. The error bar is the 95% CI.

## M.10    THE VARIATION OF EXTRANEOUS DIRECTIONS WITHIN AND BETWEEN CLASSES

We investigate the variation of extraneous directions within each class or across different classes. Specifically, we compute the cosine similarity between the top-10 extraneous directions that we obtain in the ImageNet20 experiment. In the intra-class case, we compute pair-wise cosine similarities between the top-10 directions and average them; In the inter-class case, we find an optimal one-to-one correspondence between the two sets of directions by solving the Hungarian matching problem, which maximizes the total pairwise similarity between matched directions. We conduct this experiment with $t_e \in \{0.4T, 0.5T, 0.6T, 0.7T, 0.8T, 0.9T, T\}$. The results are shown in Figure 41, with 95% confidence interval as the error bar. First, we observe that the similarity generally increases when $t_e$ becomes larger. The values at $t_e = 0.9T$ and $T$ are consistent with the prior work (Park et al., 2023b), validating our implementation. We also find that intra-class similarity is consistently higher than inter-class similarity. This is intuitive, as images from the same class often share similar backgrounds (e.g., most photos of sea fish are taken in the sea), leading to more similar extraneous directions within a class.

## N    SIGNIFICANCE OF THE QUANTITATIVE RESULTS AND DISCUSSION

### N.1    SIGNIFICANCE OF THE IMPROVEMENTS BROUGHT BY *CaDistill*

Our proposed method, **CaDistill**, consistently improves the adversarial robustness and generalization ability of the student model. While the baseline methods can achieve better results on some benchmarks (*e.g.* the IM-C (Hendrycks & Dietterich, 2018) test on DiffAug (Shama et al., 2024)), they can worsen the student's performance on other benchmarks (Red marks). This phenomenon reveals the established observation: the multifacetedness of robustness. Despite this, **CaDistill** still consistently improves the student's performance, and it is the only one that is capable of doing so. This is neither a trivial nor marginal gain. Empirical and theoretical evidence suggest that there is a trade-off between different kinds of robustness (Moayeri et al., 2022; Rusak et al., 2020). Adversarial training can hurt corruption robustness (Rusak et al., 2020), while noise-based training weakens adversarial robustness (Table 1, row DiffAug). Adversarially robust models may even rely more on spurious cues (Moayeri et al., 2022). Our results and the CFGDistill rows exemplify these trade-offs, while **CaDistill** mitigates them by reducing reliance on spurious correlations.

### N.2    ORIGIN OF IMPROVEMENTS

We argue that CFGDistill and **CaDistill** use different mechanisms to achieve adversarial robustness. First, CFGDistill shows a trade-off between adversarial robustness and generalization, a typical failure mode of current deep learning models (see above Significance of the improvements brought by **CaDistill**). In contrast, **CaDistill** overcomes such a trade-off. CFGDistill relies more on spurious correlations than **CaDistill**, as shown in Table 2. This shows that Canonical Samples matter as much as the loss. Our losses indeed contribute to adversarial robustness, but the structure of the involved samples plays an important role. We add a PlainDistill variant, where the samples are generated without CFG using DiT, to ablate sample effects. The results are given in Table 18.

Table 18: Quantitative comparisons between *CaDistill* and CFGDistill, PlainDistill on ImageNet (Deng et al., 2009) (ResNet-50). Adversarial robustness benchmarks: PGD (Madry et al., 2018), CW (Carlini & Wagner, 2017), APGD-DLR / APGD-CE (Croce & Hein, 2020); Evaluations of generalization ability : ImageNet-C (Hendrycks & Dietterich, 2018), ImageNet-A (Djolonga et al., 2021), ImageNet-ReaL (Beyer et al., 2020). Data$_{DM}$ is the portion of data for which the DM acts as teacher. Higher is better. Values lower than the vanilla model are in red.

| Model | Data$_{DM}$ | Clean | PGD | CW | APGD-DLR | APGD-CE | IM-C | IM-A | IM-ReaL |
|---|---|---|---|---|---|---|---|---|---|
| Vanilla | / | 75.9 | 15.6 | 13.7 | 17.2 | 16.7 | 45.9 | 6.3 | 82.8 |
| PlainDistill | 10% | 75.4 | 17.3 | 15.7 | 18.3 | 18.7 | 45.3 | 5.8 | 82.6 |
| CFGDistill | 10% | 75.7 | 20.8 | 20.3 | 20.8 | 21.4 | 45.6 | 6.0 | 82.7 |
| *CaDistill* | 10% | 75.9 | **21.9** | **21.7** | **22.5** | **22.3** | 46.1 | **6.7** | **83.1** |

Adversarial robustness benefits from a converging prior in feature space (Pang et al.), that same-class samples are pulled together. Both CFGDistill and *CaDistill* provide such a prior, but they converge to different manifolds (Section 3.3): CFG's manifold can encode non-causal structure, while Canonical samples encode the class core, yielding broader robustness. The ablation study in Section M.7.6 shows that excessive CFG magnitude degrades performance, demonstrating that CFG alone can impose the wrong structure (discussion in Section G.3; visual results in Figure 34). *CaDistill*'s advantage stems from both the Canonical Samples' semantic structure and our loss design, producing comprehensive gains across various types of robustness. The results also suggest that Canonical Samples encode fundamentally different information, as the distillation from the two kinds of representations yields qualitatively (the trend of robustness improvement) and quantitatively (the absolute values of the robustness improvement) different models.

### N.3    DATA EFFICIENCY OF *CaDistill*

The baseline methods that do not use our proposed feature distillation pipeline all necessitate access to the full dataset, while our methods achieve competitive performance with access only to 10% of the data. Given that the teacher model is often large and running it on the whole dataset can lead to high computational costs, this reduction in data dependency demonstrates the efficiency of our method.

### N.4    DIFFERENCE OF THE RESULTS ON CIFAR10 AND IMAGENET

The performance trends differ in CIFAR10 and ImageNet (Table 1). On CIFAR10, even if simply distilling the raw diffusion features to the student via DMDistill can contribute to the performance on all benchmarks, while on ImageNet, all the baseline methods can perform worse than the vanilla model in certain cases. We assume that two factors can lead to such a difference. The first is that CIFAR10 is a simpler dataset than ImageNet. Most images in CIFAR10 contain purely foreground objects, while the images in ImageNet are much noisier and harder to classify. The evidence is that CIFAR10 is a nearly solved dataset, as the classification accuracy approaches 100% (Dosovitskiy et al., 2020), while the top models on ImageNet can achieve ∼90% (Yu et al.).

More importantly, we identify a critical difference in the CDM on CIFAR10 and on ImageNet. That is, ImageNet-trained CDMs are typically trained in a low-resolution latent space of a pre-trained variational autoencoder (VAE) (Kingma & Welling, 2014; Rombach et al., 2022). This low-resolution space loses detailed information compared to the pixel space, reducing the discriminative power. To validate, we train a ResNet50 (He et al., 2016) in this latent space for image classification, receiving the input as the VAE-encoded images. Notably, the clean accuracy drastically drops from 86.5 to 76.6. We assume that the missing discriminative information can negatively affect the performance of all feature distillation methods, because the diffusion features lie in the low-resolution latent space. Due to the limits on computational resources, we leave the investigation on pixel-space DM on ImageNet as a future direction.

Table 19: The Pearson correlation $r$ between the feature similarity matrices of different models and the ground truth class structure matrix (Huang et al., 2021) obtained by Wu-Palmer distance (Wu & Palmer, 1994) in Figure 18.

|  | $r$ | $p$ |
|---|---|---|
| ResNet50 | 0.19 | $< 0.0001$ |
| DeiT-Base-cls | 0.05 | 0.6 |
| DiT | 0.31 | $< 0.0001$ |
| CFG | 0.28 | $< 0.0001$ |
| Canonical Features | 0.48 | $< 0.0001$ |

## O  DISCUSSION AND POTENTIAL APPLICATIONS OF CLARID

CLARID identifies CanoReps that encode core class semantics while suppressing class-irrelevant information. Our *subtractive view contrasts and complements current representation learning research*, rely on supervised signals or contrastive objectives to learn semantics from inputs or to enforce invariances across inputs (e.g., vision–language models, VLMs, and self-supervised learning, SSL). We demonstrate that representations beneficial for robustness and generalization can be extracted from the full feature set used by the generator to synthesize the inputs. The full set itself, however, is less useful for high-level recognition as it encodes too much redundant information, as shown by our DMDistill experiments. This perspective does not contradict prior findings in diffusion-based representation learning showing gains in distillation (Li et al., 2023b; Yang & Wang, 2023), because they focus on low-level dense prediction tasks such as image segmentation and face landmark detection. In contrast, CLARID and *CaDistill* show that high-level semantics are also present in diffusion models. They simply need to be extracted by our principled method.

At a high level, this perspective aligns with generative similarity (Marjieh et al., 2024), where reliable object relationships and human-like class structure perception emerge from generative features, not purely from supervised or contrastive signals. The implication is that the generative process yields causal features that define the core semantics of objects (Section 3.3). We take a first practical step toward validating this claim and show promising gains in Table 1, 2. We further test whether our representations mirror human notions of class relationships. We evaluate the Pearson correlation $r$ between the feature similarity matrix and the ground truth class structure matrix (Huang et al., 2021) in Figure 18, on ImageNet20. The results are given in Table 19. For ResNet50, we use the pre-trained model in torchvision and use the feature of the last layer after average pooling. DeiT-Base-cls is the class token of DeiT-Base (Touvron et al., 2021). These results indicate that Canonical Features capture inter-class relationships more faithfully than either CFG or supervised counterparts.

Diffusion inference produces reliable, human-like decisions (Jaini et al.; Li et al., 2023a) but is too compute-heavy for many deployments. *CaDistill* addresses this by transferring the CanoRep structure from the diffusion teacher to a lightweight student. This allows the student to have a similar semantic understanding. Acting as an amortized inference engine, the student approximates the teacher's reasoning while avoiding its computational cost. This is important in resource-constrained settings like autonomous driving, where fast, reliable recognition is critical.

We also envision alternative practical use cases of CLARID on itself.

- **Semantic augmentation.** Canonical Samples capture each class's core. By adding selected extraneous directions with controllable strength, we can generate high-diversity yet class-consistent variants. Such augmentations can complement underrepresented or long-tail categories during multi-modal pre-training. Figure 42 shows examples of the semantic augmentation.

- **Stronger visual grounding.** Canonical Samples suppress background clutter, offering clean contrasts for noisy real-world images. Combined with current visual pre-training pipelines, they can reduce reliance on spurious visual cues and improve grounding under domain shift.

- **Extract interpretable, compact class summaries and detect dataset bias.** Figure 6 and Section Q demonstrate that Canonical Samples capture the core class information, offering prototype-level class summaries. Canonical Samples highlight what the model considered irrelevant (background, co-occurring objects) versus indispensable to the class data, revealing potential biases of the dataset.

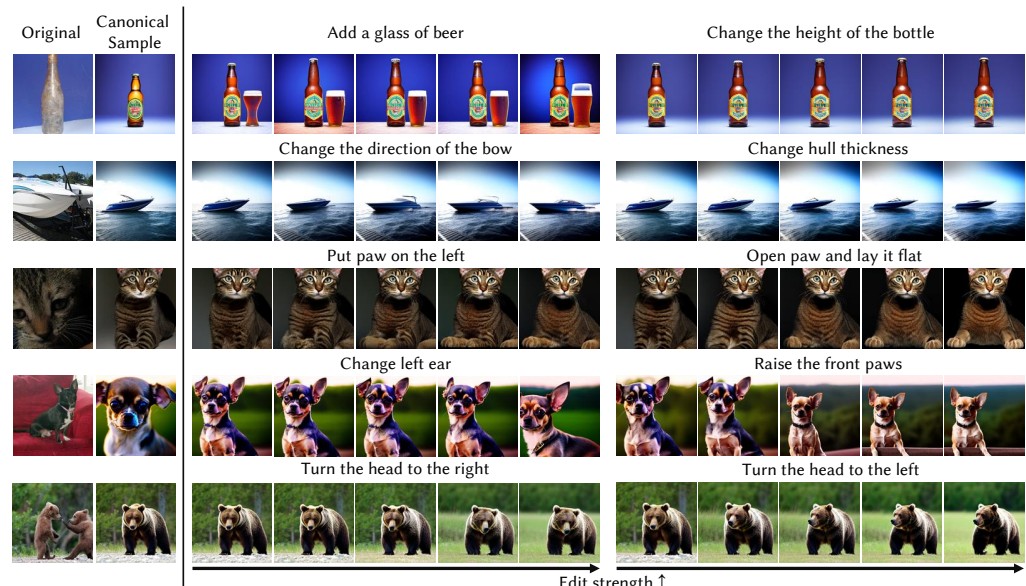

Figure 42: Semantic augmentation by adding extraneous directions back to CanoReps and decode, using Stable Diffusion 2.1. We show the effects of two extraneous directions for each sample, and different edit strengths. Note that all directions do not affect the class identity of the object.

For example, we spot that the Canonical Samples of the "Academic Gown" class (n02669723) always contain humans. After visual examination of the original class samples, we find that all images in this class have humans. This is an easily exploitable bias that can hinder model generalization. CLARID thus offers a diagnostic tool for dataset curation and auditing.

## P    ON MORE SOPHISTICATED FEATURE DISTILLATION FRAMEWORKS

In Section M.1, we demonstrate that our feature distillation loss outperforms the ones used in existing diffusion-based feature distillation frameworks. In this experiment, we use a single-layer distillation framework. That is, the alignment between the student and the teacher only happens at one layer, respectively. We do not consider more sophisticated feature distillation frameworks such as multi-layer alignments (Li et al., 2023b; Yang et al., 2024) due to limited computational resources. We believe investigating the combinations between *CaDistill* and different feature distillation frameworks is a promising future direction.

## Q    MORE VISUAL RESULTS

We provide more visualizations of CanoReps using Canonical Samples obtained from the DiT (Peebles & Xie, 2023) used in our *CaDistill* experiments on ImageNet (Deng et al., 2009), in Figure 43, 44, 45, 46, 47. All the classes are from ImageNet.

## R    BROADER IMPACT

CLARID introduces a new avenue into the field of DM research, focusing on interpreting the discriminative signals inside CDMs rather than directly probing the raw feature space. Our findings advance the interpretability of CDMs, contributing to safer usage of them. The application of CanoReps challenges the common assumption that the usage of DMs for discriminative tasks necessitates a large amount of data, providing new directions in diffusion-based feature distillation.

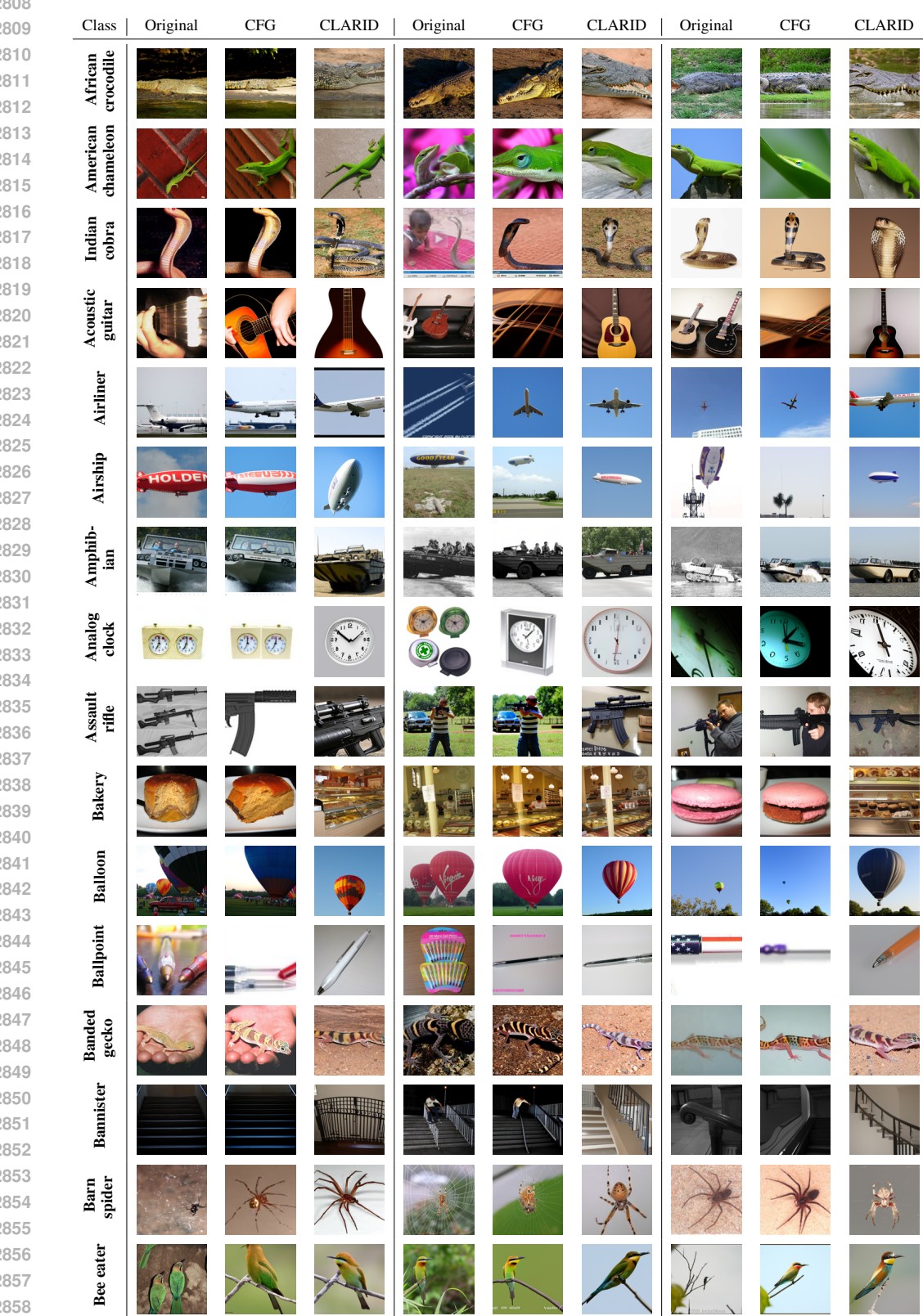

Figure 43: Visualizations of CanoReps using Canonical Samples obtained from the DiT (Peebles & Xie, 2023) used in our *CaDistill* experiments on ImageNet.

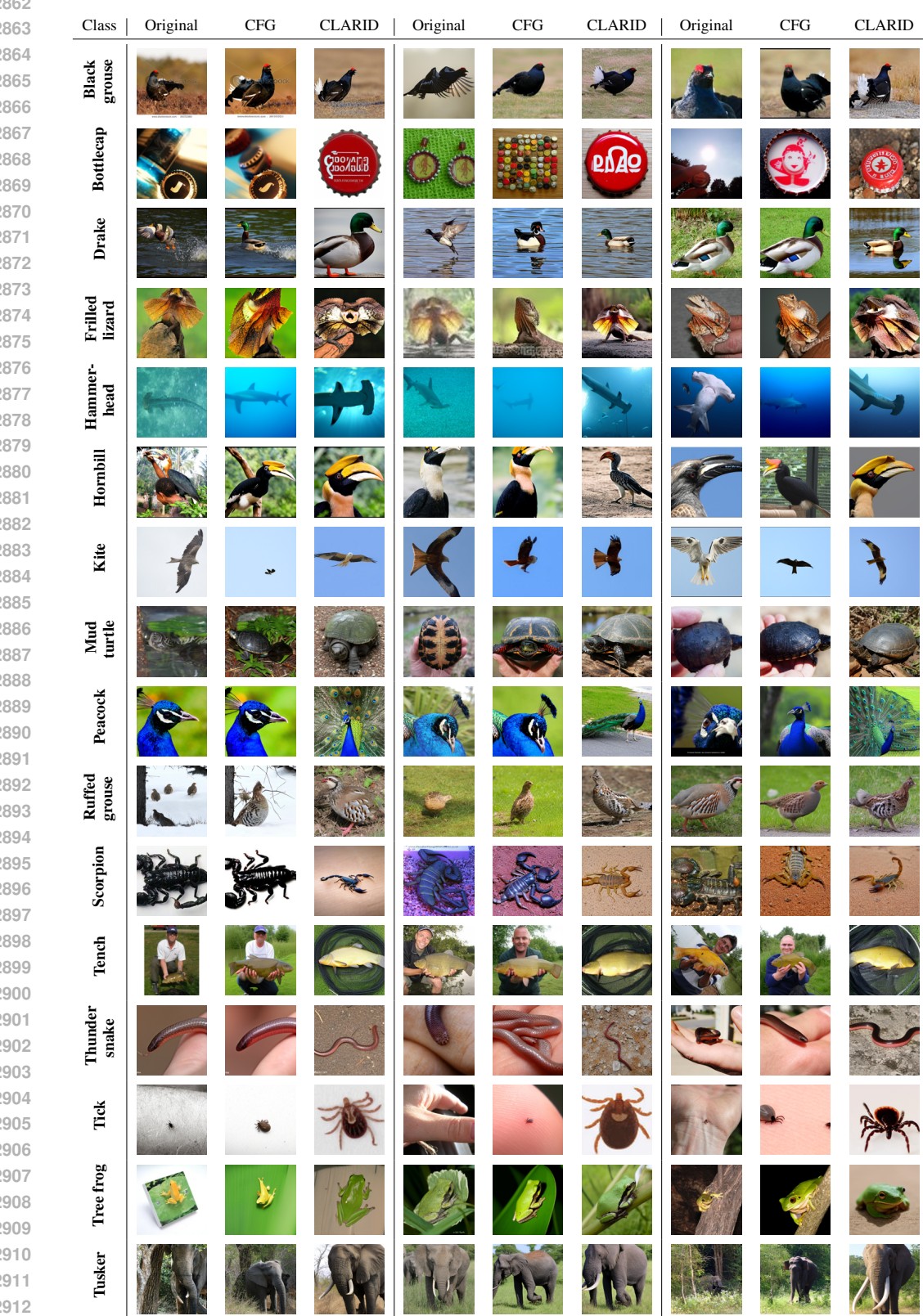

Figure 44: Visualizations of CanoReps using Canonical Samples obtained from the DiT (Peebles & Xie, 2023) used in our *CaDistill* experiments on ImageNet.

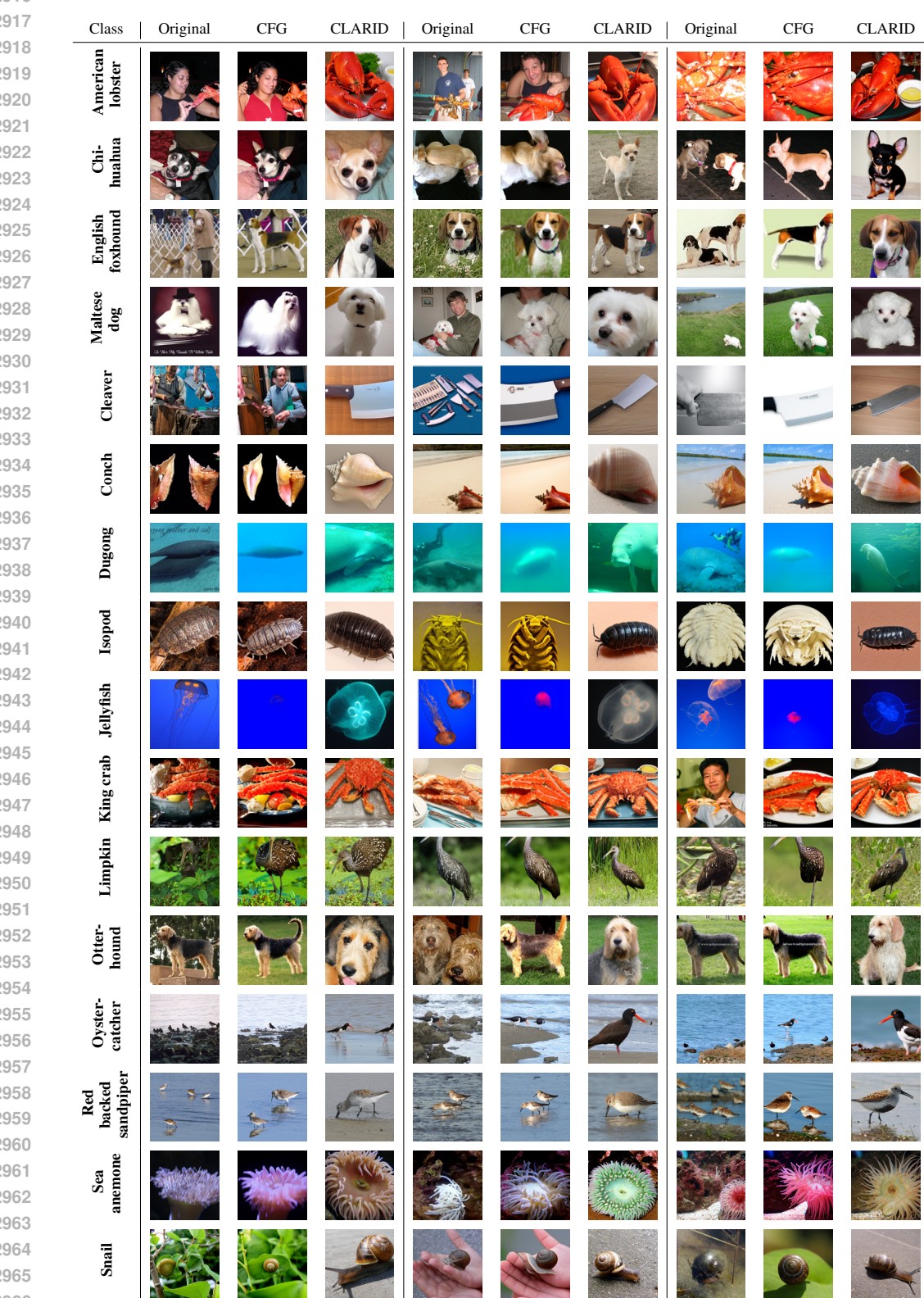

Figure 45: Visualizations of CanoReps using Canonical Samples obtained from the DiT (Peebles & Xie, 2023) used in our *CaDistill* experiments on ImageNet.

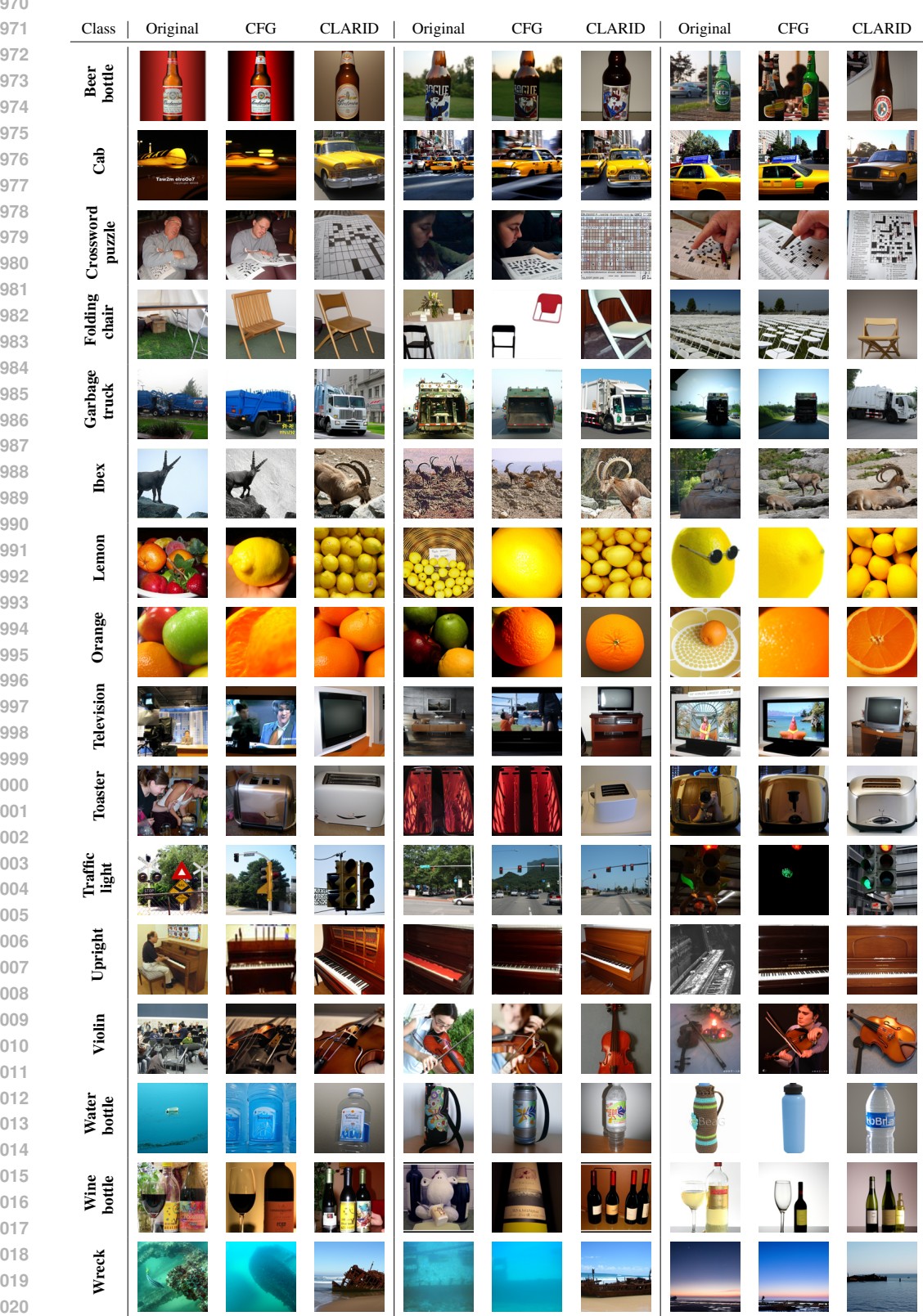

Figure 46: Visualizations of CanoReps using Canonical Samples obtained from the DiT (Peebles & Xie, 2023) used in our *CaDistill* experiments on ImageNet.

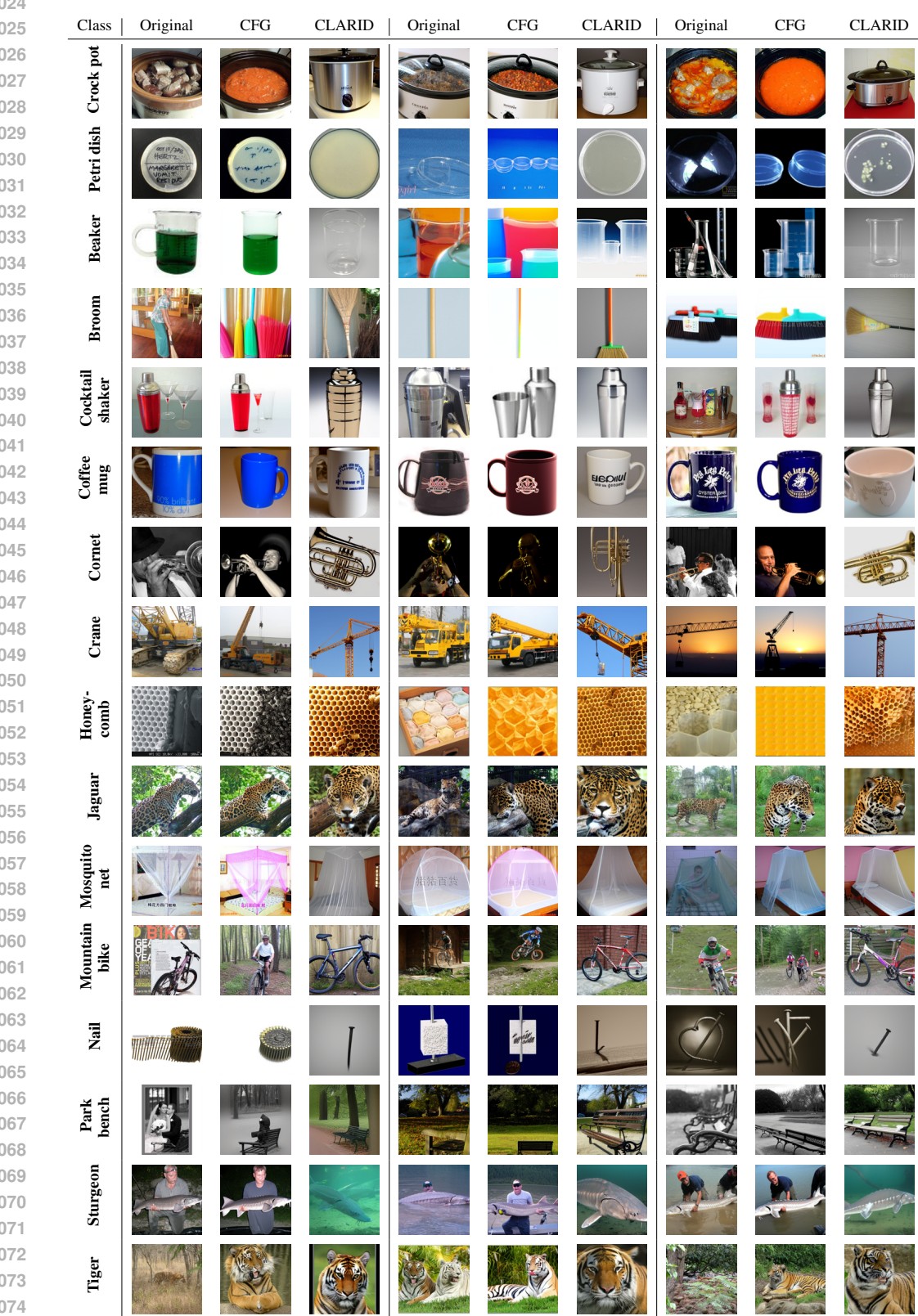

Figure 47: Visualizations of CanoReps using Canonical Samples obtained from the DiT (Peebles & Xie, 2023) used in our *CaDistill* experiments on ImageNet.

## S  LICENSE INFORMATION

### S.1  DATASETS INFORMATION AND LICENSE

- ImageNet1K (Deng et al., 2009). This dataset contains 1.28M training images and 50000 images for validation. We report the top1 accuracy on the 50000 validation images. License: Custom (research, non-commercial).
- ImageNet-C (Hendrycks & Dietterich, 2018). This dataset contains 15 types of 2D image corruption types that are generated by different algorithms. Higher accuracy on this dataset indicates a more robust model against corrupted images. License: CC BY 4.0.
- ImageNet-A (Djolonga et al., 2021). This dataset contains naturally existing adversarial examples that can drastically decrease the accuracy of ImageNet1K-trained CNNs. It is a 200-class subset of the ImageNet1K dataset. License: MIT license.
- ImageNet Reassessed Labels (ImageNet-ReaL) (Beyer et al., 2020): This is a dataset with 50000 reassessed labels of the ImageNet validation set, aiming at testing the in-distribution generalization ability of a classifier. License: Apache 2.0 License.
- CIFAR10-C (Hendrycks & Dietterich, 2018). This dataset contains 15 types of 2D image corruption types that are generated by different algorithms. Higher accuracy on this dataset indicates a more robust model against corrupted images. License: CC BY 4.0.

### S.2  MODEL AND CODE LICENSE

- Code for adversarial attacks (Kim, 2020): MIT License.
- PyTorch Image Model (Wightman, 2019): Apache 2.0 License.
- Diffusion Transformer (DiT) (Peebles & Xie, 2023): Attribution-NonCommercial 4.0 International.
- Stable Diffusion 2.1 (Rombach et al., 2022): CreativeML Open RAIL++-M License.
- EDM2 (Karras et al., 2024b): Creative Commons BY-NC-SA 4.0 license.
- Supervised Contrastive Learning (Khosla et al., 2020): BSD 2-Clause License.
- Swin (Liu et al., 2021): MIT License.
- FAN (Zhou et al., 2022): Nvidia Source Code License-NC.
- CIFAR10.1 (Recht et al., 2018): MIT License.

## T  LARGE LANGUAGE MODELS USAGE

We use Large Language Models to polish the writing.

