# OpenReview forum: "Canonical Latent Representations in Conditional Diffusion Models"
_ICLR.cc/2026/Conference — Submitted to ICLR 2026_

### Official Review · Reviewer_uuYL · 2025-10-30

**Soundness:** 2
**Presentation:** 2
**Contribution:** 2
**Rating:** 4
**Confidence:** 4

**Summary:**

The paper proposes CanoRep, a latent code within conditional diffusion models (CDMs) that preserves only the core class semantics. Through a training-free procedure called CLARID, the method estimates extraneous (non-discriminative) directions in the latent variable obtained during reverse diffusion, using the right singular vectors (Jacobian’s right singular vectors). These components are then removed via orthogonal projection to obtain the CanoRep (Eq. (2)). When decoded, CanoReps generate Canonical Samples, which are representative examples with minimal irrelevant context. From these, Canonical Features are extracted and used in the student model for CaDistill, which optimizes three objectives: Alignment loss, CanoRep clustering loss, and CKA-based representation alignment. Experimentally, even when using only 10% of the dataset as CanoReps on CIFAR-10 and ImageNet, the method demonstrates improved robustness and generalization.

**Strengths:**

- Clear problem definition & concise solution:
The paper clearly identifies the issue that CDMs (Conditional Diffusion Models) encode both class-relevant and class-irrelevant information. It proposes a simple yet general solution — extracting and removing non-discriminative directions via a Jacobian-based linear approximation. The method is broadly applicable regardless of architecture or sampler, and visualizations demonstrate that it conceptually works even under text conditioning and across various samplers/architectures (e.g., DDIM, EDM, U-ViT).

- Improved representation quality:
Using CanoRep improves representation quality, as evidenced by higher NMI scores and consistently more compact and well-separated clusters in UMAP visualizations (Figures 3 and 4). The method also visually reduces class-irrelevant artifacts compared to CFG (Figure 5).

- Practical benefits & data efficiency:
CaDistill achieves consistent improvements in both adversarial robustness (PGD, CW, APGD) and generalization (ImageNet-C, ImageNet-A, ReaL), even when using only 10% of the data as CanoReps (Table 1). In background perturbation experiments, it also shows advantages when the background is randomized or only the foreground is preserved (Table 2).

**Weaknesses:**

- **No theoretical analysis**: Fundamentally, this study considers the separation between class-relevant and class-irrelevant latent information. I believe that research of this kind would benefit from an information-theoretical explanation (See [1] for fundamentals). In relation to diffusion models, [2] provides an information-theoretic analysis of how latent information is partitioned. It would be helpful if the authors could include a discussion on this aspect, for example, by explaining the changes in mutual information.

- **Experimental scale**: It seems that using a T2I (text-to-image) diffusion model could also improve zero-shot discriminative tasks. I recommend including such experiments as well.

- Too many hyperparameters appear in Eq.6.

[1] (NeurIPS 18) Isolating Sources of Disentanglement in VAEs

[2] (ICLR 25) Diffusion Bridge AutoEncoders for Unsupervised Representation Learning

**Questions:**

See Weaknesses.

---

> ### Author Response · Authors · 2025-11-19
>
> We thank you for your constructive feedback and your recognition of our clear problem formulation, the simplicity, efficiency, and generality of our method, and the practical gains in representation quality as well as robustness. We now address your remaining comments in detail below.
>
> - **Theoretical analysis**
>
> We appreciate your suggestion, and here we provide an explanation of CLARID from the perspective of the change of mutual information and have **updated our paper** to include it in Appendix D. We denote $X$ as the image, $Y$ as the corresponding label, and $Z$ as the CDM feature at time step $t_e$. Because the CDM is trained to model the full distribution $p(X|Y)$, $Z$ contains information predictive of $Y$ as well as information about the remaining variability in $X$ (e.g. background, style, or texture) that is independent of $Y$. We have the following decomposition of the mutual information $I(Z;X,Y)$:
>
> $$I(Z;X,Y)=I(Z;X)+I(Z;Y|X)=I(Z;Y)+I(Z;X|Y)$$
>
> Hence,
>
> $$I(Z;X)=I(Z;Y)+I(Z;X|Y)-I(Z;Y|X)$$
>
> - $I(Z;Y)$ measures how much the latent Z is aligned with the class information.
> - $I(Z;X|Y)$ represents extra information $Z$ contains about $X$ given the class $Y$.
> - $I(Z;Y|X)$ is zero, assuming that the label is correct and unique (and hence $I(Z;Y|X) \le H(Y|X)=0$) in the dataset, and CLARID uses the ground truth dataset label.
>
> CLARID works by identifying and removing the high-variance components, i.e., extraneous directions, in $X$ given $Y$. By construction and by our editing experiments in Appendix D, **moving along those directions changes the image appearance while keeping the class identity unchanged**, so they are a proxy for directions that contribute significantly to $I(Z;X|Y)$ but not to $I(Z;Y)$.
>
> Denote the diffusion latent after removing the extraneous directions as $\tilde{Z}$. CLARID reduces the total information that $Z$ contains about $X$, as it removes certain things from $Z$, hence $I(\tilde{Z};X)<I(Z;X)$. Meanwhile, it aims to keep the class-relevant term $I(\tilde{Z};Y) \approx I(Z;Y)$ as unchanged as possible, through a fixed $Y$ during the whole process. From this perspective, CLARID implicitly creates an **internal information bottleneck [1] inside a pretrained CDM**. Therefore, the fraction of latent information that is label-relevant is increased by CLARID. As shown in Figure 3 and 27 in our submitted version (Figure 3 and 30 in the updated one), after removing the extraneous directions, the **normalized mutual information between the feature cluster assignments and the ground truth labels increases**, indicating that the remaining latent is more predictive of the semantic label $Y$.
>
> We further develop a toy generative model in Section D, including the corresponding probabilistic graphical model, in our **updated paper**. Consider that the full distribution $p(X|Y)$ can be decomposed into $p(Y)p(Z|Y)p(X|Z)$. The latent $Z$ can be further decomposed as $Z=S+E$, where $S$ depends on the label $Y$ and $E$ is independent of it. $S$ is known when $Y$ is given. Therefore, to model the full distribution $p(X|Y)$ given $Y$, the generative model **must encode all variance of $X$ in $E$, the label-independent component**, because $S$ is fixed when $Y$ is known. CLARID identifies those high-variance components given $Y$ and removes them from $Z$ to preserve core class information $S$. After removing $E$, $I(Z;X)$ decreases to $I(S;X)$ while $I(Z;Y)$ does not change and equals $I(S;Y)$, hence the information bottleneck effect.
>
> [1] Tishby, Naftali, et al. "The information bottleneck method." arXiv preprint physics/0004057 (2000).

---

> ### Author Response · Authors · 2025-11-19
>
> - **Theoretical analysis (Cont.)**
>
> **_Relationship to $\beta$-TCVAE and DBAE_**
>
> 1. $\beta$-TCVAE
>     - **Objective:** Learn disentangled latent factors **during training** of a VAE.
>     - **How:** It designs a new loss function for the generative model training. It focuses on the Total Correlation (TC) term within the ELBO decomposition during training. By increasing the weight on this TC term, the model explicitly penalizes entangled latent dimensions, thereby preserving information predictive of ground truth factors while discarding statistically dependent factors.
>
> 2. DBAE
>     - **Objective:** Resolve the **information split between the diffusion process and an auxiliary encoder**: Concentrate all sample information into the encoder output $z$ of a clean image, instead of splitting it between $z$ and the fully corrupted endpoint $x_T$.
>     - **How:** It modifies the training objective and architecture so that $x_T$ becomes a purely $z$-dependent, learnable endpoint. It requires an auxiliary encoder to encode the clean image to $z$ and a decoder to map $z$ to $x_T$. The entire approach is formulated **within an encoder-based diffusion framework**. The encoder, decoder, and diffusion model are jointly trained.
>
> 3. Ours (CLARID)
>     - **Objective:** Reveal and exploit a class-semantic subspace that is already present in pretrained CDMs. Prove that the class-essential features can be directly extracted **without any retraining or additional modules**.
>     - **How:** We compute the Jacobian of the pretrained CDM to identify extraneous directions along which moving in latent space changes $X$ while keeping $Y$ unchanged. We remove those label-invariant, high-variance directions from the full latent $Z$, yielding $\tilde Z$ that preserves class-relevant information $I(\tilde Z;Y) \approx I(Z;Y)$ while reducing label-independent variability $I(\tilde Z;X) < I(Z;X)$. By doing so, CLARID creates an internal information bottleneck without any retraining or additional modules.
>
> In summary, all three methods can be interpreted in an information-theoretic way, but they act on different parts of the generative pipeline. $\beta$-TCVAE and DBAE impose information constraints by redesigning objectives and architectures and then retraining the generative model. In contrast, CLARID operates purely on a fixed, pretrained CDM by creating an internal information bottleneck, revealing a class-essential semantic subspace that is already present rather than learning a new one.

---

> ### Author Response · Authors · 2025-11-19
>
> - **Experiments on T2I diffusion model**
>
> We thank the reviewer for raising this point. We conduct additional experiments with Stable Diffusion (SD) 2.1, using ResNet50 on ImageNet100. The text prompt template for SD is: _a photo of_ `class name`. We use the model-level hyperparameters found in Appendix H.2.1, i.e., the hyperparameters for CLARID on SD, to extract CanoReps for SD. Then, without further tuning, we apply CaDistill, using SD-generated images and features, with the same distillation hyperparameters as in Section 4.1 and Appendix M.7.
>
> |                | Clean Acc. | Adv. Acc. |
> |----------------|------------|-----------|
> | Baseline       | 86.50      | 15.94     |
> | CaDistill with SD | 88.02      | 28.84     |
>
> The results show that CaDistill generalizes well to Stable Diffusion, a CDM with a different architecture, training data, and text conditioning, without requiring extensive tuning, which supports the effectiveness and generality of our methods. The results also indicate that CaDistill benefits from a stronger underlying CDM. When using SD instead of DiT, the distilled student model achieves better performance.
>
> - **Hyperparameters in Eq. 6**
>
>   We acknowledge that Eq. 6 introduces several hyperparameters. However, we highlight that CaDistill is robust to different hyperparameter settings. Across extensive sweeps (Figure 33-35 in our submitted version, Figure 36-38 in the updated version), CaDistill **brings stable gains over a wide range of hyperparameters, reducing the burden of hyperparameter tuning** while confirming that each component contributes in a complementary way. In addition, our experiments with Stable Diffusion further suggest that CaDistill does not require careful per-model tuning to remain effective.
>
>   We also perform **targeted ablations to disentangle the role of each loss and its weight**. The proposed loss contains 3 components, $L_{align}$, $L_{cano}$, and $L_{dist}$.
>   - Section M.7.2 isolates the $L_{align}$ and $L_{cano}$ by making $\lambda_{cf}\in $ {0,1}. $L_{align}$ pulls each sample’s representation toward its canonical counterpart, while $L_{cano}$ enforces inter‑class separability among Canonical samples.
>   - We ablate the effect of $L_{dist}$ in Section M.7.3, showing that it contributes to the clean and adversarial accuracy.
>   - In Section M.7.4, a more fine-grained examination of different components is provided. $\lambda_{cs}$ and $\lambda_{dist}$ set the balance between student-space learning ($L_{cano}$, $L_{align}$) and teacher-guided distillation ($L_{cano}$). $\lambda_{cka}$ controls the emphasis between learning from normal samples and from Canonical samples, ensuring the student benefits from canonical structure without neglecting natural variability.

---

### Official Review · Reviewer_Nn1f · 2025-10-30

**Soundness:** 3
**Presentation:** 3
**Contribution:** 3
**Rating:** 4
**Confidence:** 4

**Summary:**

This paper proposes CLARID (Canonical Latent Representations in Conditional Diffusion Models), a framework designed to extract latent representations of data instances in conditional diffusion models. CLARID identifies directions of largest variation within images of the same class at a specific layer of the diffusion model using the right singular vectors of the Jacobian, and then removes those directions to retain the shared, class-common information as the latent representation.
Building upon these latent representations, the authors further propose CaDistill, a student-teacher training method that leverages canonical samples to obtain robust and generalized representations.

**Strengths:**

- The method introduces a simple yet effective idea: by utilizing the right singular matrix of the Jacobian, it identifies directions that cause significant local variation on the manifold and removes them, thereby isolating class-common representations.
The approach demonstrates improved adversarial robustness and resilience to distribution shift, showing consistent performance gains over baselines on both CIFAR-10 and ImageNet datasets.

- In addition, this paper demonstrates that the proposed method effectively removes background information and other class-irrelevant factors (e.g., the background context in the tench class of ImageNet), and this effect is clearly reflected in quantitative metrics.
Furthermore, the latent representations obtained by this method can be directly utilized for sampling in the base diffusion model, which constitutes one of its key advantages.

**Weaknesses:**

- Numerous hyperparameters:
The overall framework appears to be sensitive to hyperparameter choices. For instance, the process of obtaining Canonical Representations (CanoReps) depends on parameters such as $k$, $l$, and $t_{e}$; moreover, CaDistill involves additional hyperparameters ($\lambda_{align}, \lambda_{cano}, \lambda_{dist}$) during training.
This raises concerns about the practical applicability and generalizability of the proposed method, as the optimal combination of hyperparameters may vary for different base diffusion models, requiring extensive tuning for each setup.

- According to Appendix E, the parameter $l$ seems to be fixed. However, since l may vary with $t_e$, it would be helpful to see results where $k$ is fixed but $t_e$ is varied, showing how the corresponding $l$ changes and affects the outcome.

- It would also be interesting to investigate class-wise robustness. For example, showing the variation of top-$k$ directions across classes or presenting classification performance of the student model for different classes could strengthen the analysis.

- Regarding data fidelity, the paper states that CFG is applied after removing extraneous directions. It would be important to clarify whether the CFG scale used is greater than 1.
If it is, then it becomes difficult to disentangle whether the improvement in representation quality comes from the proposed method (with student network) itself or simply from a stronger CFG scale. Additional experiments verifying this would help clarify the contribution of CLARID.

**Questions:**

Please refer to the points raised in the Weaknesses section

---

> ### Author Response · Authors · 2025-11-19
>
> We appreciate your thoughtful feedback and your recognition of our method’s simplicity and effectiveness in robust representation learning. We now address your major concerns below.
>
> - **The hyperparameters and our method’s robustness to their choice**
>
> We acknowledge that our methods contain several hyperparameters. However, we argue that the proposed methods are **fault-tolerant and are effective in different hyperparameter settings**. We also provide **principled procedures to set those hyperparameters and perform targeted ablations to disentangle the role of each of them**.
>
> 1. **_The hyperparameters for obtaining CanoReps_**
>
> We emphasize that the diffusion time step and layer index parameters are unavoidable in all diffusion-based representation learning methods [1,2]. CLARID introduces **only a new parameter** $k$, which can be set in a principled, data‑driven way.
>
> - Layer index $l$: the choice follows established DM works [3,4,5] and validated in Figure 10 in our submitted version (12,13 in the updated version); discussion in Appendix E.
> - $t_e$: selected via model classification accuracy (Appendix H.1) with qualitative evidence (Figure 21, submitted; 23 in updated version) and a quantitative one (Figure 3). Cost: 0.3h (DiT), 1h (SD) on an Nvidia A100. Note that in Figure 3 and 27 in our submitted version (3 and 30 in the updated one), the $t_e$ selected by our method achieves the highest NMI among other $t_e$ values.
> - Number of extraneous directions $k$: automatic via our elbow method (Appendix H.2).
> - Total directions $n$: decided only once using a small dataset (Sec. H.2, Appendix Fig. 23 (26 in the updated one)); end‑to‑end selection cost: 4h (DiT), 9h (SD), on an Nvidia A100.
>
> We highlight that the procedure has a certain fault tolerance capacity, and we have elaborated on this point in Appendix H.2 (e.g., Figure 22 in our submitted version (25 in the updated one)). We show that changing the number of projected extraneous directions can still result in desired images, indicating that our method can be **robust to different hyperparameter choices**.
>
> [1] Li, Daiqing, et al. "Dreamteacher: Pretraining image backbones with deep generative models." ICCV. 2023.
>
> [2] Yang, Xingyi, and Xinchao Wang. "Diffusion model as representation learner." ICCV. 2023.
>
> [3] Jeong, Jaeseok, Mingi Kwon, and Youngjung Uh. "Training-free content injection using h-space in diffusion models." WACV 2024.
>
> [4] Kwon, Mingi, Jaeseok Jeong, and Youngjung Uh. "Diffusion Models Already Have A Semantic Latent Space." ICLR 2023.
>
> [5] Park, Yong-Hyun, et al. "Understanding the latent space of diffusion models through the lens of Riemannian geometry." NeurIPS 2023.
>
> 2. **_The hyperparameters in CaDistill_**
>
> We perform **targeted ablations to disentangle the role of each loss and its weight**. The proposed loss contains 3 components, $L_{align}$, $L_{cano}$, and $L_{dist}$.
> - Section M.7.2 isolates the $L_{align}$ and $L_{cano}$ by making $\lambda_{cf}\in \{0,1\}$. $L_{align}$ pulls each sample’s representation toward its canonical counterpart, while $L_{cano}$ enforces inter‑class separability among Canonical samples.
> - We ablate the effect of $L_{dist}$ in Section M.7.3, showing that it contributes to the clean and adversarial accuracy.
> - In Section M.7.4, a more fine-grained examination of different components is provided. $\lambda_{cs}$ and $\lambda_{dist}$ set the balance between student-space learning ($L_{cano}$, $L_{align}$) and teacher-guided distillation ($L_{cano}$). $\lambda_{cka}$ controls the emphasis between learning from normal samples and from Canonical samples, ensuring the student benefits from canonical structure without neglecting natural variability.
>
> Across extensive sweeps (Figure 33-35 in our submitted version (36-38 in the updated one)), CaDistill **brings stable gains** over a wide range of hyperparameters, **reducing the burden of hyperparameter tuning** while confirming that each component contributes in a complementary way.
>
> To show that our pipeline generalizes well to other diffusion models than DiT, we perform an experiment of CaDistill, using ResNet50 on ImageNet100 and **Stable Diffusion** (SD). The text prompt template for SD is: _a photo of_ `class name`. We follow the model-level hyperparameters found in Appendix H.2.1, i.e., the hyperparameters for CLARID on SD, to extract CanoReps for SD. Then, without further tuning, we apply CaDistill, using SD-generated images and features, with the same distillation hyperparameters as in Section 4.1 and Appendix M.7.
>
> |                | Clean Acc. | Adv. Acc. |
> |----------------|------------|-----------|
> | Baseline       | 86.50      | 15.94     |
> | CaDistill with SD | 88.02      | 28.84     |
>
> The results show that CaDistill generalizes well to Stable Diffusion, a CDM with a different architecture, training data, and text conditioning, **without requiring extensive tuning**, which supports the effectiveness and generality of our methods.

---

> ### Author Response · Authors · 2025-11-19
>
> - **More results of changing $l$**
>
> We appreciate your suggestion on adding the results where $t_e$ and $l$ vary with $k$ being fixed. We have **updated our paper** to include this result in Appendix E. The results confirm that our choice of layer index $l$ ensures an adequate change in the output image, or the background can remain unchanged in certain cases.
>
> - **Class-wise robustness and extraneous statistics**
>
> Thank you for pointing out this interesting problem. We use ResNet152 on ImageNet1K as in Section 4.1 to study class-wise robustness improvements. Our first question is whether the improvement is limited to a small set of classes or is spread across the entire dataset. To this end, we plot a histogram of the per-class percentage improvement in adversarial accuracy under AutoAttack (Appendix M.9 in the **updated version**). The histogram shows that almost all classes benefit from our method, indicating that the gains are broadly distributed rather than concentrated in only a few categories.
>
> We further check whether the most-improved or least-improved classes align with specific superclasses, such as "animals" or "artifacts". Appendix M.9 lists the 20 most improved and 20 least improved classes. The results show that these classes are not clustered within only a small number of superclasses, suggesting that the robustness improvement is not biased toward any particular semantic group.
>
> Moreover, we investigate the variation of extraneous directions within each class or across different classes. Specifically, we compute the cosine similarity between the top-10 extraneous directions that we obtain in the ImageNet20 experiment. In the intra-class case, we compute pair-wise cosine similarities between the top-10 directions and average them; In the inter-class case, we find an optimal one-to-one correspondence between the two sets of directions by solving the Hungarian matching problem, which maximizes the total pairwise similarity between matched directions. We conduct this experiment with $t_e \in $ {$0.4T,0.5T,0.6T,0.7T,0.8T,0.9T,T$}. The results are given in Appendix M.10 in the **updated version**. First, we observe that the similarity generally increases when $t_e$ becomes larger. The values at $t_e=0.9T$ and $T$ are consistent with the prior work [6], validating our implementation. We also find that intra-class similarity is consistently higher than inter-class similarity. This is intuitive, as images from the same class often share similar backgrounds (e.g., most photos of sea fish are taken in the sea), leading to more similar extraneous directions within a class.
>
> [6] Park, Yong-Hyun, et al. "Unsupervised discovery of semantic latent directions in diffusion models." arXiv-2023.
>
> - **CFG usage**
>
> We agree with you that it is important to disentangle the effect of CFG. We have included the details of the CFG magnitude in Section 4.1-Ablation studies as well as Appendix M.7.6 in our submitted version, and set it to 3 in our CaDistill experiments. We perform two controls on the effect of CFG.
>
> - First, we design a baseline method, CFGDistill (Table 1, Section 4.1), that shares all settings, including the CFG magnitude, with CaDistill except that the features and images are from pure CFG. In Table 1 and 2, we demonstrate that merely using CFG features and images will undermine the OOD and in-distribution generalization performance of the student, and make the student focus more on spurious background cues.
> - Second, we ablate the choice of the CFG magnitude in Section 4.1-Ablation studies and provide experiment details in Appendix M.7.6. We show the results in Figure 36 in our submitted version (Figure 39 in the updated version), which indicate that a stronger CFG magnitude does not imply a stronger student. Instead, it can largely undermine the student's performance.
>
> We have discussed the implications of using CFG in Appendix G.3 and a toy visual example in Appendix K, providing intuition on why CFG alone cannot improve representation quality as our method does. The main idea is that CFG on its own does not provide a meaningful class geometry for the student to learn, and an overly high CFG magnitude merely provides a trivial label-conditioned converging prior rather than rich class semantics.

---

> > ### Comment · Reviewer_Nn1f · 2025-11-28
> >
> > Thank you for your careful response. I have confirmed sufficient experimental evidence and rationale addressing the concerns I raised.
> >
> > I plan to raise my score to 6. The OpenReview system appears to be currently unavailable. I will adjust the score as soon as the system is back operational.

---

> > > ### Author Response · Authors · 2025-12-01
> > >
> > > Thank you for your careful review and for increasing your score!

---

### Official Review · Reviewer_9Ei2 · 2025-11-01

**Soundness:** 3
**Presentation:** 2
**Contribution:** 3
**Rating:** 6
**Confidence:** 3

**Summary:**

This paper aims to address the issue that conditional diffusion models often entangle class-defining features with irrelevant contextual information. The authors propose a training-free procedure to identify canonical latent representations and further introduce a diffusion-based feature distillation method to enhance the adversarial robustness and generalization ability of a student model.

**Strengths:**

1. The problem of extracting robust and interpretable representations in conditional diffusion models is important and timely.
2. The proposed pipeline is conceptually sound: it first identifies canonical latent representations that retain essential categorical information while removing non-discriminative signals, and then leverages these representations to guide the training of a student model with improved adversarial robustness and generalization.
3. Experiments on real-world image datasets demonstrate the effectiveness of both the canonical representation extraction and the subsequent distillation process.

**Weaknesses:**

1. The writing and structure of the manuscript could be improved. While the technical content is substantial, the dense presentation and frequent deferrals to the Appendix (e.g., Lines 254–269, 308–320) make the paper difficult to follow. A clearer exposition of the key steps in the main text would improve readability.
2. Experimental settings are insufficiently described. For instance, it is unclear how the “20 different ImageNet classes” (Line 259) were selected. Providing a rationale for this choice and clarifying the data preprocessing or sampling procedure would strengthen reproducibility. If class selection was random, multiple runs should be conducted and the variance reported (e.g., in Figure 3) to support claims of robustness and generalization.
3. The ImageNet classes used across different subsections are inconsistent, making it difficult to assess the generality of the findings.

**Questions:**

Please refer to "Weaknesses".

---

> ### Author Response · Authors · 2025-11-19
>
> We thank you for your positive feedback on our work and your acknowledgment of the importance of CLARID, the soundness of our pipeline, and the efficiency as well as practicality of CaDistill. We now address your questions.
>
> - **Writing improvements**
>
> We highly appreciate your suggestion on improving the paper writing. According to your suggestion, we have **updated the paper** and rewritten Section 3.2.1-Empirical Validation and Section 3.4 to remove some appendix references, ensuring the coherence of the main text.
>
> - **The ImageNet20 dataset**
>
> We agree that ImageNet20 is a new subset introduced in this work. The 20 selected classes have been listed in Appendix G in our submitted version. Our goal is to construct a subset with a clear structure while keeping the class-separation task neither too easy nor too hard. On the one hand, the classes should be dissimilar enough so that separation is meaningful; on the other hand, if they are too dissimilar, separation becomes trivial, and it is hard to draw reliable conclusions. To balance this, we start from the widely used 16-way ImageNet split [1,2,3,4], and randomly select one class from each of the 16 superclasses. We then increase the difficulty by randomly selecting 4 additional classes from the "bird" and "dog" superclasses, which contain the most subclasses. The resulting subset has a similar number of classes from the two main partitions of ImageNet, animals and artifacts, ensuring a well spread over ImageNet1K.
>
> Following your recommendation, we perform an ablation study on the choice of the classes. We select two different sets of classes from the 16-way ImageNet and perform the same analysis as in Figure 3, combining the results with those in our submitted version. We have **updated our paper to include the result, including a new Figure 3, text change in Section 3.2.1, and more information about the ImageNet20 dataset in Appendix G.** We observe that our adaptive choice of $k$ still achieves the best NMI among all fixed $k$. Although the trend of NMI when varying $k$ changes a bit (e.g., there is a local maximum at $t_e=0.8,k=3$), it does not affect our main conclusion, proving that our proposed method on hyperparameter selection is robust and can generalize well.
>
> [1] Geirhos, Robert, et al. "Generalisation in humans and deep neural networks." NeurIPS 2018.
>
> [2] Geirhos, Robert, et al. "ImageNet-trained CNNs are biased towards texture; increasing shape bias improves accuracy and robustness." ICLR 2018.
>
> [3] Subramanian, Ajay, et al. "Spatial-frequency channels, shape bias, and adversarial robustness." NeurIPS 2023.
>
> [4] Gavrikov, Paul, and Janis Keuper. "Can biases in ImageNet models explain generalization?." CVPR 2024.
>
> - **Different ImageNet classes across subsections**
>
> We agree that the ImageNet classes used across different subsections are not identical. This is intentional: each subset is chosen for a specific purpose, and in each case it is sufficient for that goal. First, we use a small subset, ImageNet20, to identify model-level hyperparameters such as the feature extraction layer, the feature recording time step, and the total number of extraneous directions. The small size of the dataset ensures efficiency. We show that hyperparameters selected on ImageNet20 lead to similar conclusions as those obtained by linear probing on larger datasets (Appendix G.1), which supports the use of ImageNet20 for this purpose. We also verify that these conclusions are robust to the choice of 20 classes.
>
> For the ablation study, we use ImageNet100, a standard and representative subset of ImageNet1K. Prior works, as well as our own experiments, indicates that trends observed on ImageNet100 transfer reliably to ImageNet1K (Appendix M.7).
>
> Finally, we conduct CaDistill experiments on the full ImageNet1K to demonstrate the effectiveness of our method on a large-scale, real-world dataset. Together, these three datasets are used in a complementary way: ImageNet20 for efficient hyperparameter selection, ImageNet100 for controlled ablations, and ImageNet1K for large-scale validation.

---

### Official Review · Reviewer_GCUn · 2025-11-01

**Soundness:** 2
**Presentation:** 2
**Contribution:** 2
**Rating:** 4
**Confidence:** 3

**Summary:**

This paper aims to improve downstream discriminative learning in conditional diffusion models by extracting robust and interpretable representations. To this end, the paper proposes CLARID, a training-free method for obtaining CanoReps, latent representations that capture conditional information while suppressing irrelvant signals. Through both quantitative and qualitative analyses, the paper demonstrates that these latent representations contain compact, claa-relevant infomration without encoding unnecessary background or non-class features. Building on this, the paper further propose CaDistill, a knowledge distillation framework that leverages these representations.

**Strengths:**

* The proposed approach of projecting onto otrhogonal directions using the singular vectors of Jacobian is a reasonable and elegant idea. It is also a novel contribution within the context of diffusion-based representation learning for enhancing class-discriminative information.

* The observed performance improvement when applying the method to knowledge distillation demonstrates its practical applicability.

**Weaknesses:**

* It would be beneficial to provide a clearer methodological or theoretical discussion on why CanoReps work effectively for feature distillation. It remains somewhat unclear why CanoReps sould be preferred over other possible representation extraction methods. For example, how would performance differ if alternative representations were used? In addition to quantiative results, qualitative analyses could further support the claimed effectiveness of CanoReps.

* CanoReps are randmoly selected per category for CaDistill. It would be helpful to include an ablation study examining this choice. For example, what happens if CanoReps corresponding to specific input images are used instead of randomly sampled ones?

* Section 3.5 mentions that CanoReps can be extended to text-conditioned models. In this context, it would be intersting to know whether the CaDistill framework can also be applied to text-image correspondence models, and if so, whether any preliminary results or insights are available.

* In the distillation experiments, comparisons are primarily made with other diffusion-based distillation methods. Includign results from other recent distillation approaches would provide a more comprehensive evaluation. This would also help justify the diffusion-based design by showing that it achieves superior performance despite additional computational cost involved in representation generation.

* The paper currently presents CaDistill as the only application of CanoReps. If the approach could be extended to other downstream tasks, such as transfer learning and image editing, it would strenghten the contribution and highlight broader applicability.

* The description of Figure 3 lacks sufficient details. The meaning of each curve is unclear from the current legend and text description.

* Some important methodological explanations and results are deferred entirely to the appendeix, which hinders comprehension. For example, Section 3.3 is difficult to follow without consulting Appendix K, even though it is presented as a subsection of the main text.

**Questions:**

Please provide discussions or clarifications on the points raised in the Weaknesses section.

In particular, I would like to understand the fundamental reason why CaDistill achieves superior performance. What underlying property of CanoReps makes it especially effective for distillation?

I am also intersted in how CanoReps could be applied beyond distillation and whether it can extend to other downstream tasks.

---

> ### Author Response · Authors · 2025-11-19
>
> We thank you for the valuable suggestions as well as the acknowledgment of the novelty of CLARID, and the practical applicability of CLARID through CaDistill. We now address your questions below.
>
> - **The reason why CanoReps are preferred**
>
> We highlight that CanoReps are proposed to extract core class information and suppress class-irrelevant context in conditional diffusion models (CDMs), a property that other representations do not explicitly target. Section 3.2 explains why this can lead to a class-information-preserving representation: the right singular vectors of the Jacobian matrix are the directions that **the CDM can move along while leaving the class conditioning unchanged.** By removing these "class-invariant" directions, CanoReps retain variations that matter for the class while discarding those that do not. We have also discussed the special advantage of CanoReps over other representations in Appendix G.3, and provided a toy visual example in Section 3.3 and Appendix K. The main idea is that CanoReps effectively encode the essential class structures while other representations do not. At a high level, this view is consistent with the idea of **generative similarity**, where meaningful object relations and human-like class structure arise from generative features rather than purely supervised or contrastive signals (see Appendix O for a more detailed discussion). We hope this elaboration provides clearer methodological reasons for preferring CanoReps in feature distillation.
>
> We also **compare CanoReps against several alternative representation choices in our feature distillation experiments**, including:
>
> - 1. DMDistill (Table 1). Using the raw features in CDMs for distillation.
> - 2. RepFusion (Table 1). Adaptively selecting the diffusion time step for diffusion feature extraction and distillation.
> - 3. CFGDistill (Table 1). Using purely CFG images in CaDistill.
> - 4. Class token in a pretrained ViT (Appendix M.4, Table 14). Using the class token in DeiT-III Huge for feature distillation.
>
> None of these alternatives includes an explicit mechanism to separate essential class information from spurious context. As a result, students distilled from these features tend to have worse generalization and robustness, and rely more on background cues. In contrast, CanoReps, which explicitly remove class-irrelevant directions, provide more focused guidance on what the student should learn.
>
> We further support this with **qualitative analyses**. Figure 4 and 18 in our submitted version (4 and 21 in the updated version) show that CanoReps form tighter and more semantically aligned clusters than CFG features or original CDM features, with cluster structures that follow the ground-truth labels. Figure 5 and 38-42 in our submitted version  (6 and 43-47 in the updated version) visualize the Canonical Samples generated from CanoReps, illustrating that they act as intuitive and compact summaries of each class rather than full, cluttered scenes. In addition, Section 3.3 and Appendix K presents a 2D toy example where the underlying class-defining factors are known: CLARID recovers these factors via CanoReps, whereas methods such as CFG or plain inversion retain class-irrelevant noise.
>
> - **Ablation studies on the choice of CaDistill samples**
>
> We thank the reviewer for raising this point. In CaDistill, CanoReps are randomly selected per class from the training set. This is a deliberate design choice: **our CLARID module is meant to extract essential class features without relying on any special properties of the input images**. As a result, CaDistill does not require a hand-crafted or tuned data selection strategy, which **broadens the applicability of the framework and makes it easier to use in practice.**
>
> To examine the effect of this random choice, we perform an ablation on ImageNet100 where we keep the number of CanoReps per class fixed but resample a different set of CanoReps from the training data and rerun CaDistill. The experimental setup and metrics are described in Appendix M.7.
>
> |                | Clean Acc. | Adv. Acc. |
> |----------------|------------|-----------|
> | Baseline       | 86.50      | 15.94     |
> | CaDistill Run1* | 87.46      | 25.70     |
> | CaDistill Run2 | 87.42      | 25.50     |
> | CaDistill Run3 | 87.56      | 25.86     |
> | Mean (Std) | 87.48 (0.059)      | 25.69 (0.15)     |
>
> The "Run1*" trial is the one used throughout our ablation studies, and we compare it against additional runs with different random selections. The results show that the performance across runs is very similar, indicating that CaDistill is robust to the random choice of CanoReps and does not depend on carefully chosen, image-specific canonical samples.

---

> ### Author Response · Authors · 2025-11-19
>
> - **CaDistill on Text-conditioned model**
>
> We appreciate the suggestion to demonstrate the applicability of CaDistill on text-conditioned CDMs. We split our answer into two parts: (1) An experiment that uses a text-conditioned CDM as the teacher in CaDistill; (2) Discussion on how CaDistill can be applied to improve the performance of text-image correspondence models, such as vision-language models (VLMs).
>
> 1. **_Text-conditioned CDM as the teacher in CaDistill_**
>
> We conduct additional experiments with Stable Diffusion 2.1, using ResNet50 on ImageNet100. The text prompt template for SD is: _a photo of_ `class name`. We use the model-level hyperparameters found in Appendix H.2.1, i.e., the hyperparameters for CLARID on SD, to extract CanoReps. Then, without further tuning, we apply CaDistill, using SD-generated images and features, with the same distillation hyperparameters as in Section 4.1 and Appendix M.7.
>
> |                | Clean Acc. | Adv. Acc. |
> |----------------|------------|-----------|
> | Baseline       | 86.50      | 15.94     |
> | CaDistill with SD | 88.02     | 28.84    |
>
> The results show that CaDistill generalizes well to Stable Diffusion, a CDM with a different architecture, training data, and text conditioning, without requiring extensive tuning, which supports the effectiveness and generality of our methods. The results also indicate that CaDistill benefits from a stronger CDM teacher. When using SD instead of DiT, the student model achieves better performance.
>
> 2. **_Insights into applying CaDistill to VLMs_**
>
> Current VLMs are known to rely on spurious visual cues [1,2]. Building on CaDistill, which focuses on essential class information, one could constrain the image encoder to align its core visual semantics with the text encoder, while explicitly suppressing background and style variations that do not support the text label. This could improve image-text alignment robustness, e.g., by reducing the tendency of VLMs to exploit such spurious cues in the visual branch. In addition, CLARID can generate multi-level image-text pairs by controlling the text prompts of the diffusion teacher (see examples in Appendix F.1), thereby providing more diverse and precise supervision for VLM training. Combining CLARID and CaDistill with current VLMs is therefore a promising way to reduce their reliance on spurious visual cues and to improve both performance and robustness.
>
> [1] Hosseini, Parsa, et al. "Seeing What's Not There: Spurious Correlation in Multimodal LLMs." arXiv 2025.
>
> [2] Ye, Wenqian, et al. "Mm-spubench: Towards better understanding of spurious biases in multimodal LLMs." NeurIPS 2024 Workshop RBFM
>
> - **Comparing with other distillation approaches**
>
> We have compared CaDistill with a supervised-ViT-based distillation baseline in Appendix M.4, under controlled conditions: (1) the teacher architecture is a ViT, (2) the teacher size is roughly 600M, and (3) the teacher is trained on ImageNet1K, all matching the DiT used in CaDistill. While distilling from this ViT does improve the adversarial robustness of the student, it hurts both in-distribution and OOD generalization. We also note that feature distillation from ViTs to CNNs remains an open problem (Appendix M.1), and recent methods in this direction achieve worse performance than CaDistill in our setting (details and related works in Appendix M.1, Table 11, and Appendix M.4).
>
> - **The broader applicability of CanoReps**
>
>     - We agree that our experiments mainly focus on CaDistill, a form of transfer learning via knowledge distillation, as one of the applications of CanoReps. We highlight that CaDistill can be broadly applicable. We discuss other potential applications in Section 5, and elaborate in Appendix O.
>
>     - First, the structure of CanoReps **aligns well with the human notion of class structures**, as shown in Table 19. It indicates that CanoReps can be used to identify the underlying semantic structures in a given dataset.
>     - The second one is to **perform semantic augmentation via editing the Canonical Samples**. We show some examples in Figure 37 in our submitted version (42 in the updated version), in which we add the identified extraneous directions with controllable strength back to CanoReps to generate **controllable and class-consistent variants**.
>     - Finally, **Canonical Samples highlight what the model considered irrelevant (background, co-occurring objects)** versus indispensable to the class data, **revealing potential biases of the dataset**. We present an example of identifying the bias in the "Academic Gown" class in ImageNet in Appendix O.
>
> - **Writing improvements**
>
>   We highly appreciate your suggestion on improving the paper writing. We have **updated the paper** to:
>
>   - Include a detailed description of Figure 3 in Section 3.2.1-Empirical validation.
>   - Put a simplified version of Appendix K in the place of Section 3.3 to illustrate our toy experiment to help understand CLARID better.

---

### Author Response · Authors · 2025-12-01

Dear reviewers,

Thank you very much for your constructive feedback. We are glad that you find the problem we study important and timely (**9Ei2, uuYL**), and that you recognize the novelty (**GCUn**), simplicity and effectiveness (**Nn1f, uuYL**), as well as the soundness (**9Ei2**) of CLARID in identifying core class semantics and improving representation quality in CDMs. We also appreciate that you find CLARID broadly applicable to different CDMs (**uuYL**).

We are further encouraged that you consider CaDistill practical and effective (**GCUn, 9Ei2, Nn1f, uuYL**) in improving both robustness and generalization of discriminative models, and that you find it data-efficient (**uuYL**).

As already detailed in our responses to each reviewer, we have revised the paper and added new analyses and experiments to address your comments more thoroughly. We highlight updates to the paper in $\textcolor{red}{\text{red}}$. Specifically, we:

- Include a more detailed description of Figure 3 in Section 3.2.1-Empirical validation.
- Put a simplified version of Appendix K in the place of Section 3.3 to illustrate our toy experiment to help understand CLARID better.
- Remove the multiple references to the Appendix in Section 3.2.1 and Section 3.4, to better expose the key steps of our method.
- Update Figure 3 to include the variance over multiple independent runs as the error bar.
- Add more details about the ImageNet20 dataset in Appendix G, including how the classes are selected and why this subset is suitable for our analysis.
- Include additional results of changing the layer index $l$ and time step $t_e$ in CLARID in Appendix E.
- Include additional results of class-wise robustness improvement in Appendix M.9, demonstrating that CaDistill brings stable gains over all classes.
- Include additional results of the variation between the identified extraneous directions in Appendix M.10.
- Include an information-theoretical analysis of the effects of CLARID in Appendix D, with a toy generative model. We also discuss how this connects and contrasts with $\beta$-TCVAE and Diffusion Bridge AutoEncoders (DBAE): they modify objectives and architectures during training, whereas CLARID operates on a frozen CDM to reveal an existing class-semantic subspace.

We plan to incorporate all of the new results and analysis in our revised manuscript. We hope our clarifications, new analyses, and additional experiments address the main concerns and further support the contributions of CLARID and CaDistill as general and practical frameworks for extracting and exploiting core class semantics in CDMs, and more broadly establish **_canonical representation learning_** as a complementary paradigm to current representation learning methods.

---

### Meta-Review · Area_Chair_vKbS · 2025-12-29

**Summary:**

Reviewers broadly agreed the paper tackles an important and timely problem—extracting class-relevant, robust representations from conditional diffusion models—and found the core idea (Jacobian/SVD-based removal of class-invariant, high-variance directions to form “canonical” representations) to be novel and potentially useful. However, the concerns that most informed my decision were (i) insufficient clarity and self-contained exposition in the main paper, with essential explanations and results deferred to the appendix; (ii) uncertainty about whether the observed gains are attributable to the proposed canonicalization/distillation pipeline versus confounding factors (notably the role of CFG and several interacting hyperparameters); and (iii) limited breadth of comparisons and evidence for broader applicability beyond the primary CaDistill setting, with some claimed extensions remaining largely speculative. Taken together, while the work is promising and empirically supported in the authors’ chosen setup, reviewers’ confidence was held back by presentation/reproducibility issues and by the extent to which key claims rely on appendix-heavy justification and selective experimental coverage.

**Reviewer Concerns:**

Concerns largely addressed by the rebuttal / author responses
	•	Hyperparameters and robustness (Nn1f, also echoed by uuYL): The authors provide additional sweeps/ablations and argue fault-tolerance, and they articulate more concrete procedures for selecting diffusion timestep/layer and number of projected directions. The rebuttal also emphasizes that only one truly new model-level parameter is introduced in CLARID, with others being standard in diffusion-representation work. This substantially reduces (though does not fully eliminate) the “too many knobs” concern.
	•	CFG confound (Nn1f): The authors clarify the CFG scale used, add a controlled baseline (CFGDistill) that matches CFG magnitude, and present a sweep over CFG scales suggesting CFG alone does not explain the gains and can even degrade performance. This addresses the core disentanglement request reasonably well.
	•	Dataset subset construction / variance (9Ei2): The authors clarify how ImageNet20 is constructed, why multiple subsets are used (IN20 for hyperparameters, IN100 for ablations, IN1K for validation), and add re-runs with variance/error bars across alternative class subsets. This improves reproducibility and addresses the inconsistency concern.
	•	Requested additional analyses (Nn1f, GCUn): The rebuttal adds or points to class-wise robustness analyses and stability to random CanoRep sampling, and provides additional qualitative/quantitative comparisons against several alternative representation/distillation choices (raw CDM features, RepFusion, CFG-only, ViT class token). These responses are helpful and move several issues from “missing” to “partially supported.”

Concerns partially addressed / still outstanding
	•	Main-paper clarity and reliance on appendix (GCUn, 9Ei2): Although the authors state they rewrote sections, expanded figure explanations, and moved a simplified toy example into the main text, the fundamental concern remains: core intuitions, methodological details, and some key evidence still appear appendix-dependent and hard to evaluate from the main paper alone. This continues to limit confidence.
	•	Why CanoReps are the “right” object for distillation (GCUn): The rebuttal provides intuition and some additional comparisons, but the explanation remains somewhat post-hoc and does not fully nail down a principled, falsifiable reason that separates CanoReps from other plausible representation extraction strategies under controlled conditions.
	•	Breadth/strength of baselines, especially beyond diffusion-based distillation (GCUn): The rebuttal mentions a ViT-based distillation baseline and notes limitations of ViT→CNN feature distillation, but the evaluation still feels narrow relative to the claims of generality and the computational overhead of the approach. Stronger, more standard distillation/robustness baselines under matched compute would further strengthen the case.
	•	Theoretical framing (uuYL): The added information-theoretic discussion is a useful perspective, but it reads more as interpretive support than as a theory that predicts outcomes or guides design choices; it does not fully resolve the “no theory” concern.
	•	Broader applicability beyond CaDistill (GCUn): The rebuttal outlines plausible extensions (structure discovery, semantic augmentation, bias detection, potential VLM use), but the evidence remains largely centered on CaDistill; most extensions are not demonstrated empirically and thus remain aspirational.

**Reviewer Scores:**

Below I list each reviewer’s original score and my estimate of what they would likely converge to given the rebuttal and (partial) discussion signals visible in the thread.
	•	Reviewer Nn1f (original: 4) → 6
Rationale: This reviewer explicitly stated they confirmed sufficient experimental evidence/rationale addressing their concerns and planned to raise the score to 6 once the system allowed. So the most reasonable estimate is 6.
	•	Reviewer 9Ei2 (original: 6) → 6 (likely unchanged)
Rationale: Their main issues were writing/structure and ImageNet20 selection/inconsistency. The authors’ response directly targets these (rewriting, clarified subset construction, added multiple class-set runs and variance). This reviewer was already at 6 and framed the paper as acceptable but borderline; I expect they would remain at 6 rather than move strongly upward.
	•	Reviewer GCUn (original: 4) → 4–5 (most likely 4, at best 5)
Rationale: The rebuttal addresses several concrete requests (more explanation, additional comparisons, ablation on random CanoRep selection, SD teacher experiment, writing improvements). Still, their core reservations—mechanistic justification for CanoReps’ superiority, broader baseline coverage, and limited demonstrated breadth beyond CaDistill—are only partially resolved. I would expect at most a small upward movement, but it’s plausible they remain at 4.
	•	Reviewer uuYL (original: 4) → 4–5 (most likely 4)
Rationale: The rebuttal adds an information-theoretic appendix and the requested T2I/Stable Diffusion experiment, which should help. However, concerns about “too many hyperparameters” and limited theory are only partially alleviated (theory is still largely interpretive; complexity remains). I would expect no more than a +1, and it’s plausible they stay at 4.

---

### Decision · Program_Chairs · 2026-01-26

Reject